# A centrally positioned cluster of multiple centrioles in antigen-presenting cells fosters T cell activation

Isabel Stötzel [1], Ann-Kathrin Weier [1], Apurba Sarkar [2], Subhendu Som[2], Luisa Bach [3], Peter Konopka[1], Eliška Miková [4,5], Shaunak Ghosh [1], Jan Böthling [1], Mirka Homrich [1], Laura Schaedel [6], Uli Kazmaier [7], Konstantinos Symeonidis[8], Stefan Ebner[9], Philip Weidner [10], Zeinab Abdullah[8], Felix Meissner [9], Stefan Uderhardt [10], Miroslav Hons[4], Dirk Baumjohann [3], Raja Paul [2], Heiko Rieger [11] ✉ & Eva Kiermaier [1,12,13] ✉

Cellular polarization plays a crucial role in regulating immunological processes and is often associated with reorientation of the centrosome. During immune synapse formation, centrosome repositioning in lymphocytes assists in T cell activation. While a single centrosome, consisting of two centrioles, is present in T cells, antigen-presenting cells such as dendritic cells amplify centrioles during maturation and immune activation. How centriole amplification in antigen-presenting cells affects immune synapse formation and T cell activation is unclear. In this study, we combine experimental data with mathematical and computational modelling to provide evidence that extra centrioles in dendritic cells form over-active microtubule organizing centers, which cluster during dendritic cell-T cell interactions and, unlike in T cells, localize close to the cell center. Perturbing either centrosome integrity or centriole numbers and configuration in dendritic cells results in impaired T cell activation. Collectively, our results highlight a crucial role for centriole amplification and optimal centrosome positioning in antigen-presenting cells for controlling T cell responses.

The centrosome is a key organelle in eukaryotic cells, serving as the primary microtubule-organizing center (MTOC) in most animal cells. Structurally, centrosomes consist of two centrioles connected by a flexible linker, and surrounded by a dense layer of pericentriolar material (PCM)[1,2]. This organization allows the centrosome to effectively nucleate and anchor microtubule (MT) filaments, which are essential for maintaining cell shape, enabling intracellular transport, and facilitating cell division. Typically, centriole numbers are tightly

[1]Life and Medical Sciences (LIMES) Institute, Immune and Tumor Biology, University of Bonn, Bonn, Germany. [2]School of Mathematical & Computational Sciences, Indian Association for the Cultivation of Science, Kolkata, West Bengal, India. [3]Medical Clinic III for Oncology, Hematology, Immuno-Oncology and Rheumatology, University Hospital Bonn, University of Bonn, Venusberg-Campus 1, Bonn, Germany. [4]BIOCEV, First Faculty of Medicine, Charles University, Vestec, Czech Republic. [5]Faculty of Science, Charles University, Prague, Czech Republic. [6]Department of Experimental Physics and Center for Biophysics, Saarland University, Saarbrücken, Germany. [7]Organic Chemistry, Saarland University, Saarbrücken, Germany. [8]Institute of Experimental Immunology, University Hospital of Bonn, Bonn, Germany. [9]Institute of Innate Immunity, Department for Systems Immunology and Proteomics, Medical Faculty, University of Bonn, Bonn, Germany. [10]Department of Internal Medicine 3 - Rheumatology and Immunology, Deutsches Zentrum für Immuntherapie (DZI), Friedrich-Alexander University Erlangen-Nürnberg (FAU) and Universitätsklinikum Erlangen, Erlangen, Germany. [11]Department of Theoretical Physics and Center for Biophysics, Saarland University, Saarbrücken, Germany. [12]Department of Medicine 1/CITABLE, Friedrich-Alexander University Erlangen-Nürnberg (FAU), Erlangen, Germany. [13]Deutsches Zentrum Immuntherapie (DZI), Universitätsklinikum Erlangen, Erlangen, Germany. ✉e-mail: heiko.rieger@uni-saarland.de; eva.kiermaier@uni-bonn.de

regulated during the cell cycle, leading to two centrioles (one centrosome) in G1 phase and four centrioles (two centrosomes) before mitosis[3–5]. Yet, centriole numbers can vary significantly during organismal development and differentiation into specialized cell types, leading to cells that contain either no centrosome or multiple[6–8].

While originally named according to its position at the geometrical center of the cell, it is now well established that centrosome repositioning to the cell cortex is a prerequisite for various fundamental cellular processes such as polarized secretion, directional migration, asymmetric cell division, and immune surveillance[9–12]. Within the adaptive immune system, T cells play a central role in orchestrating antigen-specific immune responses. T cell activation is facilitated by the formation of the immune synapse (IS), a specialized signaling interface between antigen-presenting cells (APCs) and T cells. The formation of the IS starts with the recognition of a cognate antigen by the T cell receptor, which is presented via major histocompatibility (MHC) complexes on the surface of the APC. This interaction drives the spatial organization of receptors, adhesion molecules, and signaling complexes into distinct supramolecular activation clusters (SMACs), structuring the IS into a highly ordered, functional domain[13,14]. Formation of the IS is further coupled to a major reorganization of the T cell's MT cytoskeleton: antigen recognition results in the repositioning of the T cell's centrosome from the uropod to a position adjacent to the IS[15,16]. Centrosome reorientation toward the IS strictly depends on antigen recognition by the cognate T cell receptor[17] and is accompanied by the movement of other organelles, such as the Golgi apparatus, toward the IS[18–20]. Polarization of Golgi-derived vesicles is considered to facilitate directional vesicle release toward the target cell[19,21]. Moreover, MTOC reorientation in T cells has been demonstrated to be required for sustained T cell receptor signaling downstream of IS formation[22].

While the role of centrosome polarization in T cells is well-characterized, far less is understood about centrosome function and dynamics in APCs. Dendritic cells (DCs) represent the most potent APCs of the innate immune system, which have the unique ability to activate T cells in vivo[23]. They are strategically located in peripheral tissues, where they patrol the environment for invading microbial pathogens. Upon antigen encounter, DCs become highly migratory and translocate to draining lymph nodes to present peripherally acquired and processed antigens to naïve T cells. Besides antigen presentation, DCs shape T cell responses by secreting large amounts of soluble cytokines that activate T cells[24].

There is accumulating evidence that functional specialization of the DC MT cytoskeleton plays a pivotal role in determining their immunostimulatory function[25,26]. Upon antigen encounter, DCs enter a robust cell cycle arrest, which can be accompanied by the acquisition of additional centrioles in G1 phase[26]. Extra centrioles facilitate persistent migration of cells along chemotactic cues and enhance cytokine secretion, yet how these additional centrioles influence IS formation and T cell activation remains largely unknown. In particular, it is unclear how multiple centrioles are organized during the formation of antigen-specific immune synapses, and how the resulting MT architecture in APCs influences T cell activation.

In this study, we investigate the functional organization of multiple centrioles and centrosomal MTs in murine DCs during antigen-specific T cell responses in single cells and within lymphoid tissues. Our interdisciplinary approach, combining experimental data with computational modeling, reveals optimal centrosome positioning in APCs and how centrosome integrity and function impact efficient T cell activation.

## Results

### Centrosome and MT integrity in DCs are required for T cell activation

To address whether an intact centrosome in APCs is a prerequisite for efficient T cell activation, we first sought to pharmacologically interfere with centrosome integrity specifically in APCs prior to T cell conjugation. To this end, we used antigen-presenting DCs as a model cell type for studying antigen-specific T cell activation. To obtain sufficiently large numbers of cells, we generated DCs from the bone marrow of Centrin-2 (CETN2)-GFP expressing mice in the presence of granulocyte-macrophage colony-stimulating factor (GM-CSF) and stimulated the cells with lipopolysaccharide (LPS) overnight to induce cell maturation. As mature bone marrow-derived DCs (BMDCs) are a heterogenous population of cells consisting of diploid and tetraploid cells as a result of an incomplete mitosis[26], we separated BMDCs based on DNA content into diploid (2 N) and tetraploid (4 N) cells by flow cytometry and concentrated exclusively on the 2 N cell fraction for all further experiments (Supplementary Fig. 1a). Sorted 2 N DCs were further loaded or not loaded with different concentrations of the model antigen ovalbumin-peptide (OVAp) and incubated with naïve CD4+, OVA-specific T cells, which express a transgenic T cell receptor recognizing the OVAp (OT-II T cells). Under these experimental conditions, OVAp-loaded DCs efficiently activated naïve T cells as measured by surface levels of early activation markers such as CD69 and CD62L downregulation, as well as T cell proliferation assessed by proliferation-mediated dilution of the fluorescent dye carboxyfluorescein succinimidyl ester (CFSE)[27] (Supplementary Fig. 1b-e).

To specifically target centrioles in DCs, we further treated BMDCs with the polo-like kinase 4 (PLK4) inhibitor Centrinone during differentiation and maturation[28] (Fig. 1a). Centrinone treatment efficiently depleted CETN2-GFP+ foci in more than 60% of all mature cells but did not interfere with terminal differentiation of DCs as determined by upregulation of DC-specific cell surface markers such as CD11c and MHCII (Fig. 1b and Supplementary Fig. 2a,b). Note that Centrinone-treatment solely inhibits the generation of new procentrioles but does not deplete existing centrioles leading to a mixed population of cells that contain either no or one centriole, two, three or four, or more than four centrioles (Fig. 1b). When probing cells with antibodies against γ-tubulin to monitor PCM proteins surrounding the centrioles, we found residual staining present in Centrinone-treated cells, which appeared less intense than in untreated controls (Fig. 1c). This indicates some degree of residual PCM organization after Centrinone treatment. To confirm the lack of centrioles at higher resolution, we carried out expansion microscopy of Centrinone-treated and control cells. Mature DCs were immobilized on poly-L-lysine-coated cover slips and stained against acetylated (ac)-tubulin to identify centrioles, γ-tubulin to monitor PCM proteins, and N-Hydroxysuccinimide (NHS) ester to nonspecifically label free amino groups of proteins. In control samples, we routinely found cells with two, four, and more than four barrel-shaped centrioles identified by ac-tubulin and NHS ester staining, that were surrounded by PCM proteins (Fig. 1d, left). In the presence of PLK4-inhibition, ac-tubulin staining was not detectable, but cells had pronounced PCM structures shown by the presence of γ-tubulin (Fig. 1d, right). Similarly, NHS ester staining did not yield prominent barrel-shaped structures that co-localized with ac-tubulin staining in Centrinone-treated cells. These results indicate that pharmacological inhibition of PLK4 leads to depletion of centrioles but maintains part of the centrosomal PCM in DCs. In line with these findings, residual PCM organization has recently been described in cytotoxic T lymphocytes after genetic depletion of centrioles[29].

The presence of γ-tubulin in the absence of centrioles prompted us to further examine whether MT organization and architecture was affected by the lack of centrioles. Immunostaining of Centrinone-treated and control cells against α-tubulin revealed that acentriolar cells exhibited the same number of MT filaments within defined areas around PCM proteins compared to control cells (Fig. 1e and Supplementary Fig. 2c). Moreover, the overall number of MTOCs was unaltered in the presence of Centrinone (Fig. 1e), demonstrating that under our assay conditions, DCs lacking centrioles are able to nucleate and organize MT filaments to form a functional MTOC. When co-culturing

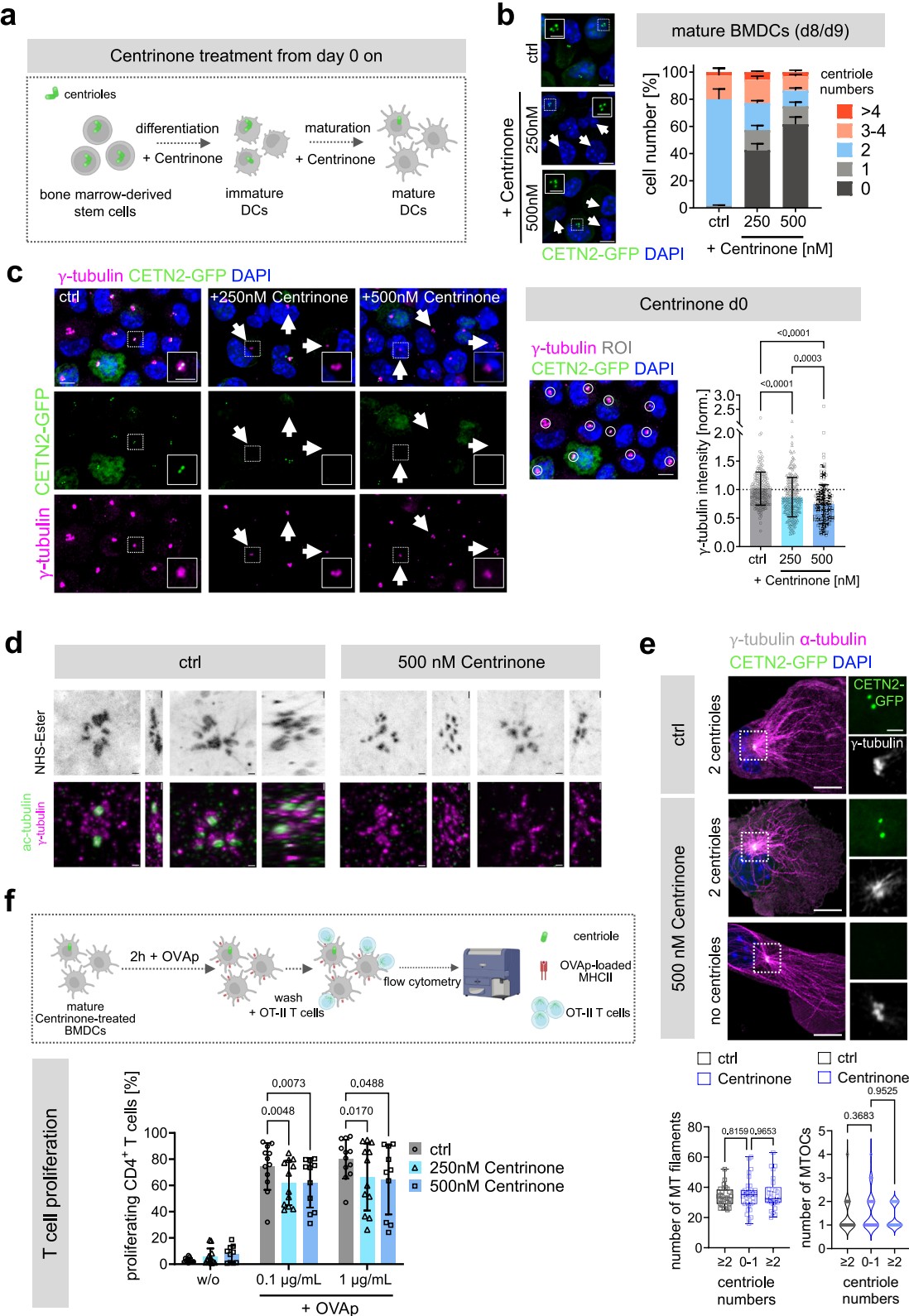

antigen-loaded Centrinone-treated cells with OT-II T cells, we found that cells lacking centrioles exhibited a significantly diminished capacity for inducing CD4+ T cell proliferation compared to control cells (Fig. 1f). These results reveal that centrioles and PCM integrity in APCs are essential for efficient T cell activation.

To further analyze the role of the MT cytoskeleton during T cell activation, we next perturbed MT growth in DCs by pre-treating cells with the MT destabilizing agent pretubulysin[30,31]. While permanent presence of pretubulysin for 24 h efficiently depolymerized MT filaments in DCs, drug wash-out led to the regrowth of single filaments after 24 h post-treatment (Fig. 2a). Of note, pretubulysin wash-out did not restore MT networks to the level of control cells, but led to fewer MT filaments growing predominantly from a single MTOC (Fig. 2b). Regrowing MTs appeared significantly shorter and less straight

**Fig. 1 | Centriole integrity in DCs is required for efficient T cell activation.**
**a** Schematic representation of experimental set-up of Centrinone-treatment.
**b** Confocal images of Centrinone-treated and control cells. Maximum z-projections of merged channels of CETN2-GFP (green) and DAPI (blue) are shown. White arrowheads indicate cells without centrioles. Scale bars, 5 $\mu$m. Insets show magnified region of centrioles (CETN2-GFP). Scale bars, 2 $\mu$m. Graph shows quantification of centriole numbers in mature diploid DCs after Centrinone treatment (day 8/9). Mean values ± s.d. of three independent experiments with $N = 224/259/176$ (ctrl), 207/263/135 (250 nM Centrinone), 247/210/146 (500 nM Centrinone) cells are displayed. **c** Left: immunostaining of mature CETN2-GFP BMDCs against γ-tubulin after 250 and 500 nM Centrinone treatment. Maximum z-projections of merged and individual channels of CETN2-GFP (green), γ-tubulin (magenta) and DAPI (blue) are shown. Scale bar, 5 $\mu$m. White arrowheads indicate cells without centrioles but with prominent γ-tubulin foci. Insets show magnified region. Scale bar, 2 $\mu$m. Middle: example of ROIs drawn around defined areas of γ-tubulin⁺ foci to quantify γ-tubulin signal intensity. Scale bar, 5 $\mu$m. Right: graph shows relative mean values normalized within each condition to cells with two centrioles ± s.d. Each data point represents one cell derived from one representative experiment out of three independent experiments. $N = 184$ (ctrl) /211 (250 nM Centrinone) /186 (500 nM Centrinone) cells. Dotted line drawn at 1.0. *P* values from one-way Anova with Dunnett's multiple comparisons. **d** Expansion microscopy of mature control or Centrinone-treated (500 nM) CETN2-GFP BMDCs. Top panels: NHS-Ester staining. Bottom panels: merged channels of acetylated (ac)-tubulin (green) and γ-tubulin

(magenta). Images are shown as top view and right view. Scale bars, 1 $\mu$m.
**e** Immunostaining of MTs in Centrinone-treated and control cells. Upper panel: Maximum z-projections of mature CETN2-GFP (green) BMDCs stained against γ-tubulin (white) and α-tubulin (magenta). Nuclei were counterstained with DAPI (blue). Scale bars, 10 $\mu$m. Magnifications of the indicated regions show individual channels of CETN2-GFP (green) and γ-tubulin (white). Scale bars, 2 $\mu$m. Lower panel: quantification of MT filaments (left) and MTOCs (right) in mature CETN2-GFP expressing BMDCs after Centrinone treatment. Left graph shows median, interquartile range and minimum to maximum values derived from three independent experiments. $N = 28$ (ctrl) / 32 (Centrinone: 0-1 centrioles)/ 26 (Centrinone: ≥2 centrioles). *P* values from one-way Anova with Dunnett's multiple comparisons. Right graph shows median and distribution of data points from three independent experiments. $N = 30$ (ctrl) /40 (Centrinone: 0-1 centrioles) /36 (Centrinone: ≥2 centrioles). *P* values from Kruskal-Wallis test with Dunn's multiple comparisons. In both graphs each data point represents one cell. **f** Upper panel: Schematic representation of experimental workflow of OT-II transgenic T cell activation by Centrinone-treated, mature BMDCs loaded with OVAp. Lower panel: Quantification of proliferating T cells after co-culture with Centrinone-treated DCs. Graph displays mean values ± s.d. Each data point represents one independent experiment with $N = 10.000$ cells analyzed per condition. Cells were derived from three different mice. *P* values from two-way Anova with Dunnett's multiple comparisons.
**a, f** Schematic pictures created with BioRender. Source data are provided as a Source Data file. ctrl: control, d: day, norm.: normalized, w/o: without.

---

compared to control-treated cells (Fig. 2c) indicating that pretubulysin treatment is, to some extent, irreversible. This allowed us to study the role of MTs specifically in DCs, while leaving the T cell's MT cytoskeleton unaffected (Supplementary Fig. 2d, e).

To this end, we first loaded mature DCs with OVAp and subsequently treated the cells for 1 h with pretubulysin to disassemble MT filaments after antigen loading (Fig. 2d). Pre-treated and control DCs were either washed two times (wash-out) or directly (w/o wash-out) co-cultured with OT-II T cells. To assess T cell activation, we first measured IL-2 cytokine secretion, which is predominantly produced by activated T cells[32], after 24 h of co-culture by ELISA. We found that IL-2 levels were markedly reduced in the presence of or after pretubulysin wash-out indicating reduced T cell activation after MT perturbation in DCs (Fig. 2e). Note that IL-2 levels were still below the detection limit after 24 h of co-culture when loading the cells with 0.1 $\mu$g/mL OVAp. Similar to IL-2 secretion, OX40 cell surface levels, a prominent T cell activation marker[33], were significantly reduced on T cells after pretubulysin treatment further confirming diminished T cell stimulation after MT perturbation in APCs (Fig. 2e). To address whether reduced T cell activation impacts subsequent T cell proliferation, we measured T cell expansion by proliferation-mediated dilution of CFSE. In accordance with reduced IL-2 levels, T cell proliferation decreased from 62 % in control samples to 27 % after pretubulysin treatment and wash-out in the presence of 0.1 $\mu$g/mL OVAp (Fig. 2f). These results highlight that MT integrity in DCs is an immediate prerequisite for efficient T cell activation. In summary, our results emphasize the importance of an intact centrosome and centrosomal MT array in APCs for eliciting CD4⁺ T cell responses.

### Enhanced MTOC activity in DCs with multiple centrioles during IS formation

Our data highlight a crucial role of the centrosome and the MT cytoskeleton for efficient T cell priming. While centriole numbers are generally limited to two in G1 phase and four prior to mitosis, DCs amplify centrioles upon antigen encounter leading to G1-arrested cells containing more than two centrioles[26]. We refer to this condition as multiple or extra centrioles. To address whether extra centrioles in DCs affect MTOC function in DCs, we visualized centrioles, PCM proteins and MT arrays in fixed samples. As extra centrioles in DCs arise due to overduplication of centrioles and by mitotic defects, which are accompanied by tetraploidization, we focused exclusively on diploid (2 N) cells. 82% of mature BMDCs showed a 2 N DNA profile of which

20% of cells contained more than two centrioles (Fig. 3a). Sorted 2 N cells were further immuno-stained against either pericentrin (PCNT), γ-tubulin or CDK5RAP2 to visualize PCM proteins surrounding the centrioles and α-tubulin to trace centrosomal MT filaments (Fig. 3b–d and Supplementay Fig. 3a, b). We found that the levels of all three PCM proteins were increased in cells with multiple centrioles indicating that extra centrioles are able to recruit PCM proteins (Fig. 3c, d; right panels and Supplementay Fig. 3a). To further address whether and how extra centrioles in DCs impact antigen-specific DC-T cell interactions, we next visualized PCM recruitment during IS formation focusing on γ-tubulin as crucial MT nucleating component. OVAp loaded CETN2-GFP expressing DCs were co-cultured with OT-II T cells and cell-cell conjugates were fixed after 2 h. Similar to unconjugated cells, we found enhanced γ-tubulin recruitment in DCs with multiple centrioles in the presence of T cells (Fig. 3e, f). High-resolution deconvolution microscopy further allowed us to identify MT filaments growing from the centrosome and quantify filament numbers within defined areas around the centrosome (Supplementary Fig. 2c). According to the role of γ-tubulin in promoting MT nucleation, we found increased numbers of MT filaments in the presence of extra centrioles during antigen-specific DC-T cell interactions, while the number of MTOCs was undistinguishable (Fig. 3g). These results demonstrate that centrosomal MT nucleation capacity in DCs is increased in the presence of additional centrioles, leading to one over-active MTOC with the capacity to nucleate a larger number of cytoplasmic MT filaments.

### DCs form multiple T cell contacts independently of centriole numbers

To address whether enhanced centrosomal MT nucleation in DCs with multiple centrioles correlates with a higher capacity to form multiple T cell contacts, we determined the frequency distribution of bound T cells in relation to DC centriole numbers. In the presence of excess T cells (1:5 DC/T cell ratio), DCs engaged with either one T cell (mono-conjugated) or several T cells simultaneously (multi-conjugated) (Supplementary Fig. 3c). Cell-cell contacts were also formed in the absence of OVAp as described before[34,35]. As CD4⁺ T helper synapses comprise at least four distinct stages that proceed over several hours and are associated with morphological shape changes as well as reorganization of cytoskeletal components in both cell types[20], we analysed conjugate formation after 1, 2 and 4 h after conjunction. One BMDC bound on average two to three T cells simultaneously in the presence of excess T cells (Supplementary Fig. 3c). We did not detect

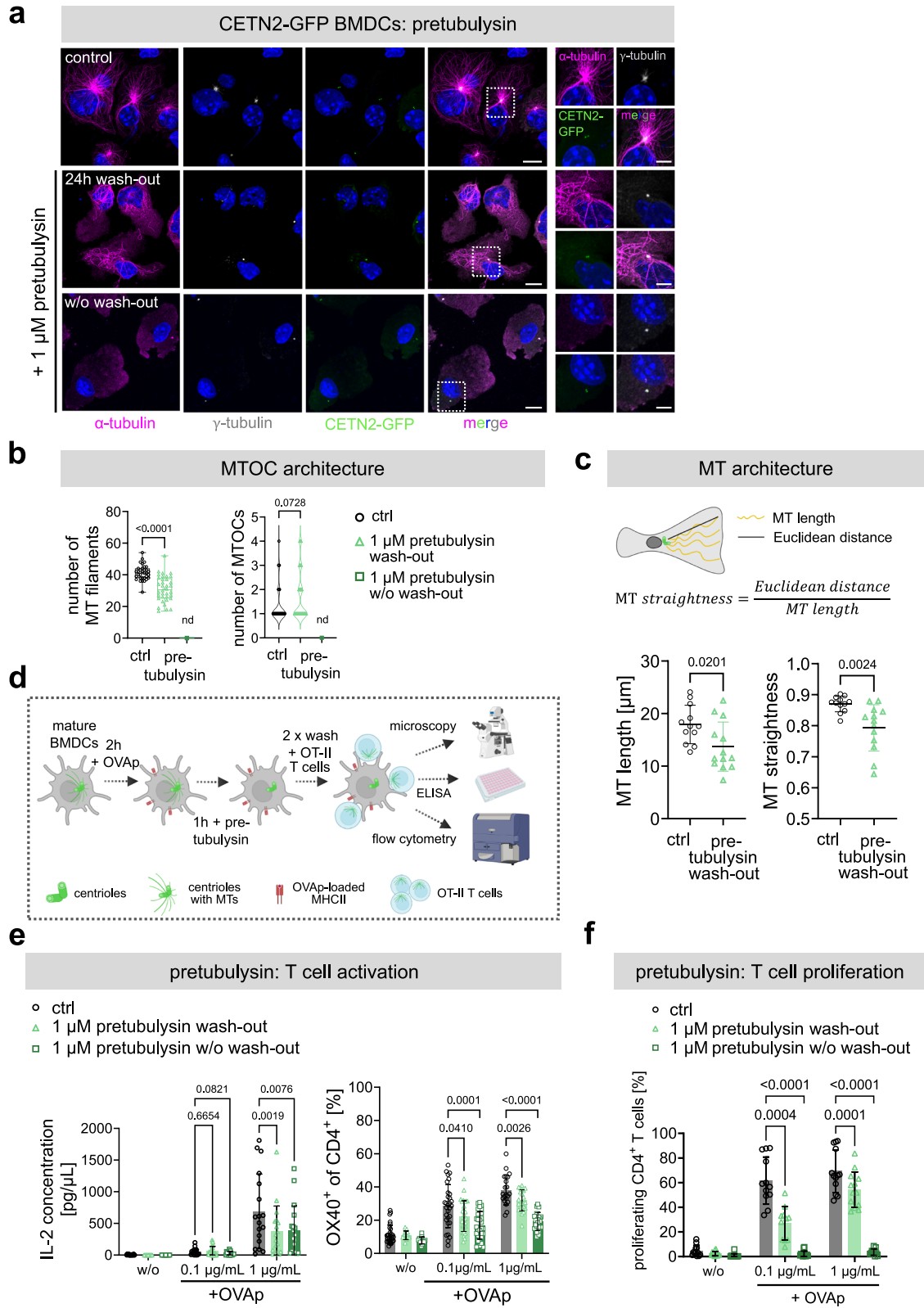

significant differences in the number of interacting T cells between DCs with two or multiple centrioles at distinct time points of conjugation (Supplementary Fig. 3d, e). Thus, we concluded that the capacity to form cell-cell contacts between DCs and T cells occurs independently of centriole numbers and the levels of centrosomal MT nucleation in DCs.

## Enhanced T cell activation in the presence of multiple centrioles in DCs

To further address whether the presence of additional centrioles and enhanced centrosomal MT nucleation impacts T cell stimulation, we sought to simultaneously visualize T cell activation and centriole numbers. To this end, we made use of Nur77[GFP] transgenic mice, which

**Fig. 2 | An intact MT cytoskeleton in APCs is required for efficient T cell activation. a** Immunostaining of CETN2-GFP expressing BMDCs after pretubulysin treatment. Maximum z-projections of merged and individual channels of CETN2-GFP (green), α-tubulin (magenta), γ-tubulin (white) and DAPI (blue) are shown. Scale bars, 10 μm. Right panels show indicated magnified regions. Scale bars, 5 μm. **b** Quantification of MT filaments (left) and MTOCs (right) in BMDCs treated with pretubulysin. Left graph shows median, interquartile range and minimum to maximum values of three independent experiments. $N = 33$ (ctrl) /34 (24 h pretubulysin wash-out). Right graph shows median and distribution of data points of three independent experiments. $N = 39$ (ctrl) /43 (24 h pretubulysin wash-out). In both graphs each data point represents one cell. $P$ values from two-tailed Mann-Whitney test. nd, not determined. **c** Quantification of MT length and straightness in CETN2-GFP-expressing BMDCs after 24 h of pretubulysin or DMSO (ctrl) wash-out. Both graphs show mean values ± s.d. of three independent experiments. Each data point represents one cell. $N = 13$ (ctrl) /12 (pretubulysin wash-out). $P$ values from two-tailed Mann-Whitney test. **d** Schematic representation of experimental workflow of OT-II transgenic T cell activation in the presence of pretubulysin-treated,

activated BMDCs and OVAp. **e** Quantification of OT-II T cell activation after co-culture with pretubulysin-treated BMDCs in the absence (wash-out) or presence (w/o wash-out) of pretubulysin during the co-culture. OT-II T cell activation was assessed by measuring IL-2 concentration in the supernatant (left) and OX40 upregulation on the T cell surface (right). Graphs display mean values ± s.d. Each data point represents one independent experiment. At least $N = 10.000$ cells analyzed per condition. Cells were derived from six (IL-2) or four (OX40) different mice. $P$ values from two-way Anova with Dunnett's multiple comparison. **f** Quantification of OT-II T cell proliferation measured by CFSE dilution after co-culture with pretubulysin-treated, OVAp-loaded BMDCs. T cell proliferation was assessed in the absence (wash-out) or presence (w/o wash-out) of pretubulysin. Graph displays mean values ± s.d. Each data point represents one independent experiment with $N = 10.000$ cells analyzed per condition. Cells were derived from three different mice. $P$ values from two-way Anova with Dunnett's multiple comparison. **c**, **d** Schematic pictures created with BioRender. Source data are provided as a Source Data file. ctrl: control, w/o: without.

express GFP under the control of the Nur77 promotor[36]. Nur77 gene expression is up-regulated by antigen-dependent T cell receptor stimulation but not by other homeostatic or inflammatory signals[37]. Accordingly, in this mouse model, the levels of GFP expression directly reflect T cell receptor signal strength and reach a maximum expression after 12-24 h of T cell receptor stimulation[36]. To estimate the timing of T cell activation after DC encounter in the context of MHCII antigen presentation, we crossed Nur77$^{GFP}$ mice with OT-II transgenic mice. Flow cytometric analysis revealed that Nur77-dependent GFP expression in T cells was low in the absence of antigen at all time points analyzed and after 1 h of co-culture with OVAp-loaded CETN2-GFP expressing DCs. By contrast, 2 h after mixing with OVAp-loaded DCs, a clear GFP signal appeared, which strongly increased after 4 h, 6 h, and 20 h of incubation (Fig. 4a and Supplementary Fig. 4a). At 20 h of co-culture in the presence of antigen, 67% of all T cells showed a prominent GFP signal, indicating efficient activation. In summary, Nur77$^{GFP}$ levels in T cells effectively reflect antigen-dependent T cell activation (Fig. 4a).

To elucidate whether T cell activation changes in the presence of different centriole numbers, we first sorted BMDCs according to CETN2-GFP signal intensities, which leads to two DC subpopulations enriched for either cells with two centrioles (CETN2-GFP$^{low}$) or cells with multiple centrioles (CETN2-GFP$^{high}$)[26] (Supplementary Fig. 4b, c). The percentage of cells carrying multiple centrioles ranged from 13–26 % in the CETN-GFP$^{low}$ population and 30–49 % within the CETN2-GFP$^{high}$ population, leading to an average enrichment of cells with multiple centrioles by a factor of 2.0 (Supplementary Fig. 4d). Sorted cells were further loaded with OVAp and incubated for either 6 h or 20 h with Nur77$^{GFP}$/OT-II transgenic T cells. After both time points of co-culture, the frequency of Nur77$^{GFP}$-positive cells was significantly higher when T cell priming was accomplished with DCs enriched for multiple centrioles, indicating that these cells activate a larger number of T cells within the same time period compared to cells with only two centrioles (Fig. 4b).

To directly observe T cell activation on a single cell level and in relation to centriole numbers, we immobilized DC-T cell conjugates at distinct time points of co-culture and determined fluorescence intensities of GFP levels in T cells. To distinguish between distinct numbers of centrioles, we used CETN2-GFP expressing DCs and counterstained cells with antibodies against γ-tubulin or CDK5RAP2. This approach allows simultaneous visualization of DC centriole numbers, T cell MTOC positioning as well as T cell activation via Nur77. We observed polarization of the T cell's centrosome to the nascent IS in an antigen-dependent manner at all time points analyzed, indicating efficient IS formation (Fig. 4c and Supplementary Fig. 5a). Similar to our flow cytometric analysis, GFP levels were significantly higher in the presence of antigen and steadily increased

from 2 h to 20 h of co-culture demonstrating T cell receptor activation and antigen-specific T cell priming (Fig. 4d and Supplementary Fig. 5b). Nur77-dependent GFP expression was elevated in T cells that had already reoriented its centrosome compared to cells with a centrally localized MTOC further demonstrating that centrosome reorientation in T cells correlates with Nur77-dependent GFP expression and T cell priming (Supplementary Fig. 5c). We next determined T cell activation in relation to the number of centrioles present in the conjugated DC. To this end, we defined a threshold for T cell activation based on the frequency distribution of GFP signal intensities in T cells co-cultured in the absence of antigen (Supplementary Fig. 5d, e). We found that as early as 2 h of co-culture, the number of GFP$^+$ T cells was elevated after priming with DCs that contain multiple centrioles and the same increase was observed after 4, 6, and 20 h of mixing (Fig. 4e). In accordance to our flow cytometric analysis, these results demonstrate that the frequency of activated T cells is higher when the APC contains multiple centrioles. Altogether, our findings provide evidence that cells with multiple centrioles activate T cells faster compared to cells with only two centrioles confirming optimized T cell responses in the presence of extra centrioles in APCs.

## Multiple centrioles cluster and localize close to the cell center in DCs during IS formation

Centrosome polarization has been reported in DCs during the interaction with naïve CD8$^+$ T cells. MTOC reorientation in DCs depends on the Rho GTPase Cdc42 and enables targeted delivery of vesicles containing T cell stimulatory molecules[38]. Based on these observations, we reasoned that the observed accelerated T cell priming capacity in the presence of multi-numerous centrioles may result from de-clustering of extra centrioles. Subsequently, centrioles could move and reorient to distinct contact sites forming additional MTOCs, which efficiently deliver stimulatory molecules to the respective IS (Fig. 5a). To test this hypothesis, we measured intracentrosomal distances (between centrioles of one centrosome) and average distances between all centrioles in DCs with multiple centrioles after conjugate formation in the presence or absence of OVAp (Fig. 5b, c, schemes). We further distinguished whether cells form mono- or multi-conjugated synapses. Analysis of intracentrosomal distances in cells with two centrioles treated with or without OVAp for distinct time points of co-culture (1, 2, 4, 6 and 20 h) revealed distances of 0.8-1.1 μm between individual centrioles independently of the number of T cells attached (Fig. 5b and Supplementary Fig. 6a). Similarly, average distances in cells with multiple centrioles ranged between 0.9–1.3 μm and did not show prominent differences between OVAp loaded and unloaded cells (Fig. 5c and Supplementary Fig. 6b), suggesting that multiple centrioles remain clustered during antigen-specific DC-T cell contacts.

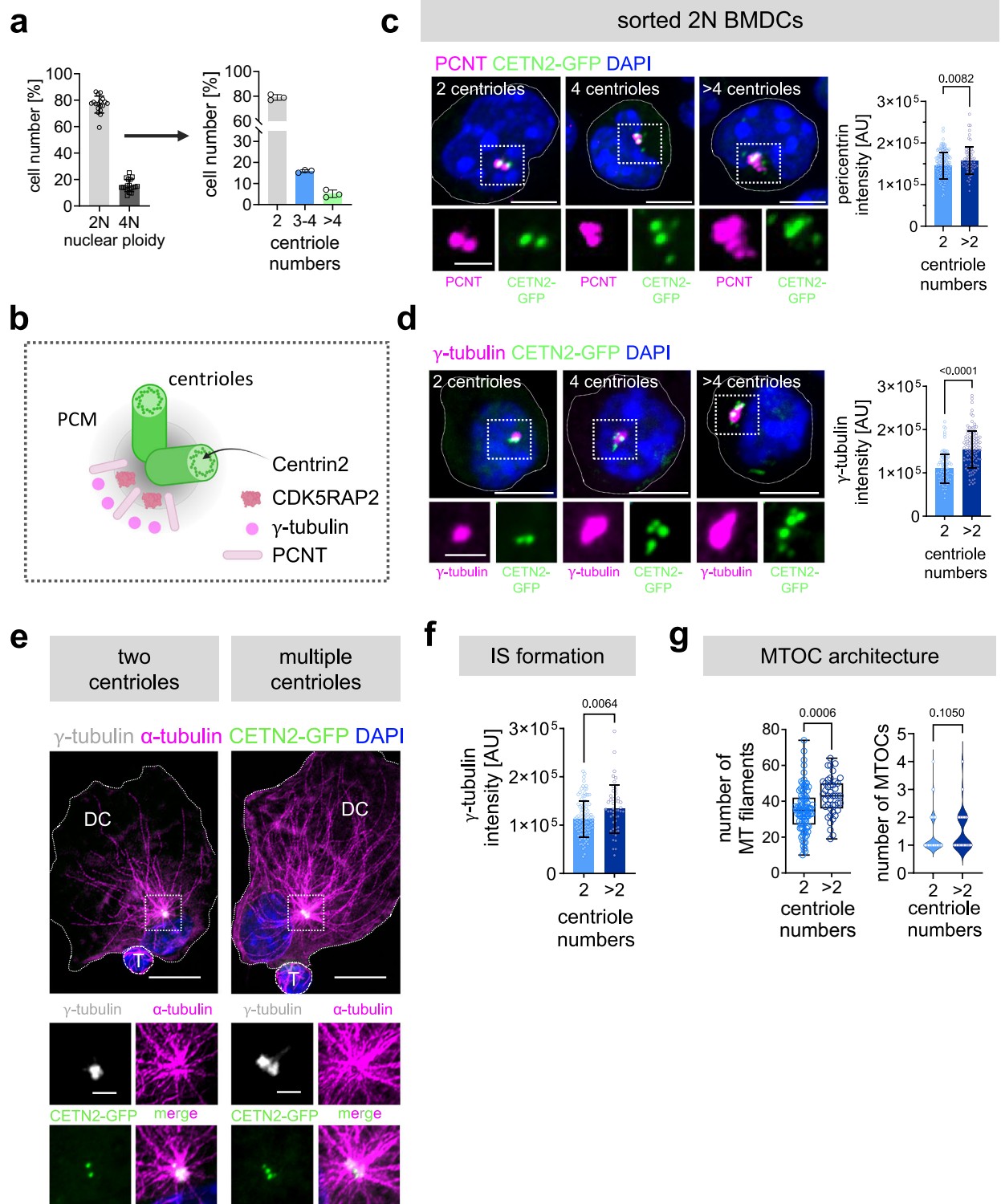

Average distances were indistinguishable not only when cells formed single contacts but also in multi-conjugated cells.

To further analyze centriole positioning in DCs, we quantified the ratio of distances between the DC centrioles and the T cell, which was normalized to the distance between the DC center point (CP) and the T cell in the presence or absence of antigen (Fig. 5d). Our analysis revealed that 1, 2, 4, 6 and 20 h after co-culture, all centrioles in DCs were still located in close proximity to the cell center and the nucleus (Fig. 5e, f and Supplementary Fig. 6c–f). This behavior was not only observed in multi-conjugated cells but also when single synapses were

formed. These results suggest that MTOC positioning in DCs was independent of the number of centrioles and T cells conjugated. Moreover, our findings demonstrate that centrosome polarization toward the IS in DCs is dispensable for T cell activation.

To further study DC centriole dynamics with high spatio-temporal resolution, in particular during early DC-T cell encounters, we recorded time-lapse images of DC-T cell conjugates. To this end, CETN2-GFP expressing DCs were loaded with OVAp, mixed with OVA-specific OT-II T cells and imaged directly after mixing. To monitor efficient DC-T cell interaction, we visualized intracellular calcium ($Ca^{2+}$) influx in T cells

**Fig. 3 | IS formation in the presence of multiple centrioles. a** Left: quantification of 2 N and 4 N cells according to DNA content (*see also* Suppl Fig. 1a). Right: quantification of centriole numbers in sorted mature 2 N CETN2-GFP BMDCs according to CETN2-GFP/PCNT⁺ foci. Graph shows mean values ± s.d. of three independent experiments. *N* = 269/258/242 cells analyzed. **b** Scheme of centrioles surrounded by PCM proteins (PCNT, γ-tubulin and CDK5RAP2) created with BioRender. **c, d** Immunostaining of PCNT and γ-tubulin in sorted 2 N mature CETN2-GFP BMDCs. Merged channels of PCNT/γ-tubulin (magenta), CETN2-GFP (green) and DAPI (blue) are shown. Scale bars, 5 μm. Insets show magnification of indicated regions. Individual channels of CETN2-GFP (green) and PCNT/γ-tubulin (magenta) are shown. Scale bars, 2 μm. Right: Quantification of PCNT/γ-tubulin signal intensity at the centrioles in DCs. Graphs show mean values ± s.d. of one representative experiment out of three independent experiments. Each data point represents one cell. PCNT: *N* = 226 (2 centrioles) /71 (>2 centrioles). γ-tubulin: *N* = 106 (2 centrioles) /117 (>2 centrioles). *P* values from two-tailed Mann-Whitney test. **e** Immunostaining of MT filaments in sorted 2 N mature CETN2-GFP expressing

BMDCs loaded with OVAp and forming conjugates with T cells. White dotted box indicates magnified region below. Merged and individual channels of CETN2-GFP (green), γ-tubulin (white), α-tubulin (magenta) and DAPI (blue) are shown. Scale bars, 10 μm (upper panels) and 2 μm (insets bottom). **f** Quantification of γ-tubulin signal intensities at the centrioles in DCs forming conjugates with T cells. Graph shows mean value ± s.d. of one out of three independent experiments. Each data point represents one cell. *N* = 108 (2 centrioles) /49 (>2 centrioles). *P* value from two-tailed Mann-Whitney test. **g** Quantification of MT filaments (left) and MTOCs (right) in sorted 2 N CETN2-GFP BMDCs loaded with OVAp and forming contacts with T cells. Left: Graph displays median, interquartile range and minimum to maximum values. Each data point represents one cell derived from 3 independent experiments. *N* = 93 (2 centrioles)/43 (>2 centrioles). Right: Graph shows median and distribution of data points of three independent experiments. *N* = 105 (2 centrioles) /25 (>2 centrioles). *P* values from two-tailed Mann-Whitney test. **c–e** All images represent maximum z-projections. Source data are provided as a Source Data file.

and considered only interactions with long-lasting Ca²⁺ responses. In cells containing two centrioles, intracentrosomal distances did not increase upon contacts with one or several T cells (Supplementary Fig. 7a, Supplementary movies 1, 2). Similarly, average distances in cells with multiple centrioles remained unaltered in conjunction with one or several T cells (Supplementary Fig. 7b, Supplementary movies 3, 4). Moreover, no net movement of centrioles in DCs could be observed within the imaging period (0-1 h after conjugation) in the presence or absence of antigen (Supplementary Fig. 7c, and Supplementary movies 1–4). Instead, centrioles were found near the nucleus independently of the number of T cells bound, similar to our fixed samples (Supplementary Fig. 7d).

As cellular behavior can vary significantly depending on whether cells are cultured on flat 2D surfaces or under confined 3D conditions[39], we further sought to study centriole configuration in physiologically relevant environments and carried out adoptive transfer experiments[40]. Naive OVAp-specific OT-II-specific T cells labelled with CellTrace Far Red Cell Proliferation dye were seeded into wildtype recipient mice and, after 24 h immunized by injection of OVAp-loaded CETN2-GFP expressing BMDCs (Fig. 6a). Mice were sacrificed 48 h after BMDC injection to isolate popliteal and inguinal lymph nodes (LNs). Immunostaining of frozen LN sections revealed antigen-specific DC-T cell interactions within T cell areas (Fig. 6b). 3D rendering of DC-T cell contacts within LNs identified intracentrosomal distances of 0.9 ± 0.2 μm (1 T cell) or 0.8 ± 0.2 μm (>1 T cell) and average distances between centrioles of 1.1 ± 0.2 μm (1 T cell) or 1.2 ± 0.3 μm (>1 T cell) (Fig. 6c, d, and Supplementary movie 5) highlighting a clustered centriole configuration in vivo. Similar to our 2D analysis, all centrioles localized closely to the cell center during DC-T cell contacts independent of the number of T cells attached (Fig. 6e). In summary, our results demonstrate that multiple centrioles cluster during antigen-specific DC-T cell contacts in vitro and in vivo and collectively stay close to the cell center when mono- or multi-conjugated contacts are formed.

### Centriole de-clustering impairs T cell activation

As we observed tight centriole clustering during DC-T cell contacts, we next addressed the importance of this phenomenon for the induction of T cell priming. Therefore, DCs were treated with the de-clustering agent PJ-34, a poly-ADP-ribose polymerase inhibitor with no MT binding activity in proliferating cells[41,42]. PJ-34 was administered for 3 h on mature DCs to test the effect on centriole organization and to rule out effects on transcriptional regulation which might impact T cell activation. MHCII cell surface expression was unaltered after PJ-34 treatment (Supplementary Fig. 8a, b). In addition, mRNA expression levels of DC-specific cytokines such as *Ccl5*, *Cxcl1* and *Il-6* were analysed as they are involved in attracting and activating immune cells[43–45]. mRNA levels of *Ccl5*, *Cxcl1* and *Il-6* were unaffected by PJ-34 treatment (Supplementary Fig. 8c). Also, intracellular protein levels were undistinguishable in PJ-34-treated

compared to control cells (Supplementary Fig. 8d). Thus, we concluded that PJ-34 treatment does not perturb antigen presentation or cytokine production in DCs. Next, centriolar distances were measured in fixed samples. Under control conditions (H₂O), cells containing two centrioles displayed mean intracentrosomal distances of 0.7 ± 0.3 μm, while intercentrosomal distances in cells with multiple centrioles were slightly larger, with average distances of 1.3 ± 0.7 μm (Fig. 7a, b). PJ-34 treatment led to a significant shift to larger distances, in particular of intercentrosomal distances in cells carrying multiple centrioles (2.2 ± 1.6 μm), indicating that pairs of centrioles get dispersed (Fig. 7b, and Supplementary Fig. 8e). Immunostaining against γ- and α-tubulin and analysis of MTOC numbers and MT filaments emanating from either clustered or dispersed centrioles revealed that PJ-34-treated cells harbor significantly more MTOCs per cell compared to control cells while the number of MT filaments was unaltered upon PJ-34 treatment (Fig. 7c). Intriguingly, when co-culturing either PJ-34-treated or control OVAp-loaded DCs with OT-II T cells, we found that in the presence of de-clustered centrioles, T cell activation was significantly diminished compared to control cells (Fig. 7d, e), demonstrating that de-clustering of multi-numerous centrioles and formation of multiple MTOCs impairs T cell activation. In summary, our findings highlight a crucial role for proper centriole and MTOC arrangement in APCs to efficiently activate T cells.

### Modeling T cell priming with geometrically optimal centriole positioning in DCs

Since centrioles are integral to the centrosome, their location directly corresponds to that of the centrosome. Our experimental data emphasize a crucial role of centriole positioning close to the cell center in DCs and provide evidence that T cell activation is increased in the presence of multiple clustered centrioles compared to DCs containing two centrioles. These observations prompt queries into whether the centrosome's proximity to the cell center is advantageous for DCs. For instance, a centrosomal position geometrically optimal to the entire cell surface may expedite the delivery of immune signals to the interacting T cells. The optimal position will also provide dynamic MTs faster access to the IS. Therefore, a key task is to examine the optimal centrosome localization and how MTs contribute to centrosome positioning. Further, we address the physical consequences of centriole clustering. In other words, does clustering of centrioles facilitate efficient MT search for the IS over dispersed centrioles acting as separate MTOCs in distinct locations? And whether the optimal position also corresponds to the mechanically stable position of the centriole cluster. To further elaborate on this, we resort to mathematical and computational models to systematically approach these questions.

First, we rationalized with simple geometric considerations the physiological consequences of centralized centrioles during T cell priming. We first seek to explore the geometrically optimal position of the centriole cluster or a single centrosome with two centrioles in DCs

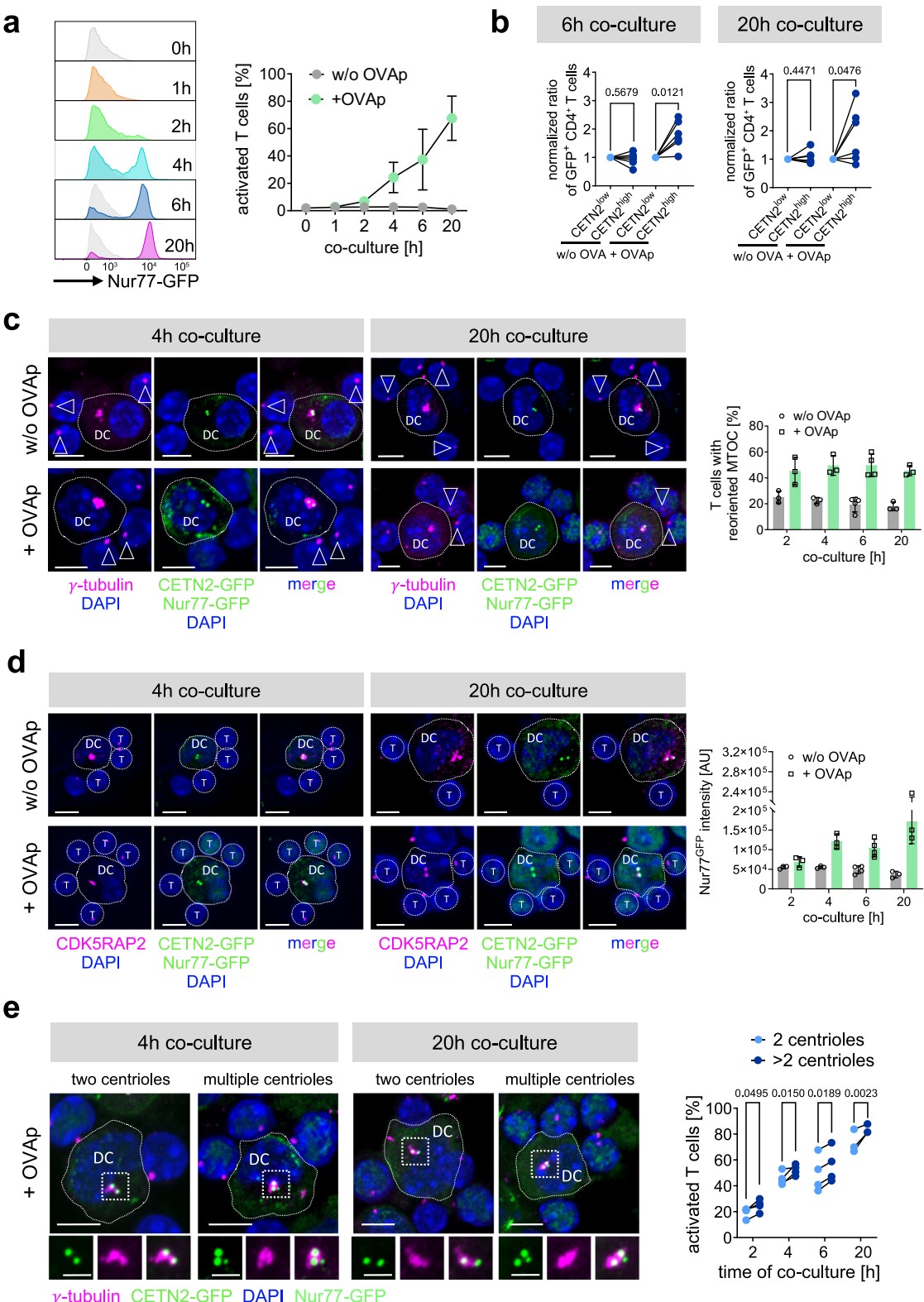

relative to the entire cell surface, without explicitly delving into the complexities of MT dynamics. Since the position of the target - the center of the IS on the cell surface - is not predetermined, a centrosomal position with minimal average distance to all possible target points on the cell surface may facilitate efficient delivery of stimulatory molecules to the IS. For cell surface regions directly accessible to the centrosome, the shortest distance is a straight line from the centrosome to the target point on the cell surface. However, for regions no longer visible due to nuclear hindrance, the minimum geometric distance would follow the scheme presented in Fig. 8a. This distance corresponds to the path length of MTs on optimal trajectories between the centrosome and the target point on the cell surface. For a detailed calculation, we refer to the "Methods" section.

In the absence of a nucleus, the optimal centrosome position coincides with the geometric center of the cell, as determined through analytical calculations in both circular and spherical cellular

**Fig. 4 | Enhanced T cell activation in the presence of multiple centrioles.**
**a** Histogram (left) and quantification (right) of Nur77$^{GFP}$ expression after different time points of DC-T cell co-culture in the presence (+ OVAp) or absence (w/o OVAp) of OVAp. Graph represents mean values ± s.d. of 4 independent experiments. $N = 10.000$ cells analyzed per condition. **b** Ratio of GFP$^+$/CD4$^+$ T cells after 6 and 20 h of co-culture with OVAp-pulsed 2 N BMDC subpopulations. Each data point represents one out of six independent experiment with $N = 10.000$ cells analyzed per condition and with pairing between sorted 2 N CETN2-GFP$^{low}$ (light blue) and CETN2-GFP$^{high}$ (dark blue) expressing cells. Data was normalized to CETN2-GFP$^{low}$ condition. *P* values from two-tailed, paired Student's *t*-test. **c** Left: Immunostaining of γ-tubulin (magenta) in DC-T cell conjugates. Merged and individual channels of γ-tubulin (magenta), CETN2-GFP (green), Nur77$^{GFP}$ (green) and DAPI (blue) are shown. DC outline is indicated with dotted line. White arrowheads point to the T cell's centrosome. Scale bars, 5 $\mu$m Right: quantification of MTOC polarization towards the IS in T cells after different timepoints of co-culture. Frequency was calculated across all T cells analysed for the indicated condition. Graph displays mean values ± s.d. Each data point represents one independent experiment with $N = 346/136/156 / 129/252/178$ (2 h w/o / + OVAp); $266/213/188 / 341/223/464$ (4 h w/o / + OVAp); $190/487/63/157 / 278/806/151/204$ (6 h w/o / + OVAp); $25/55/137 / 70/248/314$ (20 h w/o / + OVAp) cells analyzed per condition. **d** Visualization and quantification of Nur77$^{GFP}$ expression in DC-T cell conjugates after different timepoints of co-culture. Left: immunostaining of CDK5RAP2 (magenta) in CETN2-GFP (green)

BMDCs after co-culture with Nur77$^{GFP}$/OT-II CD4$^+$ T cells. Nuclei were counterstained with DAPI (blue). Dotted lines indicate DC outline; round circles the areas of GFP measurements. Scale bars, 5 $\mu$m. Right: quantification of GFP signal intensity in T cells in the presence and absence of antigen. Graph displays mean values ± s.d. Each data point represents one independent experiment with $N = 333/142/156 / 111/232/190$ (2 h w/o / + OVAp); $270/208/184 / 345/213/488$ (4 h w/o / + OVAp); $149/483/180/268 / 266/789/309/239$ (6 h w/o / + OVAp); $67/204/140 / 244/308/319$ (20 h w/o / + OVAp) cells analyzed per experiment. **e** Analysis of T cell activation in dependence of centriole numbers. Left: immunostaining of γ-tubulin in DC-T cell conjugates. Merged and individual channels of CETN2-GFP (green), Nur77$^{GFP}$ (green), γ-tubulin (magenta) and DAPI (blue) are shown. Indicated regions are magnified and shown below. Scale bars, 5 $\mu$m (top). Scale bars, 2 $\mu$m (magnified region). Right: quantification of T cell activation according to Nur77$^{GFP}$ signal intensities in dependence of DC centriole numbers for different time points of DC-T cell co-culture. Each data point represents one independent experiment with pairing between cells with two (light blue) and multiple (dark blue) centrioles from one experiment. $N = 129/230/325/250$ (2 h); $327/204/184/325$ (4 h); $265/789/239/309$ (6 h); $308/244/317$ (20 h) cells analyzed per condition. *P* values from two-way Anova with Šidák's multiple comparisons. **c**–**e** All images represent maximum z-projections. Source data are provided as a Source Data file. ctrl: control, w/o: without.

---

geometries and further validated by computational analysis (see *Supplementary Information;* Supplementary Fig. 9a). However, the presence of a nucleus covering the central region of the cell prevents the centrosome from locating at the cell center. Using computational analysis, we first explore the optimal centrosome position in the presence of a centrally located nucleus and estimate the average geometric distance, $<d_{short}>$, as a function of the centrosome's distance from the cell center, $h_{CS}$, or from the nuclear surface $d_{CS-NS}$. Our data indicate an optimal perinuclear positioning of the centrosome, marginally shifted from the nuclear surface by <1$\mu m$ (Fig. 8b). We further investigate the centrosome's position for various off-centered positions of the nucleus. Note that, for a centrally located nucleus, spherical symmetry of the cell and the nucleus ensures that estimating $<d_{short}>$ with random target points on the cell surface is independent of the direction of the centrosome placed away from the nucleus. It solely depends on the centrosome's distance from the nucleus. However, for an off-centered nucleus, $<d_{short}>$ depends on the specific three-dimensional positioning of the centrosome relative to the nucleus. We discovered that the global minimum of $<d_{short}>$ is achieved when the centrosome lies on a fixed axis pointed from the nuclear center toward the cell center (shown as Z-axis in the plots; Supplementary Fig. 9b–e). Therefore, by placing the centrosome at various locations along that axis, we estimate the optimized values of $h_{CS}$ and $d_{CS-NS}$, corresponding to the minimum of $<d_{short}>$, and plotted for three different nuclear positions: $d_{offset} = 0\mu m$, representing a centrally located nucleus; $d_{offset} = 6\mu m$, where the cell center is just outside the nuclear surface and $d_{offset} = 12\mu m$, indicating the maximally off-centered nucleus touching the cell surface (Fig. 8c). Our findings suggest that positioning the centrosome close to the nuclear periphery is advantageous when the nucleus is at the cell center. The optimal centrosome position shifts toward the cell center as the nucleus becomes off-centered. Overall, our model provides a rational accounting for the geometric positioning of centrosomes or the clustered centrioles near the cell center leading to efficient priming of T cells on the cell surface.

## Modeling highlights a critical role of MT dynamics for centrosome (or centriole) positioning in DCs

Next, we sought to understand how MT dynamics may affect centrosome positioning in DCs and aimed to determine the optimal centrosome position that minimizes the time required for searching the IS in DCs. To explore this, we adopted the well-established "search and

capture" hypothesis, initially proposed in the context of mitotic spindle assembly, which leads to the binding of MT filaments to the kinetochores of sister chromatids[46–48]. This framework was further extended in a recent study to examine the docking efficiency of MTs at the IS in T cells[49] and involves dynamically unstable MTs originating from the centrosome, exploring the three-dimensional cellular volume to locate and capture the target IS. We accommodate this established model for MT dynamics in T cells with morphological differences of the MT network of T cells and DCs: T cells are spherical (radius, $r \approx 5\mu m$), much smaller than DCs, with a relatively large nucleus ($r \approx 3\mu m$), such that the centrosome is generally located close to cell surface and MTs emanating tangentially to the surface[50,51]. DCs are large cells ($r \approx 18\mu m$) with a relatively smaller nucleus ($r \approx 6\mu m$) and MTs typically nucleated from a clustered centriole assembly positioned close to the cell center. This specific arrangement allows MTs to radiate astrally towards the surface, rarely touching it tangentially as in T cells (Fig. 8d). Consequently, we incorporate a reduced MT gliding probability into our model: When growing MTs hit the cell surface, they can undergo instant catastrophe determined by the angle of contact with the surface or glide along the surface with catastrophe frequency increasing with gliding distance, as depicted in Fig. 8e (ii), (iii). Note that such cortex-induced changes in MT growth direction were reported in animal cells[52,53], plant cells[54], yeast[55], and HeLa cells[56] and are well established in theoretical modeling[49,57]. We further hypothesized that MT growth can be similarly directed by the nuclear surface since we frequently observe MTs in the "geometric shadow" behind the nucleus (the region not accessible to straight MTs growing from the centrosome; see also Fig. 3e) and incorporated it into our model as sketched in Fig. 8e (i): When MTs reach the nuclear surface, they first glide along it and subsequently detach at a rate that increases with MT curvature along the nuclear surface. After detachment, MTs grow tangentially from the nuclear surface toward the cell surface seeking the target.

To explore how the MT's gliding ability along the cell surface affects centrosome positioning in DCs, we plot the average search time ($<T_{search}>$) against the distance of the centrosome from the surface of a centrally located nucleus, $d_{CS-NS}$, while varying the parameter $\lambda_l$ (Fig. 8f) regulating the sensitivity of the catastrophe frequency with MT's gliding distance (see Fig. 8e, iii). A higher $\lambda_l$ corresponds to a faster MT catastrophe and vice versa. We find that for higher $\lambda_l$, the average search time is minimized when the centrosome is located near the nuclear periphery. This observation correlates with the scenario

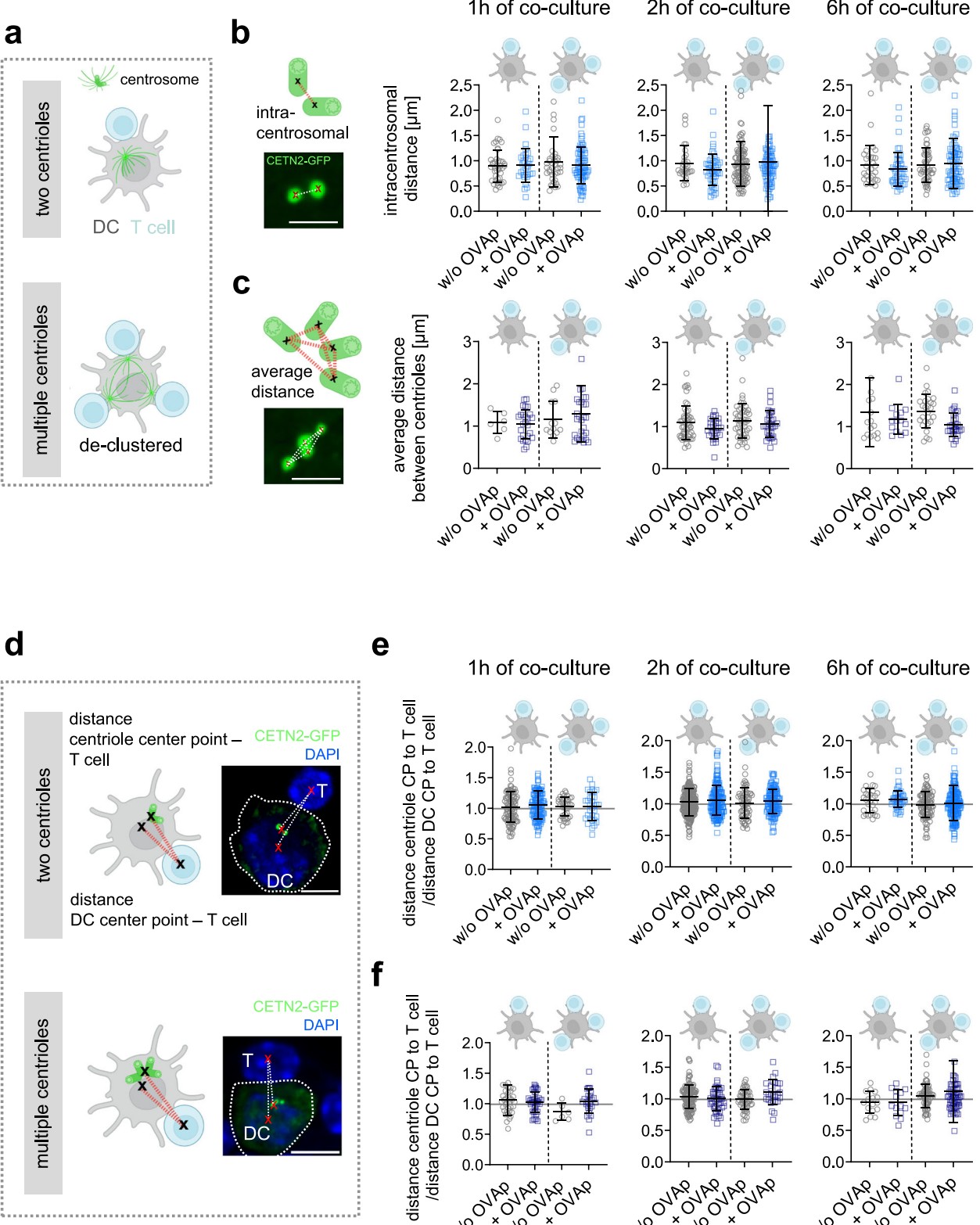

where MTs are not allowed to glide along the cell surface, resulting in instant catastrophe upon hitting the cell surface. However, for very small $\lambda_l$ values, the optimal centrosome position shifts adjacent to the cell surface. With restricted MT gliding along the cell surface, the search progresses via two pathways: MTs can directly reach the target from the centrosome, and MTs that encounter the nucleus glide along its surface and reach the target on the cell surface after detaching from the nuclear surface. In this case, MTs can capture the target most

efficiently when they are close to the nuclear surface, minimizing the average distance between the centrosome and the target IS. For a centrosome located far off from the nucleus, capturing IS located on the cell surface well below the cell's equatorial plane (hidden below the nucleus) relies on the MTs originating from the centrosome hitting the nuclear surface and gliding along it before reaching the target IS. This in turn increases the overall search time for centrosomes distant from the nucleus. Conversely, if MTs can glide along the cell surface (small

**Fig. 5 | Centriole configuration during antigen-specific DC-T cell contacts.**
**a** Sketch illustrating hypothesis about centriole de-clustering and centriole polarization in DCs toward the IS in the presence of two (top) and multiple centrioles (bottom). **b, c** Sketches and pictures indicate centriole configuration (CETN2-GFP) during DC-T cell contacts. Scale bars, 2 $\mu$m. Graphs show quantification of intracentrosomal distances (**b**) and average distances (**c**) in DCs with two or multiple centrioles at different time points of co-culture and bound to one or several T cells (separated by dashed line and indicated on top). Graphs display mean values ± s.d. Each data point represents one cell derived from $n = 5/5/3$ (1 h/2 h/6 h) independent experiments. **d** Sketches and confocal images of CETN2-GFP BMDCs (green) and

T cells indicating distances between centriole center point (CP) and T cell CP, and DC CP and T cell CP in cells with two (upper panel) and multiple (lower panel) centrioles. Nuclei were counterstained with DAPI (blue). Scale bars, 5 $\mu$m.
**e, f** Quantification of ratio between distance from centriole CP to T cell CP and distance DC CP to T cell in cells with two (**e**) and multiple (**f**) centrioles. Graphs show mean values ± s.d. for DCs attached to one T cell (left) and multiple T cells (right) separated by dashed line. Each data point represents one cell derived from $n = 5/5/3$ (1 h/2 h/6 h) independent experiments. **a**–**f** Pictures created with BioRender. Source data are provided as a Source Data file. w/o: without.

$\lambda_l$), placing the centrosome near the cell surface accelerates the target capture. Note that, given the experimentally consistent optimal centrosome positioning without MT gliding along the cell surface (or with larger $\lambda_l$), henceforth, we proceeded with simulations, preventing the MTs from gliding along the cell surface.

Next, we explored the sensitivity of our findings to off-centered nuclear positioning (Fig. 8g). In line with geometric predictions, our results indicate the optimal perinuclear position of the centrosome for a centrally located nucleus and in the vicinity of the cell center for an off-centered nucleus (Fig. 8g, h). Surprisingly, even with severely shorter MTs (~ 30% of the average MT length considered in the model), the optimal centrosome position remains consistent, with prolonged search times (Supplementary Fig. 9f–h). Evidently, for a largely off-centered nucleus ($d_{offset} = 12\mu$m), a perinuclear centrosome results in a significant increase in the distance to targets on the cell surface away from the nucleus, leading to extended search times. In contrast, MT arrays from a centrally located centrosome efficiently capture targets across the cell surface due to the minimum average distance to the centrosome. Interestingly, like spherical cells, flattened DCs display optimal centrosome positioning at the cell center for an off-centered nucleus and in the perinuclear region along the short axis of the cell for a centrally located nucleus (Supplementary Fig. 10a–d).

**Clustering of multiple centrioles enhances MT search efficiency**
Our experiments show that PJ-34-induced de-clustering of centrioles, resulting in distant MTOCs within the cell, reduces DCs' T cell activation efficiency (Fig. 7). Therefore, to further investigate how clustering of multiple centrioles through centrosomal aggregation affects the search process, we performed simulations involving four independent clusters of centrioles (i.e., four centrosomes) acting as independent MTOCs, with an evenly distributed array of MTs (Fig. 9a–c). These centrosomes were separated and randomly positioned around the optimal centrosomal position determined earlier for a centrally located and off-centered nucleus (see Fig. 8g, h). Strikingly, our findings revealed that irrespective of the nuclear positioning, the average search time is minimized when all the centrosomes are close together, forming clustered centrioles - a trend that holds even in largely flattened cells (Supplementary Fig. 10e–g). Note that, to ensure unrestricted MT growth along nuclear periphery, we flattened the cell up to the point such that the nuclear envelope does not contact the cell boundary. Clustered centrioles promote efficient capture of the IS by a unified radial array of MTs. The collective search conducted by MTs nucleating from all the centrosomes originating from a single optimal position increases the likelihood that at least one MT captures the target. However, in the case of dispersed centrosomes, the capture process is not as efficient since the MTs do not originate from the same optimal position. Fewer MTs from each of the centrosomes searching independently for distant targets on the cell surface decreases the chance of a successful capture which effectively increases the overall search time.

Noting that our experiments revealed a ~ 10 − 20% increase in the total number of MTs in DCs containing multiple clustered centrioles compared to DCs with only two centrioles, we further employ our computational model to investigate the correlation between MT

search time and the number of MTs (see Fig. 3g, left). Interestingly, we find a decrease in the search time with an increasing number of MTs (Fig. 9d). This behavior is consistent with our analytical prediction of average search time without a nucleus, estimated within the model framework introduced earlier[49] (see *Supplementary Information*). Note that, dispersing the centrosomes may also facilitate faster target capture when MTs are allowed to glide freely along the cell surface rather than restricting their growth (Supplementary Fig. 11a–c). Under such conditions, the dispersed centrosomes allow MTs to efficiently capture the targets by gliding along the cell surface, resulting in a reduced search time. Overall, our model highlights the pivotal role of multiple clustered centrioles, coupled with increased MT numbers, efficiently activating T cells by facilitating quicker access to the IS through dynamic MTs.

**A plausible mechanistic force-balance model describing centriole cluster positioning and centrosome integrity in DCs**
To find the mechanically stable configuration of multiple centrioles, irrespective of a putative attractive force between them, we used an established mechanistic model of MT-generated forces resulting from interactions of centrosomal MTs with the nucleus and the cell surface[58–60]. We considered initially four closely spaced centrosomes (each containg centriole doublets) with independent microtubule nucleation activity. While pushing against the cell or nuclear surfaces, a polymerizing MT is buckled, and the corresponding buckling force is applied on the centrosomes. The resultant force causes the centrosomes to move through the cytoplasm[59–61] as sketched in Fig. 9e. Similar to the search and capture model described earlier with frequent observations of MTs at the "geometrical shadow" behind the nucleus (see also Fig. 3e), we hypothesize that MT growth can be directed by the nuclear surface. Accordingly, our model assumes that the majority of MTs slide along the nuclear surface and only a small fraction (~ 10%) buckles at the nuclear surface, whereas all MTs hitting the cell surface generate buckling-induced-pushing force on the centrosomes.

We observe that all centrosomes stayed together and positioned along the line joining the nuclear center and the cell center. The centrosomes remain close to the nuclear surface ($\sim$ 1$\mu$m away from the nuclear surface) when the nucleus is fixed at the cell center ($d_{offset} = 0$) and near the cell center when the nucleus is off-centered ($d_{offset} = 6\mu$m and 12$\mu$m) (Fig. 9f, Supplementary movies 6-8). These findings qualitatively corroborate our experimental observations and simulation results of the optimal centrosome (or centrioles) positions estimated geometrically and by the search-and-capture scheme. As expected, when more MTs are assumed to buckle at the nuclear surface, the distance between the centroid of the four closely spaced centrosomes (or centriole cluster) and the nuclear surface increases due to the higher pushing forces generated on the centrosomes from the nuclear surface (Supplementary Fig. 11d).

Altogether, our computational approach highlights a beneficial role for multiple clustered centrioles in APCs, which nucleate a larger number of MT filaments and collectively stay close to the cell center to optimally form cell-cell contacts and ultimately activate T cells.

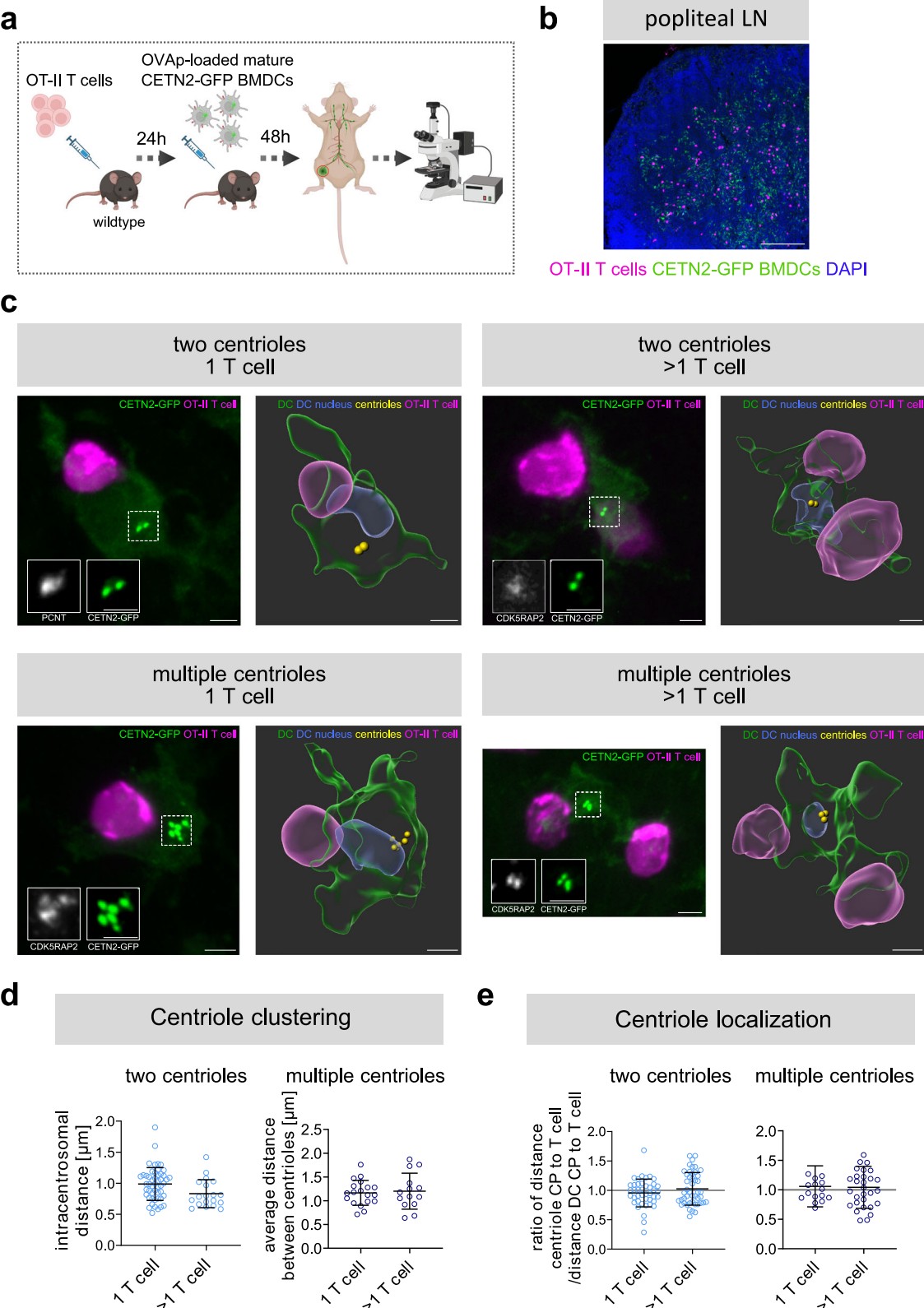

## Discussion

The initiation of an effective immune response depends on precise cell-cell communication, particularly between antigen-presenting cells and T cells. During the past decades much progress has been made in elucidating the dynamic behavior of cell surface and intracellular signaling molecules associated with IS formation in T cells, while much less is known about the dynamic changes within APCs during T cell encounter. Using single-cell analysis in combination with a computational approach, we show here that DC centrosomal integrity and microtubule organization - rather than centrosome polarization - are crucial for CD4$^+$ T cell activation, with excess centrioles enhancing immunogenicity by forming a centralized, hyperactive MTOC.

In line with our findings on centrosome integrity in DCs, other studies have suggested that a functional centrosome in T cells is

**Fig. 6 | Centriole organization in murine lymph nodes. a** Scheme of OT-II T cell and CETN2-GFP expressing BMDC injection created with BioRender. **b** Maximum z-projection of popliteal LN containing OT-II T cells (magenta) and CETN2-GFP BMDCs (green) counterstained with DAPI (blue). Scale bar, 200 μm. **c** Left (in each panel): Maximum z-projection of merged channels of CETN2-GFP (green) and pre-stained OT-II T cells (magenta). Scale bars, 3 μm. Insets show individual channels of PCNT or CDK5RAP2 (magenta) and CETN2-GFP (green). Scale bars, 2 μm. Right (in each panel): 3D rendering of DC (green), DC nucleus (blue), DC centrioles (yellow) and OT-II T cell (magenta). For better visibility of DC-T cell contacs, 3D rendered images were rotated. Scale bars, 3 μm. **d** Quantification of intracentrosomal

distances in cells with two centrioles (left) and average distances between centrioles in cells with multiple centrioles (right) during contact with one or multiple T cells. Graphs show mean values ± s.d. Each data point represents one cell derived from 5 independent experiments. $N = 50/19$ (two centrioles) and 19/14 (multiple centrioles). **e** Quantification of distance centriole CP to T cell CP normalized to distance DC CP to T cell CP in cells with two centrioles (left) and multiple centrioles (right) during contact with one or multiple T cells. Graphs show mean values ± s.d. Each data point represents one cell derived from 5 independent experiments. $N = 42/46$ (two centrioles) and 16/30 (multiple centrioles). Source data are provided as a Source Data file. CP: center point.

required for efficient cytotoxic T cell-mediated killing[29]. However, it is controversially discussed whether T cell centrosome polarization is dispensable or required for trafficking and directional secretion of lytic granules towards the cytotoxic synapse[29,62,63]. Intriguingly, our study and the studies of others imply, that T cell activation can occur in the absence of DC MTOC polarization to the synapse[35,64]. One possible explanation for the distinct intracellular MTOC reorganization observed during cell–cell contact formation in immune cells could be differences in cell size. Smaller cells such as T- and B cells need to reorient their centrosome due to geometric hindrance of the nucleus in order to efficiently deliver cargos via MT filaments to the contact zone. By contrast, larger cells such as DCs exhibit a centrally localized centrosome, which forms an astral MT array that reaches out to the cell membrane without being strongly disturbed by the presence of a comparably small nucleus. Moreover, DCs form multi-conjugated synapses which make a centrally localized centrosome advantageous for minimizing the average MT search time to reach every point on the cell membrane that are potential sites to form additional cell-cell contacts. Consequently, centrosome re-orientation to one particular site within the cell would increase MT search time to form contacts at opposite sites. In this context, the connection between MT dynamics and centrosome positioning in DCs was further established by our computational model: promoting gliding of MTs along the nuclear surface while restricting it along the cell surface, we demonstrate a consistent and optimal centrosome position close to the cell center for off-centered positioning of the nucleus and right above the nuclear surface for a centrally located nucleus. Our model construction is in tune with the observed MT arrangement in DCs, presenting a limited number of MTs growing along the cell surface, and a significant number of them surrounding the nuclear surface. Whether the hindrance of MT growth along the cell surface is attributed to the surface ruggedness of DCs or whether it involves the potential role of MT-end binding proteins actively promoting MT catastrophe along the surface[55,65,66], requires further experiments. Interestingly, enhanced MT gliding along the cell surface yields a very different outcome. It changes the optimal centrosome position from perinuclear to near the cell surface. The outcome is significant in MT-driven search processes, where stable MTs are guided by the topology of the cell surface. For instance, in T cells, which share the common IS with DCs, MTs appear to predominantly slide and curve along the cell surface while approaching the IS. The dynein molecules residing at the IS capture the approaching MT filaments and facilitate centrosome relocation toward the IS by generating tension on the MT[50,67,68]. Our findings underscore the heterogeneity of immune cells regulating their intracellular organization to generate immune responses specific to particular cell types.

DCs carrying multiple centrioles demonstrated a propensity for tight clustering of centrioles during antigen-specific DC-T cell interactions. Our computational study, in line with experimental observations, further illuminated that during the activation of multiple T cells, clustering of centrioles in DCs led to efficient capture of the IS by dynamic MTs, accelerating the activation of T cells. Previous studies highlighted the role of a linker composed of Rootletin and C-Nap1 in centriole tethering, physically holding the centrosomes together during interphase and dissociating during the onset of mitosis, enabling

centrosome separation[69–72]. Two recent studies pointed to the role of the kinesin-14 family members Kif25 and KIFC1 in coalescing super-numerary centrosomes into a single pole[73,74]. Intriguingly, our study suggests an alternative mechanism for positioning closely placed multiple centrioles during interphase in DCs, involving the dynamic interaction of MTs with the cell and nuclear surfaces. Our findings reveal that due to these mechanical forces, multiple centrioles remain tightly clustered throughout their temporal evolution, without requiring any explicit molecular interactions between the individual centrioles. Multiple pathways likely coordinate to lay down a robust clustering of the centrioles crucial for the functioning of DCs. While our simplified model demonstrates that MT-based pushing forces from the cell cortex and nuclear surface can suffice to position centrosomes near the nuclear periphery, previous studies have identified a broader range of factors contributing to centrosome positioning. These include MT pushing against the cell boundary[59,75,76], cortical dynein pulling[60,77,78], pushing forces due to MT-organelle interactions[60], cytoplasmic dynein-mediated pulling via vesicles attachment[60,79], and actomyosin-driven centripetal flows[60,80]. Though beyond the scope of this study, incorporating these mechanisms into future models will further illuminate the multifaceted regulation of centrosome positioning in DCs.

From a clinical perspective, centriole clustering is emerging as a novel strategy to specifically target cancer cells, which frequently harbor amplified centrioles[81–84]. To avoid spindle multipolarity, cancer cells cluster amplified centrioles during cell divisions in order to form a pseudo-bipolar spindle configuration[85,86]. However, transient centriole de-clustering leads to the formation of a multipolar spindle intermediate and mis-segregation of chromosomes[85]. In this context, de-clustering agents are currently tested in pre-clinical trials as centriole de-clustering induces multipolar spindle formation and subsequent cell death[42,87]. Similar to cancerous cells, centriole clustering in immune cells prevents the formation of multiple MTOCs, thereby maintaining one single centriolar cluster at the cell center, that allows efficient T cell priming. As a physical and functional consequence of de-clustered centrioles and multiple MTOCs, MT docking at the IS is delayed, as shown by our computational model. However, the mechanistic link between MT docking and T cell priming remains incompletely understood. Moreover, the precise contribution of MT organization - and possible off-target effects of PJ-34 treatment - on synapse architecture and vesicle trafficking are unclear and merit further investigation.

In line with our findings on extra centrioles boosting DC immunogenicity, enhanced effector functions of immune cells in which centrioles have been artificially amplified by PLK4 overexpression, were also observed in B cells processing and presenting antigens and in microglia phagocytosing dead neurons[88,89].

Enhanced T cell activation by B cells seems to be linked to changes in antigen quality indicating that multiple centrioles may influence intracellular antigen processing and/or trafficking pathways. In this context, future studies need to clarify, whether and how enhanced MT nucleation impacts trafficking and secretion of T cell stimulatory cytokines in DCs. In addition to DCs, extra centrioles have recently been documented during early B cell development[90] further demonstrating that amplification of centrioles may contribute to regular cell

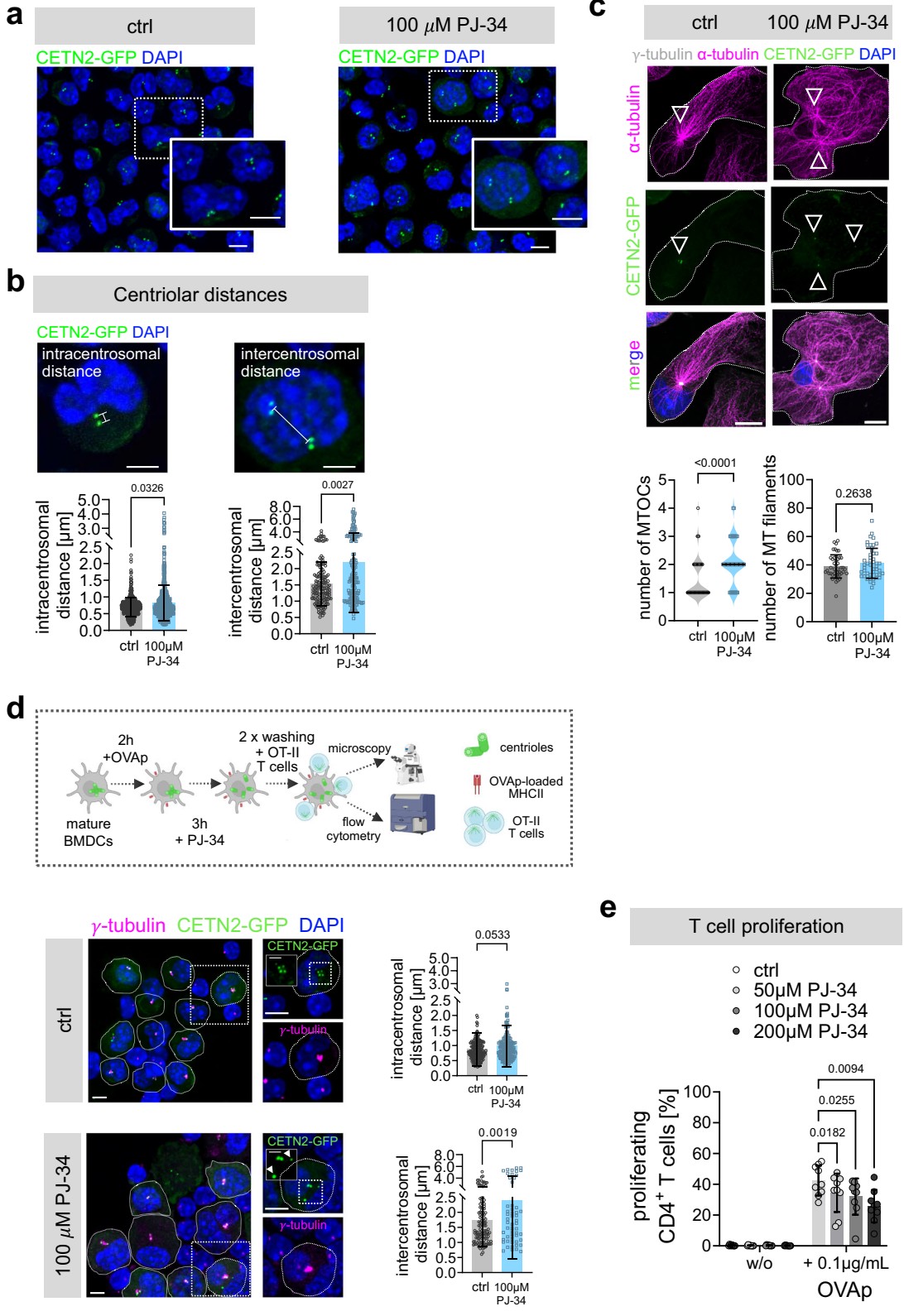

and tissue physiology within the immune compartment to enhance specific effector functions in a context-dependent manner[91].

In summary, our study highlights the importance of optimal centrosome positioning, centriole clustering, and MT organization in APCs in driving effective immune responses, and underscores the need to further explore the functional implications of amplified centrioles in immune cell effector mechanisms.

## Methods

### Mice

All mice used in this study were bred on a C57BL/6 J background and maintained at the institutional animal facilities in accordance with the German law for animal experimentation. Permission of all experimental procedures involving animals was granted and approved by the local authorities (Landesamt für Verbraucherschutz und Ernährung

**Fig. 7 | Centriole de-clustering impairs T cell activation. a** Visualization of centrioles in CETN2-GFP expressing BMDCs after PJ-34 treatment. Insets show magnification of indicated region. Nuclei were counterstained with DAPI (blue). Scale bars, 5 $\mu$m. **b** Visualization and quantification of intracentrosomal (left) and intercentrosomal (right) distances in cells treated with PJ-34. Merged images of CETN2-GFP (green) and DAPI (blue). Scale bar, 2 $\mu$m. Graphs display mean values ± s.d. Each data point represents one cell derived from three independent experiments. Left: $N$ = 542 (ctrl) /558 (PJ-34); Right: $N$ = 153 (ctrl) /159 (PJ-34). $P$ values from two-tailed Mann-Whitney test. **c** Upper panel: immunostaining of PJ-34-treated and control cells. Merged and individual channels of CETN2-GFP (green), α-tubulin (magenta), γ-tubulin (white) and DAPI (blue) are shown. Scale bars, 10 $\mu$m. Lower panel: quantification of MTOCs (left) and MT filaments (right) in 2 N PJ-34 treated and control cells. Left: Graph shows median and distribution of data points of at least three independent experiments. $N$ = 54 (ctrl) /51 (PJ-34). $P$ value from two-tailed Mann-Whitney test. Right: Graph displays mean values ± s.d. Each data point represents one cell derived from three independent experiments. $N$ = 45 (ctrl) /48 (PJ-34). $P$ value from two-tailed, unpaired Student's $t$-test. **d** Upper panel: schematic experimental layout created with BioRender. Below: 2 N CETN2-GFP (green)

expressing DCs treated with or without the de-clustering agent PJ-34 and after 2 h of co-culture with OT-II T cells stained against γ-tubulin (magenta). Nuclei were counterstained with DAPI (blue). Scale bars, 5 $\mu$m. Insets show magnification of indicated regions with CETN2-GFP (green) and γ-tubulin (magenta) channels separated. Scale bars insets, 5 $\mu$m. CETN2-GFP inset scale bars, 2 $\mu$m. White arrowheads point to de-clustered centrioles. Right: Quantification of intracentrosomal and intercentrosomal distances in cells treated with PJ-34. Graphs display mean values ± s.d. Each data point represents one cell derived from three independent experiments. Upper graph: $N$ = 226 (ctrl) /195 (PJ-34) Lower graph: $N$ = 120 (ctrl) /56 (PJ-34). $P$ values from two-tailed, unpaired Student's $t$-test. **e** OT-II T cell proliferation with or without PJ-34 treatment of OVAp-loaded mature BMDCs. BMDCs were sorted for 2 N and enriched for cells with >2 centrioles (CETN2-GFP^high). Graph displays mean values ± s.d. Each data point represents one independent experiment with $N$ = 10.000 cells analysed per condition. Cells were derived from three different mice. $P$ values from two-way Anova with Dunnett's multiple comparison. **a**–**d** All images represent maximum z-projections. Source data are provided as a Source Data file. ctrl: control, w/o: without.

North Rhine-Westphalia [LAVE NRW under AZ81-02.05.40.19.022 and AZ81-02.04.2021.A319]). CETN2-GFP and Nur77^GFP mice were purchased from Jackson (CB6-Tg(CAG-EGFP/CETN2)3-4Jgg/J and B6N.B6-Tg(Nr4a1-EGFP/cre)820Khog/J). OVA-specific OT-II mice were a gift of Sven Burgdorf. Alternatively, OT-II mice were purchased from Charles River and intercrossed with B6.SJL-Ptprca Pepcb/BoyJ (congenic CD45.1 mice, also purchased from Charles River) for one generation to obtain heterozygous OT-II CD45.1/2 mice that were then used as donors for the in vivo experiments. Mice from both sexes were used and sexes matched for in vivo experiments.

Housing conditions: Animals are provided food and autoclaved water *ad libitum*. Cages are ventilated via a central air conditioning system and kept at a temperature of 21 ± 1 °C and a relative humidity of 55 ± 15%. The air is exchanged 70 times per hour. Furthermore, animals are kept at a 12 h/12 h day/night rhythm. During the day 400 lux of light are provided whereas during the night a red light is switched on.

### Dendritic cell culture

Femurs and tibias from legs of 3–5 month-old CETN2-GFP expressing mice were removed and placed in 70 % ethanol for 2 min. Bone marrow was flushed with PBS using a 26 gauge needle. $2 \times 10^6$ cells were seeded per 100 mm petri dish containing 9 ml of complete medium (Roswell Park Memorial Institute (RPMI) 1640 supplemented with 10% Fetal Calf Serum, 2 mM L-Glutamine, 100 U/mL Penicillin, 100 µg/mL Streptomycin, 50 µM ß-Mercaptoethanol; all Gibco) and 1 mL of Granulocyte-Monocyte Colony Stimulating Factor (GM-CSF, supernatant from hybridoma culture). On day 3 and 6 complete medium supplemented with 20% GM-CSF was added to each dish. To induce maturation, cells were stimulated overnight with 200 ng/mL lipopolysaccharide (LPS) and used for experiments on day 8-9. Alternatively, DCs were frozen in FCS containing 10% DMSO on day 7. For experimental use they were thawed the day before the experiment and stimulated with 200 ng/mL LPS overnight.

To prevent new pro-centriole formation cells were cultured in the presence of the PLK4 inhibitor Centrinone (Tocris; 250 nM or 500 nM) or control (solvent DMSO) during differentiation and maturation. To induce microtubule depolymerization, cells were treated with 1 µM pretubulysin or control (solvent DMSO) for 1 h after antigen-loading. Cells were washed (wash-out) or not (w/o wash-out) with full media before T cell addition. To induce centrosome de-clustering PJ-34 (Sigma-Aldrich) was used. Cells were loaded with OVAp for 2 h and subsequently treated with 50, 100 or 200 µM of PJ-34 or control (solvent $H_2O$) for 3 h. Cells were washed two times with full media and incubated with OT-II-specific T cells at the indicated time points.

### Flow cytometry

For flow cytometric analysis, cells were washed with PBS once and incubated 10 min with anti-CD16/CD32 antibody (1:100) in blocking buffer (1x PBS, 1% BSA, 2 mM EDTA). Staining with fluorescently labelled antibodies diluted in blocking buffer was carried out for 20 min at 4 °C. For intracellular cytokine staining, cells were fixed and permeabilized with the Foxp3/Transcription Factor Staining Buffer Set (eBioscience) according to the manufacturer's protocol. The following antibodies were used: hamster anti-mouse CD11c-PE (N418, BioLegend 117308, 1:500), rat anti-mouse MHCII (I-A/I-E)-APC-Cy7 (M5/114.15.2, BioLegend 100222, 1:800), rat anti-mouse MHCII (I-A/I-E)-eFluor450 (M5/114.15.2, Invitrogen 48-5321-82, 1:800), rat anti-mouse CD4-APC (RM4-5, BioLegend 100516, 1:500), rat anti-mouse CD19-PE (6D5, BioLegend 115507, 1:500), rat anti-mouse CD19-Pacific Blue (6D5, BioLegend 115523, 1:500), rat anti-mouse MHCII (I-A/I-E)-PE-Dazzle (M5/114.15.2, BioLegend 107648, 1:600), hamster anti-mouse CD69-FITC (H1.2F3, BioLegend 104505, 1:200), hamster anti-mouse CD69-PE-Dazzle (H1.2F3, BioLegend 104536, 1:200), rat anti-mouse CD62L-PE-Cy7 (MEL-14, BioLegend 104417, 1:500), rat anti-mouse OX40 (CD134)-BV711 (OX-86, BioLegend 119421, 1:300), rat anti-mouse IL-6-PE (MP5-20F3, BioLegend 504503, 1:300), mouse anti-mouse CCL5-PE (2E9/CCL5, BioLegend 149103, 1:300), rabbit anti-mouse CXCL1-Alexa Fluor 594 (1174 A, R&D Instruments IC4532T, 1:300). For live dead staining DRAQ7 (BioLegend 424001, 1:500) or the Zombie Aqua Fixable Viability Kit (BioLegend 423101) was used. After staining, cells were washed once with blocking buffer and data acquired at the LSRII flow cytometer (BD Bioscience). Data analysis was performed using FlowJo v10.8.1.

To determine T cell activation via CD69 upregulation and CD62L downregulation, DC co-cultures with splenocytes were analysed after 20-22 h via flow cytometry as described above. OX40 upregulation on the T cell surface was determined after 2.5 days. T cell proliferation rates were assessed by dilution of CFSE. To this end, prior to incubation with DCs, splenocytes were stained with a final concentration of 0.5 µM CFSE (Invitrogen) for 7 min at 37 °C in PBS and washed with complete medium. Co-cultures were incubated for 2.5 days. Nur77-dependent GFP upregulation was also determined via flow cytometry after the indicated timepoints of naïve CD4$^+$ T cell co-culture with DCs.

### Sorting of 2 N BMDCs

To sort DCs based on their DNA content, mature BMDCs were harvested and stained with Vybrant DyeCycle Violet Stain (1:1000, Thermofisher; V35003) in RPMI without phenol red for 20 min at 37 °C. Cells were sorted at the ARIAIII Sorter (BD Bioscience) according to their ploidy level with focusing on diploid cells (2 N) and dismissing polyploid cells. Cells were re-analysed after the sort to ensure purity of

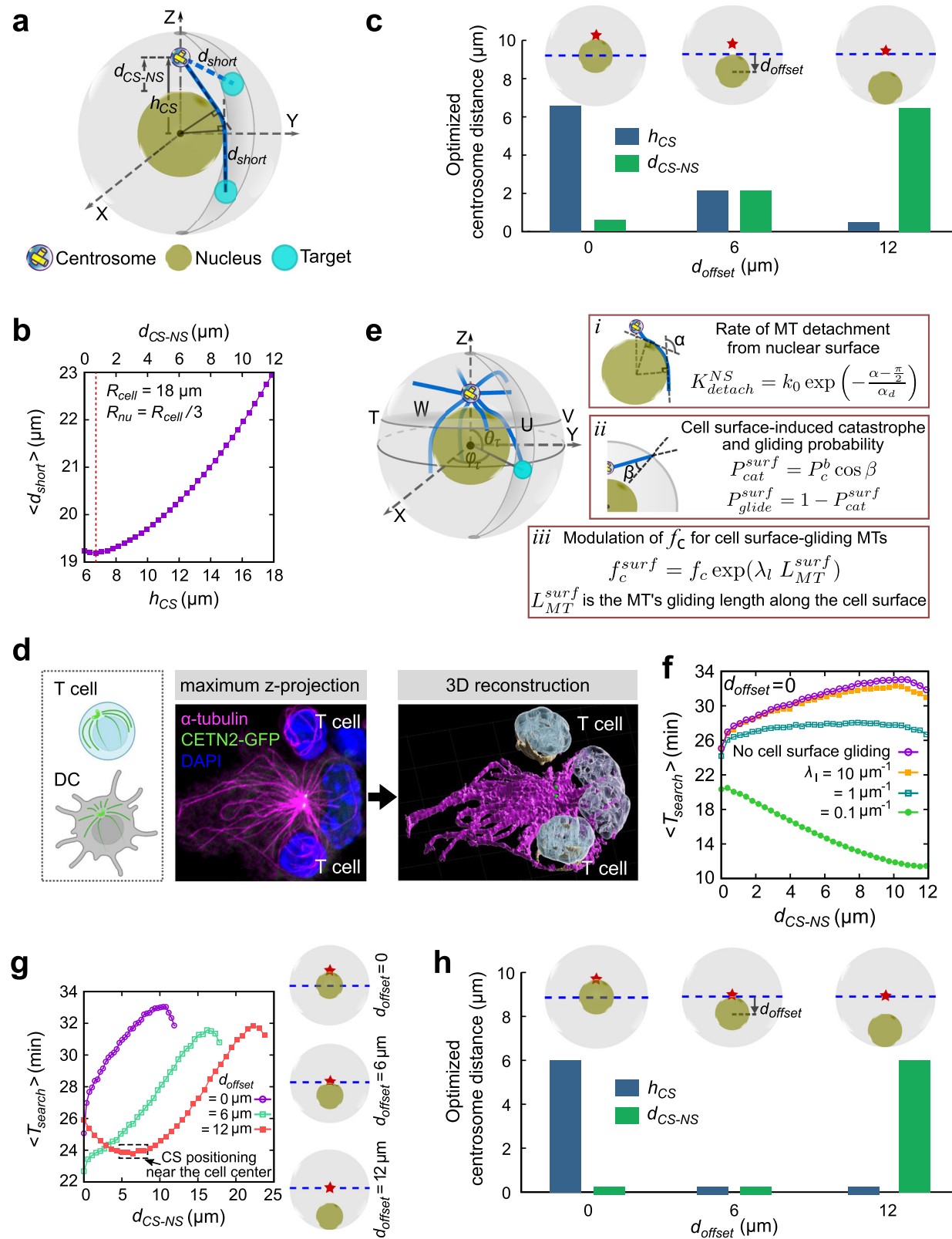

the individual subpopulations. Afterwards, cells were recovered in full medium at 37 °C for at least 30 min.

**Mixed lymphocyte reactions (MLR)**

For DC-T cell co-culturing assays sorted DCs ($1 - 2 \times 10^4$ cells/well) were seeded in 96-well U-bottom plates. After recovering time of 30 min, cells were incubated with OVAp antigen (OVA$_{323\text{-}339}$: specifically

recognized by CD4$^+$ OT-II T cells; 0.01 µg/mL, 0.1 µg/mL, 1 µg/mL) or without OVAp (controls) for 2 h. In the meantime, splenocytes were isolated from OT-II mice or Nur77$^{GFP}$/OT-II mice. Therefore, splenic, inguinal, axillary and brachial lymph nodes were removed and smashed through a 70 µm filter using PBS and a syringe piston. After centrifugation (400 x g 5 min), ACK lysis buffer (Gibco) was added for 5 min at RT before stopping the red blood cell lysis with PBS containing

**Fig. 8 | Modeling centrosome positioning in DCs. a–c** Average shortest geometric distance between centrosome and target center on cell surface regulating centrosome position in DCs. **a** A schematic representation of the shortest distance, $d_{short}$, for the target positions directly or indirectly accessible due to nuclear hindrance. **b** Plot of $<d_{short}>$ vs $h_{CS}$ in the presence of a nucleus located centrally to the cell. $h_{CS}$, represents the centrosome distance from the cell center. The top x-label in (b) represents the centrosome distance away from the nuclear surface, $d_{CS-NS}$. The vertical red dotted line marks the location of the minimum $<d_{short}>$ with respect to $h_{CS}$ and $d_{CS-NS}$. **c** Optimized values of $h_{CS}$ and $d_{CS-NS}$, corresponding to the minimum of $<d_{short}>$ for different centrosome positions along the z axis above the nucleus and for different off-centered positions of the nucleus, described by the values of $d_{offset}$. $d_{offset}$ denotes the distance between the cell center and the nucleus center and is graphically depicted in the inset of (c). **d** 3D reconstruction of the MT cytoskeleton in DCs and T cells. Left: sketch illustrating centrosomal MT growth in T cells and DCs created using BioRender. Note that MTs in T cells grow tangentially relative to the cell membrane, while in DCs MTs project astrally from the centrosome to the cell periphery. Middle and right: immunostaining of DC-T cell conjugates against α-tubulin (red) and 3D reconstruction of MTs in DCs (red) and T cells (gold). Nuclei were counterstained with DAPI (blue) and after 3D reconstruction displayed in different shades of blue in DCs (dark blue) and T cells (light blue). **e–g** Optimized search and capture of IS dictating optimal centrosome

positioning. **e** A schematic representation of the simulation model, involving dynamic MTs (blue) emanating from the centrosome (yellowish) and searching for the IS (cyan) located on cell surface. **e, i, ii)** Visual representation and associated probabilistic considerations of the MT dynamics upon hitting the nuclear and cell surface, respectively. The angle α determines the chances of MTs dissociation from nuclear surface with increased MT curvature along the nuclear surface. The parameter, β, dictates the chances of cell surface-induced instant catastrophe of MTs. **e, iii)** Modulation of the catastrophe frequency of the MTs gliding along the cell surface. **f** Average search time $<T_{search}>$ vs $d_{CS-NS}$ for a centrally located nucleaus ($d_{offset}=0$) and for different $\lambda_l$ that modulates MTs catastrophe along cell surface, compared with the scenario where MTs are not allowed to glide along the cell surface (no cell surface gliding). **g** Plot of $<T_{search}>$ vs $d_{CS-NS}$ for $d_{offset}=0$, 6 μm, and 12 μm, respectively, and without MTs gliding along the cell surface. The optimal centrosome positions denoted by the red star marks are schematically shown on the right. The horizontal blue dashed lines represent the position of the cell's mid-plane. **h** Optimized values of the centrosome distance from the cell center, $h_{CS}$, and from the nuclear surface, $d_{CS-NS}$, corresponding to the minimum of average search time, $<T_{search}>$, for different off-centered positions of the nucleus (see Fig. 8g). **a, e** Yellow cylinders represent the centriole assembly. **(a, c, e, g, h)** Schemes were custom-drawn using Inkscape (v0.92.5). Source data are provided as a Source Data file.

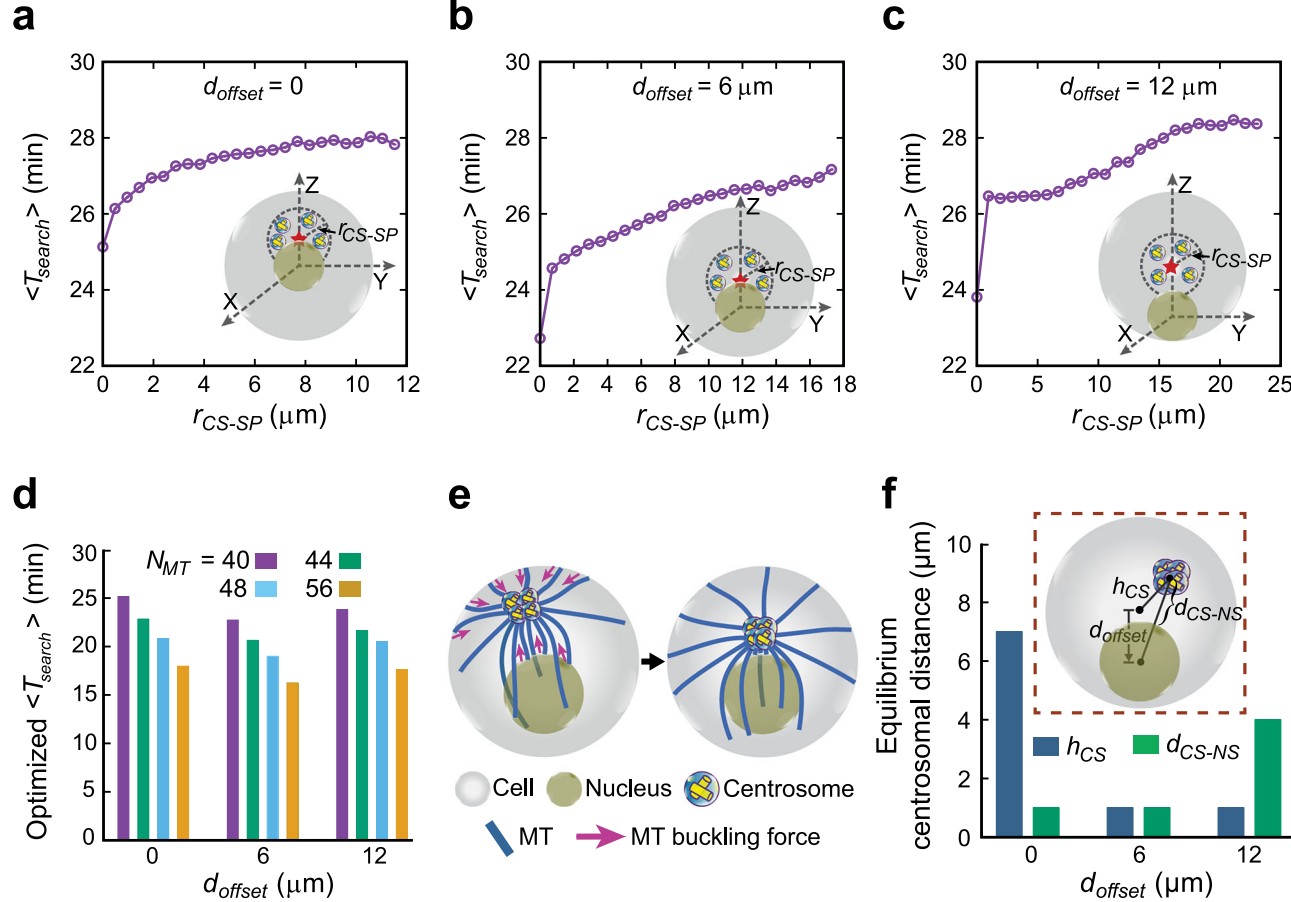

**Fig. 9 | Modeling T cell priming in the presence of multiple centrioles.**
**a–c** Clustering of multiple centrioles through centrosomal aggregation promotes optimized search. Average search time $<T_{search}>$, is plotted against $r_{CS-SP}$ for different off-centered positions of the nucleus and without MTs gliding along the cell surface. $r_{CS-SP}$ is the radius of the imaginary sphere centered at the optimal centrosome position (denoted by red star marks) obtained in Fig. 8g, h within which centrosomes are randomly placed. The insets provide illustrations of nuclear positioning and the placement of centrosomes around optimal positions for various off-centered positions of the nucleus. **d** Average search time $<T_{search}>$ vs $d_{offset}$ for clustered centrioles (all centrosomes co-localized, $r_{CS-SP}=0$) and for

different number of centrosomal MTs. **e, f** A mechanistic force balance model, considering MT's interaction with the nuclear and cell surfaces, supports observed centrosomal positioning in DCs. **e** A schematic representation of the positioning of four closely placed centrosomes in DCs, consisting centriole aggregates, governed by pushing forces generated by MT buckling at the cell and nuclear surface. **f** The final equilibrium distance of the centriole-cluster (centroid of the centrosomal aggregate) from the nuclear surface, $d_{CS-NS}$ and cell center, $h_{CS}$ (see the inset for a schematic depiction) for $d_{offset}=0$, 6μm, and 12μm, respectively. Yellow cylinders represent the centriole assembly. **a–c, e, f** Schemes were custom-drawn using Inkscape (v0.92.5). Source data are provided as a Source Data file.

2 % FCS and 2 mM EDTA. Cells were filtered through a 40 μm filter, centrifuged and adjusted to the respective cell number to co-culture with DCs or for subsequent naïve CD4$^+$ T cell isolation. Naïve CD4$^+$ T cell isolation was performed according to the manual of the EasySep Mouse Naïve CD4$^+$ T Cell Isolation Kit (STEMCELL Technologies). After removing the antigen from the DCs, T cells were added to DCs in a ratio of 1:2 or 1:5.

### IL-2 ELISA of MLR
For quantification of IL-2 cytokine levels via ELISA, DCs and T cells were co-cultured as described above (MLR). $5 \times 10^4$ DCs and $10 \times 10^4$ T cells were co-cultured in 200 μL culture volume for 20 h. Supernatants were harvested and incubated with the mouse IL-2 ELISA Kit (Invitrogen) according to the manufacturer's instructions.

### mRNA expression levels
For mRNA quantification after PJ-34 treatment, $1 \times 10^6$ BMDCs were harvested in 350 μL Lysis Buffer (RNeasy Lysis Buffer + 1% ß-Mercaptoethanol), and RNA isolation was carried out using the RNeasy Mini Kit (all products purchased from Qiagen). RNA concentration was determined using Nanodrop 1000 spectrophotometer. Gene expression was assessed using the TaqMan RNA-to-CT 1-Step Kit (Thermo Fisher Scientific) with a reaction volume of 20 μL containing 250 ng RNA template and 1 μL of Taq Man Gene Expression Assay (Thermo Fisher Scientific; duplicates performed). Samples were run on a CFX96 Real-Time System (BioRad) according to the manufacturer's instructions. Data were normalized according to the expression of a housekeeping gene in DCs (TATA-binding protein). Analysis of relative gene expression was carried out using the CFX Manager Software Version 3.1 (BioRad).

### Under-agarose interaction assay
To allow visualization of centrioles during live cell imaging, DC-T cell co-cultures were injected under a block of agarose to prevent cell floating during the imaging period. Therefore, a custom-made chamber was built by gluing a 1-cm plastic ring with paraffin into a glass-bottom dish. 1 % agarose solution was prepared by mixing 0.2 g UltraPure agarose (Invitrogen, 16500-100) with 5 mL water, 5 mL 2x Hanks' Balanced Salt Solution (HBSS 10x, Gibco, 14185052) and 10 mL phenol red-free RPMI 1640 Medium supplemented with 20% FCS and 1% penicillin 100 U/mL/streptomycin 100 μg/mL. For live cell imaging, ascorbic acid was added to a final concentration of 50 μM. 500 μL of the heated agarose was poured into each chamber. After polymerization, the dishes were filled with water around the agarose and incubated for 45–60 min at 37 °C, 5% CO$_2$ to equilibrate the agarose. For the experiment, cells were injected with a small tip under the agarose in a volume of 0.4–0.6 μL. To allow visualization of interaction during live cell imaging, T cells were stained with the calcium-sensitive dye Cal520 (Abcam) prior to injection. For efficient staining, cells were incubated with 3 μM Cal520 in phenol red-free RPMI 1640 medium supplemented with 20% FCS for 30 min at 37 °C. To avoid toxic effects the dye was efficiently removed by washing two times. Live cell imaging was started directly after DC and T cell injection. Alternatively, to allow immunofluorescence staining, cells were fixed after 60-90 min in the incubator with 4 % paraformaldehyde (PFA) solution overnight at 4 °C. The next day, agarose was removed carefully and the cells were washed with PBS two times before staining.

### Adoptive transfer experiments
For in vivo DC-T cell interaction experiments, naïve OT-II T cells were isolated using the EasySep Mouse Naive CD4$^+$ T Cell Isolation Kit (STEMCELL Technologies) according to the manufacturer's instructions. Naïve CD4$^+$ OT-II T cells were labelled with the CellTrace Far Red Cell Proliferation dye (Invitrogen) at a concentration of 0.5 μM and $2 \times 10^6$ naïve OT-II cells were injected into the tail vein of C57BL6/J

wildtype mice. After 24 h, $1 \times 10^6$ mature CETN2-GFP expressing BMDCs, which were loaded with 1 μg/mL OVAp, were injected subcutaneously into the hocks of both hind legs. Popliteal and inguinal lymph nodes were harvested after 48 h and incubated in 1 % PFA in PBS over night at 4 °C. Subsequently, LNs were washed with PBS and dehydrated in a sucrose gradient (10 % - 30 % sucrose in PBS). LNs were embedded in cryomedium (Tissue-Tek O.C.T. Compound, Sakura) and stored at -80 °C until they were cut into 20 μm thick sections.

**Staining of cryosections.** Staining of cryosections was carried out at room temperature. Sections were permeabilized with 0.3% Triton X-100 in PBS for 10 min and then blocked with 1 % BSA in 0.3 % Triton X-100 in PBS for 30 min. All primary and secondary antibodies were used in a dilution of 1:200 in 1% BSA in 0.3% Triton X-100 in PBS. Sections were incubated with primary antibodies overnight. After washing 3 × 10 min with PBS, secondary antibodies were added for 2 h. DAPI staining (1:1000 in PBS, Stock 1 mg/mL, Sigma) was carried out for 30 min. After 3 × 10 min washing in PBS, sections were incubated in Ce3D tissue clearing solution (BioLegend) for at least 2 h and then mounted with mounting medium (Invitrogen).

### Immunofluorescence
Cells were immobilized by incubating 1-2 μL of cell suspension on uncoated cover slips for 5 min at 37 °C before adding 4% PFA for 20 min. For MT staining, cells were fixed after injection under agarose as described above (section: Under-agarose interaction assay). Fixed cells were washed twice with PBS for 10 min. To allow intracellular antibody staining cells were permeabilized adding 0.2% Triton X-100 (Sigma) in PBS for 30 min at room temperature. After washing 3 × 10 min with PBS samples were incubated in blocking solution (5% BSA (Roth) in PBS) for 1 h to prevent unspecific binding of the antibodies. Next, samples were incubated with primary antibodies diluted in blocking solution over night at 4 °C. Staining for PCNT was carried out for 1 h at room temperature. Afterwards, cover slips were washed 3 × 10 min with PBS and stained with secondary antibodies diluted in blocking solution in the dark for 1 h at room temperature. After three times washing 10 min with PBS cover slips were mounted with DAPI-containing mounting medium (Invitrogen) and sealed with nail polish before imaging.

The following primary antibodies were used: rat anti-mouse alpha-tubulin (YL1/2, Invitrogen MA1-80017, 1:500), mouse anti-mouse acetylated-tubulin (6-11B-1, Sigma-Aldrich T7451, 1:500), mouse anti-mouse γ-tubulin (GTU-88, Sigma-Aldrich T6557, 1:500), rabbit anti-mouse γ-tubulin (polyclonal, Abcam AB11317, 1:500), rabbit anti-mouse CDK5RAP2 (polyclonal, Sigma-Aldrich 06-1398, 1:500), rabbit anti-mouse pericentrin (EPR21987, Abcam AB4448, 1:200), hamster anti-mouse TCRβ chain-Biotin (H57-597, BD Bioscience 553168, 1:100).

The following secondary antibodies were used in a dilution of 1:400: Donkey Anti-Mouse Alexa Fluor 647 AffiniPure F(ab')$_2$ Fragment IgG (H + L)(715-606-150), Donkey Anti-Mouse Cy3 AffiniPure F(ab')$_2$ Fragment IgG (H + L)(715-166-151), Donkey Anti-Rat Cy3 AffiniPure F(ab')$_2$ Fragment IgG (H + L)(712-165-150), Donkey Anti-Rabbit Alexa Fluor 647 AffiniPure F(ab')$_2$ Fragment IgG (H + L)(711-606-152), Goat Anti-Rabbit Cy3 AffiniPure F(ab')$_2$ Fragment IgG (H + L)(111-165-144), Streptavidin-Cy3 (016-160-084) (all from Jackson ImmunoResearch).

### Microscopy
Confocal microscopy was performed on a motorized stage at RT with an inverted microscope equipped with an Airyscan module; an EC Plan-Neofluar 10×/0.30 objective; a Plan-Apochromat 63×/1.4 oil DIC objective; 488, 561, and 633 laser lines; and a photomultiplier tube (all Zeiss). For fixed samples 0.2 μm sections were acquired leading to z-stacks of mostly 4-8 μm height (cells) or 16-20 μm height (tissue samples). To analyse MT filaments, images were acquired using the Airy module and posttreated by deconvolution. The same confocal

imaging set up was used for live cell imaging. During live cell acquisition dishes were placed in a 37 °C chamber and cells imaged at a 10 to 20 sec interval for 30-60 min. The auto-focus option was used to keep the centrioles in focus. For all experiments, imaging software ZEN Black 2.3 SP1 was deployed.

**Image analysis.** Image processing and data analysis was performed using ImageJ and Imaris 10.2.0. To prepare images for display in the manuscript, brightness and contrast were adjusted and smoothing was applied. Number of centrioles was determined from multi-z-stack images by counting individual CETN2-GFP foci co-localizing with a PCM marker (γ-tubulin, CDK5RAP2, PCNT) or ac-tubulin. Quantitative intensity measurements were carried out on maximum z-projections measuring RAW integrated density or integrated density of selected regions of interest (ROIs). ROIs around the centrioles were selected by keeping the same size (~ 8 μm²) (*see* Fig. 1c). Overlapping cells were excluded from the analysis.

For determining intracellular MT numbers, MT filaments were counted manually at a defined round-shaped area (25 μm²) around the centrioles or acentriolar MTOCs (*see* Supplementary Fig. 2c). All z-planes were used to precisely detect individual MT filaments. Tracing of MTs was carried out by using the NeuronJ plugin using maximum z-projections. MTOCs were defined by the following criteria: *i)* a clear PCM foci (γ-tubulin, CDK5RAP2 or PCNT) is visible, which *ii)* nucleates MT filaments from the respective region.

For measurements of distances between or from centrioles, maximum z-projections of fixed samples were used to determine centriole coordinates in two dimensions leading to x,y values. However, whole confocal stacks were used to determine the position of the T cell's centrosome and identify cell-cell contacts. In time lapse videos, centrioles were tracked by using the Manual Tracking plugin. Obtained coordinates were used to calculate the centriole center point coordinates using the following formulas: $x_{CP} = \frac{\sum x_n}{n}$ *and* $y_{CP} = \frac{\sum y_n}{n}$ with $n$ : number of single centriole.

In maximum z-projections of fixed samples as well as in time lapse videos, cell outlines were marked with the 'freehand selection' tool. Subsequent measurement of this ROI revealed area and centroid. Distances between two points were calculated using the formula $distance\ d = \sqrt{(x_2 - x_1)^2 + (y_2 - y_1)^2}$.

**3D reconstruction of DC-T cell contacts in lymph nodes.** 3D TIF files were processed and analyzed using Imaris 10.2.0 (Bitplane) for quantitative analysis. T cells and DCs and their respective nuclei were segmented based on DAPI staining, cytoplasmic fluorescence signals from the CellTrace far red cell proliferation dye (T cells) or thresholding GFP expression (DCs). The built-in watershed algorithm in Imaris was used for cell boundary delineation, and centriole detection utilized the "spots" function with automated size filtering, followed by 3D distance calculations between paired structures using Imaris' measurement function. All volumetric reconstructions were generated through Imaris' integrated rendering engine.

**Expansion microscopy (ExM)**
Control or Centrinone-treated BMDCs were seeded on poly-L-lysine (0.1 mg/mL, Sigma) coated coverslips and allowed to adhere for 1 h at 37 °C. U-ExM protocol was carried out as described previously[92,93] with mild adjustments. Cells were fixed with 4 % PFA in PBS at 37 °C for 15 min, following post-fixation with 0.7% PFA and 2% acrylamide (AA) at RT overnight followed by PBS wash prior to gelation. Gelation was carried out in a wet chamber placed on ice and covered with Parafilm. Washed coverslips were transferred promptly on a small drop of gelation solution (19 % sodium acrylate, 10 % AA, and 0.1 % N, N'-methylenebisacrylamide initiated with 0.5% tetramethylethylendiamine and 0.5 % ammonium persulfate). Gelation was

initiated on ice for 5 min and then transferred to 37 °C for 30 min to allow polymerization. Gels were then detached from the coverslips in a denaturation buffer (50 mM Tris-base, 200 mM NaCl, 200 mM SDS in ddH₂O, pH 9) and denatured at 95 °C for 1 h. Gels were expanded with 3x ddH₂O wash and a small piece of gel was cut out for staining. Staining was performed sequentially overnight at RT in a staining buffer (2 % BSA, 10 % sodium azide in PBS) with 3x ddH₂O wash between each step with the following primary: mouse anti-mouse acetylated-tubulin (C3B9, Sigma-Aldrich 00020913, 1:10), rabbit anti-mouse γ-tubulin (polyclonal, Abcam AB11317, 1:200), rabbit anti-mouse CDK5RAP2 (polyclonal, Sigma-Aldrich 06-1398, 1:200) and secondary: goat anti-rabbit Alexa Fluor 488 IgG H + L (Invitrogen A11008, 1:500) and goat anti-mouse Alexa Fluor 555 IgG H + L (1:500, Invitrogen A21422) antibodies. NHS Ester Atto 425 (20 mg mL⁻¹ in PBS, ATTO-TEC) staining was done in PBS at RT for 1.5 h and gels were expanded with 3x ddH₂O wash before imaging. Microscopy was performed on a Nikon Eclipse Ti2 microscope equipped with Yokogawa CSU-W1 spinning disc module using the CF Plan-Apochromat VC 60×/1.2 water objective.

**Statistical analysis**
Data analysis was carried out with GraphPad Prism 10 (GraphPad Software, San Diego, CA, USA). Samples were tested for Gaussian distribution using D'Agostino-Pearson omnibus normality test to fulfil the criteria for performing Student's *t*-tests. Welch's correction was applied when two samples had unequal variances. When data distribution was not normal, Mann-Whitney test was carried out. For small data sets, Gaussian distribution was assumed but could not be formally tested. For analysis of Nur77$^{GFP}$ expression, CETN2-GFP$^{low}$ and CETN2-GFP$^{high}$ samples from individual experiments were paired. For multiple comparisons where data distribution was normal, one-way Anova was used followed by Dunnett's multiple comparisons as post-hoc test. When data distribution was not normal, Kruskal-Wallis test with Dunn's multiple comparisons was used. For multiple comparisons with more than two independent variables two-way Anova was used with Dunnett's or Šidák's multiple comparisons. All graphs display mean values ± s.d. (95% Confidence Interval). No statistical method was used to predetermine sample size. The experiments were not randomized and investigators were not blinded to allocation during experiments and outcome assessment. Individual experiments were validated separately and only pooled if showing the same trend. All *P* values are indicated in the figures.

**Computational model**
DCs were modeled as spheres with radius $R_{cell}$ containing a spherical nucleus of radius $R_{nu}$ as depicted in Fig. 8a. The line connecting the nucleus center and cell center defines the z-axis. The centrosome is placed along the z-axis at a distance $h_{CS}$, above the cell center and $d_{CS-NS} (= h_{CS} - R_{nu})$ away from the nuclear surface. The parameter $d_{offset}$ determines the distance of the nucleus center from the cell center as depicted in Fig. 8c, inset.

**Average geometrical distance between centrosomes and the target points on cell surface**
The optimal geometric centrosome position is determined by minimizing the average distance, $< d_{short} >$, between the centrosome and all target points on the cell surface. In the absence of a nucleus, this distance is the shortest straight geometric distance between the centrosome and the target points on the cell surface. In the presence of a nucleus, a region of the cell surface lies in the "geometric shadow" behind the nucleus, which is the region not accessible to straight MTs growing from the centrsosome. In such configurations, two tangents are drawn on the nuclear surface, one originating from the centrosome and the other from the target point on the cell surface, ensuring that the corresponding tangent points on the nuclear surface, centrosome,

and the centers of the nucleus and target remain on the same plane. The total distance is then determined as the sum of the lengths of the tangential segments and the intermediate arc length along the nuclear surface (see schematic in Fig. 8a). The average geometric distance, $<d_{short}>$, is then computed by averaging over $10^6$ random target points on the cell surface.

## Average search time

The optimal centrosome position is determined by the minimum of the search time, $T_{search}$, required for dynamic MTs to capture a target located randomly on the cell surface. The target zone is represented as a circular disk of radius $R_\tau$ embedded in the cell surface, as illustrated in Fig. 8e. The target is located on the cell surface at an arbitrary position, specified by the polar angle $\theta_\tau$ ($\in 0 - 180^0$) and azimuthal angle $\varphi_\tau$ ($\in 0 - 360^0$), measured from the positive $z$ axis and $x$ axis, respectively. A specific number of $N_{MT}$ MTs nucleate from the centrosome and explore the surrounding three-dimensional cellular space searching for the target. In the presence of supernumerary centrosomes with multiple dispersed centriole assemblies, the total $N_{MT}$ MTs are distributed evenly among all the centrosomes. The dynamically unstable MTs exhibit consistent growth at a velocity, $v_g$, switch to a shrinking phase with a catastrophe frequency, $f_c$, and shortening at a different velocity, $v_s$. The catastrophe frequency is chosen such that, on average, MTs could cover a distance equivalent to half of the cell perimeter. The simulations are performed with zero rescue frequency ($f_r$), preventing shrinking MTs from switching to the growth phase. A zero-rescue frequency is optimal since it minimizes the search time that MT spends exploring directions lacking the target[49,94–96]. A Monte Carlo algorithm is implemented to simulate individual MTs. At each computational time step ($\Delta t$), a uniform random number between 0 and 1 was compared with the probability $1 - \exp(-f_c \Delta t)$ that an MT switches from growth to shortening. If the random number is found to be less than this probability, the MT begins shortening. Once a MT shrinks back to the centrosome, a new growth cycle starts in another random direction.

The direction of MT nucleation from the centrosome was governed by two angles, polar angle, $\theta$ ($\in 0 - 180^0$) and azimuthal angle, $\varphi$ ($\in 0 - 360^0$), in the standard spherical polar coordinate system. Depending on the direction of nucleation and subsequent interactions with the cell or nuclear surface, two distinct scenarios arise:

(i) The model assumes that when MTs interact with the nuclear surface, they can move along the nuclear surface maintaining the same azimuthal angle $\varphi$ at which they originated from the centrosome. During this movement along the nuclear surface, the increasing curvature of the MTs can promote their detachment at a rate $K_{detach}^{NS}$. After detachment, the MTs grow tangentially from the nuclear surface and approach the cell surface searching for the target. The rate of detachment, $K_{detach}^{NS}$, is assumed to be an exponentially increasing function of the MTs' curvature along the nuclear surface, as depicted in Fig. 8e, (i). Two parameters, $k_0$ (the prefactor of the exponential term) and $\alpha_d$ (a phenomenological constant within the exponential term) regulate the sensitivity of the MTs' detachment rate (see Fig. 8e, (i)). Smaller values of $k_0$ and/or larger values of $\alpha_d$ promotes the MTs to glide along the nuclear surface, minimizing the chances of rapid detachment. MTs that extend beyond half of the nuclear perimeter from their initial contact points are immediately detached from the nuclear surface.

(ii) If the MTs reach the cell surface either after detaching from the nuclear surface or directly from the centrosome, they can eventually capture the target if the MT tip hits the target. Otherwise, they move along the cell surface seeking the target or experience catastrophe induced by the surface curvature. This cell surface induced MT catastrophe is incorporated in the

model based on previous studies demonstrating that the distribution of MTs along the cell surface is influenced by the cell shape, and ruggedness of the cell surface regulating the MT bending and catastrophe[56,77,97]. The model assumed a probabilistic catastrophe of the MTs at the cell surface depending on the angle of incidence ($\beta$) of the MT with respect to the local normal. The probability is chosen such that the tangential incidence ($\beta = \pi/2$) of MTs would promote their gliding along the cell surface and normal incidence ($\beta = 0$) would induce catastrophe at a certain rate (Fig. 8e, (ii)). Specifically, the catastrophe probability $P_{cat}^{surf} = P_c^b \cos\beta$ and the gliding probability $P_{glide}^{surf} = 1 - P_{cat}^{surf}$ are considered. The value of $P_c^b$ is chosen to be 1 to ensure that MTs experience instant catastrophe for normal incidence on the cell surface ($\beta = 0$). If the MTs begin moving along the cell surface overcoming the cell surface -induced catastrophe, the MTs' movement along the cell surface is further restrained by modulating the catastrophe frequency of gliding MTs according to $f_c^{surf} = f_c \exp(\lambda_l L_{MT}^{surf})$, where $L_{MT}^{surf}$ represents the segment of the MT's length gliding along the cell surface, and $\lambda_l$ is a phenomenological constant determining the rate at which the catastrophe frequency increased per unit length of the MT (Fig. 8e, (iii)). This additional consideration of catastrophe modulation is motivated by experimental observations of MT organization in DCs, which revealed that a very small fraction of the MTs or almost no MTs appeared to be gliding along the rugged cell surface[26]. The simulation is continued until a successful target-MT attachment is formed. The average search time, $<T_{search}>$, is calculated using $10^5$ different random positions of the target on the cell surface. The parameters used in the simulation are tabulated in Supplementary Table 1 in ***Supplementary Information***.

## A mechanistic force-balance model demonstrating centrosomal positioning in DCs

The centrosomes (CSs) are considered as small spherical objects with $r_{CS}$ ($\sim 0.5$ µm), free to move within the cellular space between the cell surface and a nucleus fixed in space at various positions within the cell. MTs are cylinders of vanishing radii and nucleated from the centrosome uniformly in all directions. The number of MTs is evenly distributed among the centrosomes, and the dynamics of each MT is governed by $v_g$, $v_s$, $f_c$ and $f_r$ discussed earlier. Based on earlier studies of centrosome positioning and stability in interphase cells, the model assumes a pushing force-dominated regime arising from MT interaction with the cell and nucleus (Fig. 9e)[59,60,98]. In our model, a growing MT of length $L_{MT}$ hitting the cell or nuclear surface buckles as per first-order Euler's buckling and translates a pushing force ($\sim 200/L_{MT}^2$) to the corresponding centrosome[59,60]. The model assumes that only a fraction of MTs hitting the nucleus buckles, while the others continue to slide along the nuclear surface. The buckling force is inversely proportional to the MT length and hence only short MTs generate significant buckling force and also undergo instant catastrophe[58,59] leading to short-lived MT pushing forces. The resultant total force on the centrosomes can move them through the viscous cytoplasm following Stokes's law[59,61]. If $F_{CS}$ is the net force acting on a centrosome, $V_{CS}$ the instantaneous velocity and $\mu$ the effective viscous drag on centrosome, then as per Stokes's law $F_{CS} = \mu V_{CS}$ (with $\mu = 6\pi\eta r_{CS}$; $\eta$ is the coefficient of cytoplasmic viscosity)[59,61]. The position of centrosomes is updated using the coarse-grained time step $\Delta t$ ($= 10^{-2}s$). The parameter values are mentioned in Supplementary Table 1 in ***Supplementary Information***.

## Reporting summary

Further information on research design is available in the Nature Portfolio Reporting Summary linked to this article.

## Data availability
Data that support the findings of this study are available within the article, its supplementary information, the Source Data file or on request from the corresponding author(s). Source data are provided with this paper.

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

## Acknowledgements

The authors acknowledge the Imaging Methods Core Facility at BIOCEV, institution supported by the MEYS CR (LM2023050 Czech-BioImaging) for their support & assistance in this work. MH and EM thank Vladimir Varga for providing the C3B9 anti-acetylated tubulin antibody. AS and RP acknowledge Indian Association for the Cultivation of Science, Kolkata, for funding and support. EK acknowledges the TRA Life and Health (University of Bonn) as part of the Excellence Strategy of the federal and state governments, the returning experts fellowship of the Ministry of Innovation, Science and Research of North-Rhine-Westphalia (AZ: 421-8.03.03.02-137069), the Deutsche Forschungsgemeinschaft (DFG, German Research Foundation – 457838313) and the Else-Kröner-Fresenius-Stiftung (2023_EKSE.15). EK, FM and DB acknowledge the DFG under Germany's Excellence Strategy – EXC 2151 – 390873048. EM acknowledges the GA UK fellowship (project No. 174725) from Charles University in Prague. MHons acknowledges the Czech Science Foundation 25-16907S, 25-16671S Czech Health Research Council NW25-08-00208, Ministry of Education, Youth, and Sports, Czech Republic via Charles University Cooperation program, research area BIOLOGY and SVV 260763 from Charles University and Czech Science Foundation 20-

24603Y and Ministry of Education, Youth, and Sports, Czech Republic via Charles University, Cooperatio program, research area BIOLOGY and SVV 260637. HR and LS acknowledge DFG - Collaborative Research Center SFB 1027 - project ID 200049484. SU acknowledges European Research Council (101039438) and DFG (533863915, 501752319 and 448121430). EK and SU acknowledge Hightech Agenda Bayern. Schematic pictures were created using BioRender as indicated in the figure legends.

## Author contributions

I.S., A.W., L.B., P.K., S.G., M.H., K.S., J.B., E.M., M.Hons., S.E., F.M. and E.K. performed experiments. P.W. and S.U. carried out 3D image analysis of lymph node sections. The U.K. provided pretubulysin, and L.S. verified its efficiency. H.R., A.S., S.S. and R.P. performed modelling of centrosome configuration and M.T. dynamics. Z.A., F.M. and D.B. gave technical support and supervised K.S., S.E. and L.B., respectively. EK designed and supervised the research. I.S., H.R., A.S., R.P. and E.K. wrote the manuscript. All authors discussed the results and implications and commented on the manuscript at all stages.

## Funding

## Competing interests

Authors declare no competing interests.
