## [Transparent Peer Review file · Nature Communications]

A centrally positioned cluster of multiple centrioles in antigen-presenting cells fosters T cell activation

Corresponding Author: Professor Eva Kiermaier

Version 0:

Reviewer comments:

Reviewer #1

(Remarks to the Author)
See attached file.

Reviewer #2

(Remarks to the Author)

In the manuscript entitled "Multiple clustered centrosomes in antigen-presenting cells foster T cell activation without MTOC polarization", by Stötzl et al. explored how centrosome amplification in DCs affects immune synapse formation and T cell activation. They integrated experimental data with computational modeling to demonstrate that extra centrioles in DCs lead to the formation of overactive centrosomes. These centrosomes cluster during DC-T cell interactions and are positioned near the cell center. Disrupting either the number of centrioles or the centrosomes configuration results in impaired T cell activation. The entire study is quite interesting, as it establishes the relationship between optimal centrosome positioning, centrosome numbers, and T cell activation. However, the author seems to have a limited understanding of the concepts, distinctions, and general functions between centrosomes and centrioles, leading to some misinterpretations of experimental data. Furthermore, I believe that using computational modeling to explain centrosome positioning may be somewhat ambitious. A more comprehensive model should incorporate additional intracellular variables and factors. If most of the issues below are resolved, I recommend that this manuscript be accepted. Specific comments:

The author raises several detailed scientific questions in the Introduction and Results sections, but it appears that few of them are thoroughly addressed.

The Introduction section is somewhat redundant, and the presentation of background knowledge is a bit disorganized and lacks logical coherence. It is recommended to succinctly present the background information and clearly state the key scientific question.

Pericentrin is the most typical marker of the PCM; it would be best if the author included additional staining for it.

The assertion that "MT organization was not affected by the loss of centrioles" is a perspective held by only some researchers. If the author wishes to make this claim, they should cite 2-3 high-quality references.

I am not an expert in the field of immunology, but I still feel that using only one experimental method, flow cytometry (Fig. 2F), to support such an important conclusion "We found that T cell activation was markedly reduced in the presence of pretubulysin", is insufficient and not solid.

For Figure 4G and its related data: Centrin-2 is primarily associated with centrioles and is a marker for centrioles. If the centrosome is concentrated in a single focus, it is defined as "one centrosome," even if it contains multiple centrioles. It is defined as "multiple centrosomes" only when there are multiple foci or if it appears dispersed. For identifying multiple centrosomes, PCNT (Pericentrin) should be used as the marker instead of Centrin-2.

I understand the novelty the author is aiming for in the final section of the Results, but explaining centrosome positioning using a geometric model seems somewhat far-fetched and unusual. If a computational model is to be used for this explanation, the evidence provided by the author is not sufficient. For instance, there are many interacting factors with the centrosome within the cell, including various organelles and variables such as fluidity. I believe this represents a highly complex computational process, unlike the current model in the paper.

Reviewer #3

(Remarks to the Author)

The authors have tried to understand how the MTOC (Microtubule organization center) organization in APC (antigen-presenting cell) has an effect on T-cell activation. It is well established that cellular polarization is essential for immune regulation, often involving centrosome reorientation. However, how centrosomes are oriented from the perspective of APCs is not entirely known. Here, the authors consider murine DCs (Dendritic cells) and observe, using experimental and mathematical models, that DCs' additional centrioles create over-active centrosomes that cluster near the cell center during DC-T cell interactions. Disrupting centriole count or centrosome positioning in DCs impairs T cell activation, emphasizing the critical role of centrosome amplification and positioning in APCs for effective T cell responses. The manuscript details an important and often overlooked aspect of T cell activation. The complementary nature of the study (experiments supported with computational models) is commendable (please note that the reviewer is not an expert in computational modeling). While the enthusiasm for the topic is high, the reviewer is less excited about the manuscript in its present form. Mainly, the loss of enthusiasm stems from overcrowding of the figures (which makes it very challenging to follow and understand the novelty of work), low sample numbers, and a few irregularities noted below.

Major comments:

1. The figures are very dense, making them hard to follow. The figure captions are very lengthy, and some require a single page. The authors are encouraged to reconsider how they should present their data.
2. Following up on the above comment, the authors should clearly highlight the novelty of their work. Does the work provide a new interpretation of T cell activation, a new method, or a new computational model? How are the experiments and model tied together?
3. The reviewer was puzzled in interpreting the data in the following figures:
 - a. In Fig 1E (left graph), what is the percentage of cells with two centrioles and no centrioles in the presence of 500 nM Centrinone?
 - b. For Fig 2, the low sample size from only two experiments raises doubts about the claims of no changes in MTOC.
 - c. The authors must provide high-resolution magnified images (insets and similar to image quality in subsequent figures) and supporting extra images in the supplementary materials (for all figures). From Figure 2b, the MTs look intact in the representative image w/o pre-tubulysin washout and with washout, which is contrary to the claims made by the authors in Line 149.
4. Following up on comment 2b, the number of MT filaments shows high variability in control experiments (Fig 1F and 2C). These numbers should be the same or similar. How are the authors defining an MT filament? Does it have a minimum length?
5. The authors mentioned that tubulin intensity is higher during IS (Immune synapse) formation in the presence of two or more centrosomes (Figure 3b). However, the provided images do not clearly show that IS formation corresponds to the centrosome number. Hence, the authors should provide magnified microscopic images of the same.
6. In Fig 4d (graph), the authors have plotted the MTOC reorientation during the co-culture at different time points. Authors must provide representative microscopic images of the centrosome alignment at other time points (4, 6, 20h) in their supporting information.
7. Since Figures 3 and 6 are inconclusive findings, the authors are encouraged to move them to supplementary materials.
8. Similar to the irregularity mentioned in comment 3, in Figure 3F, the relative frequency of mono conjugation between T cell and DC cells does not reflect the centrosome numbers in Figure 3G in the same condition.
9. In Figure 4C, It is unclear how the GFP values are observed to be below the threshold in 20h coculture. What does this signify? The reviewer is also confused on how the authors are claiming that multiple centrosomes are more efficient in activating T cells than with only one centrosome (lines 246-247), when there is no data provided for the latter.
10. In Figures 5b and 5e, the authors have provided intercentrosomal and intracentrosomal distances for 1, 2, and 4h of co-culture. Why did the authors not included the data for 6h and 20h where the maximum T cell activation was observed (Line 215 and Figure 4b)?
11. In Fig 6A and 6C (graphs), the mean values can be misleading as the individual data points are highly inconsistent. The authors should reconsider increasing the 'N' values to determine the distances.
12. In Figure 7b, c: Why did the authors use two different concentrations of PJ34 to study α -tubulin and centrosomal distance?
13. The reviewer is not an expert in computational modeling. However, the conditions outlined in the model for DCs (spherical cells) do not match the shape and culture conditions in experiments (2D). How does this impact the findings in Figures 8 and 9 and, generally, the force balance model with MT sliding?
14. A general question (and a perceived weakness of the manuscript) is how flat 2D environments recover in vivo behavior. How would the accumulation of T cells change in more physiologically relevant environments?

Minor comments:

The reviewer has some minor questions (mostly clarifications) on data organization.

- 1) Is the 500 nM Centrinone effect on depleting the 2N cells % statistically significant in supp fig 1E?
- 2) The authors should mention the different colors in the figure legends, especially for Figure 2a.
- 3) Line 62: Please correct the typo in "... week T cell priming...".

- 4) clarify the Fig 4D legend: "...each datapoint is one individual experiment, whereas N> 100 cells analyzed per condition". What does this mean?
- 5) In Figures 3c and d, the Y-axis should be the same for easy interpretation of data.
- 6) The authors should clarify the sample size information in Figure 4G- the authors state that analyzed cells are greater than 100 (line 1156), yet the data point shows only ~4.
- 7) The numbering of figures in the text is out of order.
- 8) Authors should modify the Y-axis label of Figure 5e. indicating the centrosome numbers (as per the figure legend).

Reviewer #4

(Remarks to the Author)

Reviewer #5

(Remarks to the Author)

Version 1:

Reviewer comments:

Reviewer #1

(Remarks to the Author)

The manuscript "Multiple clustered centrioles in antigen-presenting cells foster T cell activation without centrosome polarization" examines how centrosomal organization in dendritic cells contributes to immune synapse formation and activation of T cells. Overall, the manuscript addresses an important aspect of T cell activation and is timely and significant. In this revision and rebuttal, the authors have successfully addressed previous comments and improved the clarity of the manuscript. The incorporation of new adoptive transfer experiments strengthens the relevance of the authors' findings to more complex 3D environments. The authors now clearly distinguish between single-plane and projection images and have expanded the image analysis section for reproducibility, which now substantially improves transparency and replicability of their quantifications. The revised manuscript addresses most of the previous concerns and is substantially strengthened by the inclusion of additional experiments, improved figure clarity, and clearer methodological detail.

One point of note: the current title gives the impression to the reader that T cell activation does not require centrosome polarization in T cells and is thus somewhat misleading. The authors should change the title to emphasize that centrosome reorientation to the immune synapse in antigen-presenting cells is not required.

Specific comments:

- The gamma-tubulin intensity normalization methods add to the clarity. One follow-up question: were gamma-tubulin intensities normalized using the mean intensity for all two-centriole cells within the same treatment, or relative to all two-centriole cells across all treatments? Can the authors please clarify this point in the manuscript.
- The authors responded constructively to concerns regarding potential off-target effects of PJ-34. By measuring MHCII expression, cytokine mRNA and protein levels, and testing multiple concentrations, they make a strong case that the observed effects are not due to off-target disruptions in antigen presentation or transcription. However, impacts on synapse organization or vesicle trafficking cannot be fully ruled out, and authors should acknowledge this in the discussion to strengthen the interpretation.
- PCM recruitment in co-cultured vs. non-co-cultured DCs: Were pericentrin and CDK5RAP2 levels enhanced in conjugated DCs with extra centrioles (as they were in non-conjugated DCs), or was that only the case for gamma-tubulin? In non-conjugated cells, three different PCM proteins (pericentrin, gamma-tubulin, and CDK5RAP2) were more highly expressed in DCs with extra centrioles. In line 197, it is stated that "enhanced PCM recruitment" was also observed in conjugated DCs with extra centrioles. However, only gamma-tubulin levels are presented for conjugated DCs. This should be explained better.
- Additionally, the enhancement in gamma-tubulin appears to be slightly more pronounced in non-conjugated DCs (Fig. 3d) compared to conjugated DCs (Fig. 3f), which raises the question of whether similar trends might be observed for the other two PCM proteins of interest. The authors' assertion that enhanced PCM recruitment supports MTOC function hinges mainly upon the role of gamma-tubulin in promoting MT nucleation, so it is understandable why pericentrin and CDK5RAP2 results may have been omitted for concision, or perhaps not collected in favor of focusing on more critical experiments for this revision. (If the latter is true, I do not think it is critical to perform additional experiments). However, if the other two PCM proteins were indeed quantified in conjugated DCs and did not show the same enhancement, it would imply that enhanced

PCM recruitment occurs to the same extent in the context of activation, and OVAp loading and/or co-culture conditions somehow affected PCM recruitment. A brief explanation would be useful.

Minor notes:

- Line 160: Typo – change "extend" to "extent"
- Line 420: Typo – change "nucleaeted" to "nucleated"
- Line 599: Typo – change "immunogeniecy" to "immunogenicity"

Reviewer #2

(Remarks to the Author)

The authors have addressed all the concerns I raised. I recommend acceptance of the revised manuscript.

Reviewer #3

(Remarks to the Author)

In the revised manuscript, NCOMMS-24-49110, Stotzel et al. have made efforts to revise the manuscript, including the addition of in vivo experiments. Some of the concerns raised in the previous version remain insufficiently addressed. The current version of the manuscript remains quite densely written and difficult to follow. The reviewer found the mathematical modeling to be a great strength of the paper.

Major comments:

1. In Fig. 1 b, what is the time period at which the quantification was performed? Additionally, it is unclear why the zero centrin population exists in centrinone-treated cells.
2. The authors are encouraged to provide high-resolution images (Fig. 2a) for each condition.
3. In Fig. 2c, the reviewer is unclear about how the MT can have low straightness with decreasing MT length. With a known high persistence length, one would expect microtubules of short lengths to be straighter.
4. The reviewer is confused about why the pre-tubulysin data in the revised Fig. 2b is reduced compared to the original submission.
5. In Fig. 2e, can the authors expand upon why the OX40+ data are missing in the pre-tubulysin data?
6. One general concern is the quality of immunostained images. For example, it is difficult to distinguish between the number of centrioles. The authors are encouraged to provide high-resolution images that clearly show the distinction between the number of centrioles. The reviewer appreciates the challenges in imaging these structures.
7. In Fig. 3E, the γ -tubulin staining in the left ROI images for 'two centrioles' doesn't appear to correspond accurately with the merged image.
8. The authors have not included CETN2-GFP data in Fig. 4C of the revised manuscript, despite a prior request. Furthermore, the authors should improve the quality of Fig. 6h.
9. The authors should provide images in Fig. 4c, where the number of T cells interacting with the DC with and without OVAp is similar, and the associated analysis of the reorientation of MTOC in T cells.
10. The reviewer is puzzled by the size difference between T cells and DCs, as shown in Fig. 6b. Aren't the DC cells supposed to be larger than the T cells, or is this an isolated case?
11. It is encouraged to expand on the novelty of the work, while also acknowledging its shortcomings.

Reviewer #4

(Remarks to the Author)

Reviewer #5

(Remarks to the Author)

I co-reviewed this manuscript with one of the reviewers who provided the listed reports. This is part of the Nature Communications initiative to facilitate training in peer review and to provide appropriate recognition for Early Career

Researchers who co-review manuscripts.

Version 2:

Reviewer comments:

Reviewer #3

(Remarks to the Author)

The authors have, for the most part, addressed the issues. They might want to look into the following minor comment:

The authors propose (comment 1) that cell division may be facilitated by acentriolar microtubule-organizing center (MTOC) formation in cells lacking centrioles (Fig. 1e), as previously demonstrated (Wong et al. 2015; <https://doi.org/10.1126/science.aaa5111>). In Figure 1E, the authors do not present evidence of cell division in acentriolar cells. Furthermore, if the authors are citing Wang et al. (2015) in Science, the cited authors did not provide evidence of acentriolar division in immune cells. Moreover, normal cells enter a quiescent phase rather than dividing.

Point-by-point response on **NCOMMS-24-49110**:
‘Multiple clustered centrosomes in antigen-presenting cells foster T cell activation without MTOC polarization’

We like to thank the reviewers for the time and effort dedicated to evaluating our manuscript entitled *‘Multiple clustered centrosomes in antigen-presenting cells foster T cell activation without MTOC polarization’*. We greatly appreciate the insightful comments and the opportunity to clarify and further support our findings. We believe that this constructive criticism greatly improved our initial manuscript.

We understood that one major concern raised by two of the expert reviewers was the physiological relevance of our findings and the differences between 2D and 3D microenvironments that could affect cellular behavior and centriole/centrosome configuration. We agree that this is an important point and a major weakness of our initial manuscript. We carefully addressed this issue by conducting adoptive transfer experiments that allowed us to image DC-T cell interactions within lymph nodes. These experiments mimic more precisely the physiological environment of cells and represent a major advancement of our current manuscript.

Moreover, we restructured our revised manuscript (in particular the introduction and discussion section) to make it more concise and to focus on the major findings of our studies. In this context, we also slightly modified the title to *‘Multiple clustered centrioles in antigen-presenting cells foster T cell activation without centrosome polarization’* to better distinguish between centrioles and centrosome (as well as their function) as recommended by reviewer 2. Please find below our detailed point-to-point response to the specific comments and concerns raised by the reviewer experts.

Response to Reviewer 1

Reviewer #1 (Comments to the Authors):

The manuscript “Multiple clustered centrosomes in antigen-presenting cells foster T cell activation without MTOC polarization” examines how centrosomal organization in dendritic cells contributes to immune synapse formation and activation of T cells. The authors combine imaging, cell biological manipulations and mathematical modeling to study how centrosome number, integrity and spatial positioning influence efficient T cell activation. Overall, the manuscript is well presented and the conclusions appear reasonable.

We thank the reviewer for the thorough evaluation of our work and the insightful comments. We greatly appreciate her/his expertise in the field and hope that we can sufficiently respond to the major (and minor) concerns raised.

Some specific comments for the presented results

Figure 1:

Authors should specify whether the images are single-plane or maximum intensity projections. If single-plane images were used, the authors should justify how the imaging plane was selected.

In most cases, we show maximum z-projections. We have now clearly stated this in the figure legends and/or in the 'Materials and Methods' section (e.g., figure legend of Fig. 1b, p. 36, line 1197 and image analysis section p. 22-23, lines 785-807). Moreover, we extended our image analysis section to explain in more detail how we analyzed the different parameters and show representative images for clarification (when helpful) (e.g., Fig. 1c, Supplementary Fig. 2c). We hope that such detailed explanations of image analysis will improve data reproducibility.

B: (Right) Quantification of centriole numbers after Centrinone treatment: It is unclear how the authors define centriole detection through intensity thresholds. A description of how centrosomes were identified based on fluorescence intensity should be included.

We apologize for the confusion. We did not count centrioles based on fluorescence intensity thresholds after Centrinone treatment. Instead, centriole numbers were determined according to the number of CETN2-GFP⁺ foci that co-localize with a PCM marker such as γ -tubulin, CDK5RAP2 or pericentrin. In Fig. 1c we used signal intensities when quantifying γ -tubulin levels. We now added a representative image and marked the respective ROIs in which we determined integrated density levels of γ -tubulin signals. In addition, we describe more precisely how we quantified γ -tubulin levels at the centrosome in the 'Material and Methods' section (p. 22, lines 786-791).

Also, what was the rationale for testing two Centrinone concentrations (250 nM, 500 nM)? Was there a reason why certain panels only report results for one of the two concentrations? For instance, **Panel C**: Plotted results for 3 conditions (control, 250 nM, 500 nM Centrinone), but only showed montages of 2 conditions (control & 250 nM Centrinone). Perhaps this was done because there was no significant difference between the 250 nM and 500 nM conditions? This is not apparent from the presented results, since the effect of the two concentrations are not compared to each other in the beeswarm plot.

The initial study carried out by Wong et al (Wong et al, 2015; PMID: 25931445) used 125 nM and 300 nM Centrinone on cancer cell lines. Therefore, we started to use concentrations ranging from 125 nM (data not shown) to 250 nM. We were surprised that we found perfect MTOCs in the presence of Centrinone. To make sure, that we really target centrioles, we used a higher concentration and decided to double the amount (500 nM), which is still in the range of what has been published before. We included exemplary pictures for 250 nM and 500 nM Centrinone-treatment (Fig. 1c) and tested for statistical significance in the beeswarm plot as suggested by the reviewer. Indeed, the γ -tubulin intensity is even significantly decreased in the 500 nM Centrinone condition in comparison to 250 nM Centrinone

E: Quantification of MTOCs (left) and MT filaments emanating from defined regions around centrosomes (right) in mature CETN2-GFP expressing BMDCs after Centrinone treatment
Quantification of MTOCs and microtubule filaments is not clearly described. The authors should provide details on the image analysis methods used, including how regions were selected and how filament numbers were counted.

*We apologize for not being precise here. We quantified MT numbers in defined regions around the centrosome (CETN2-GFP⁺ foci surrounded by PCM makers [either γ -tubulin/CDK5RAP2/PCNT]). Whole z-stack planes were used to identify individual filaments. In the revised version of the manuscript, we now show an exemplary picture to illustrate this better (**Supplementary Fig. 2c**). Additionally, we included the respective information in the 'Materials and Methods' section (p. 22, lines 792-797).*

Perhaps the authors could elaborate on how the gamma-tubulin intensity was normalized? It could just be my lack of understanding, but I found this a bit vague. For instance, lines 1054-1055: "Graph shows normalized values relative to cells with two centrioles" but does not elaborate on the normalization procedure. Perhaps the authors wish to account for the fact that all three treatments yield a mixed population of cells, all with different numbers of centrioles (as shown in panel B) but this should be specified.

We completely agree with the reviewer that we want to account for the fact that we are analyzing a mixed population of cells. Therefore, we calculated mean values of γ -tubulin intensities of all cells with two centrioles. Next, we normalized/divided the intensities of all cells (independent on their centriole number) to this mean. The respective information can be found in the figure legend (p. 36, line 1207).

F: (Left) Quantification of proliferating T cells after Centrinone treatment according to CFSE labeling. There seems to be a clear interaction effect between the centrinone concentration and OVA loading (high OVA loading reduces the difference between the untreated and centrinone treated conditions for T cell proliferation. The interaction terms in the ANOVA should be examined more carefully to delineate how enhanced loading can overcome lack of centrioles.

*As far as we interpret our data, T cell proliferation drops in the 0.1 μ g/mL OVAp condition from the ctrl **74.38% \pm 17.79%** to **62.04% \pm 16.98%** after 250 nM Centrinone treatment and to 61.99% \pm 18.95% after 500 nM Centrinone treatment (please see also table below for better overview of all numbers). In the 1 μ g/mL OVAp condition the exact values are in the ctrl **80.06% \pm 14.79%**, after 250 nM Centrinone treatment **66.22% \pm 25.36%** and after 500 nM Centrinone treatment **64.35% \pm 26.51%**. The absolute reduction in proliferation is in the same range with both concentrations of OVAp (ctrl vs. 250 nM or ctrl vs. 500 nM) as the values at 1 μ g/mL OVAp loading condition display partially higher standard deviation the statistical significance is reduced. We feel that we cannot clearly answer whether enhanced OVAp loading can overcome lack of centrioles. This would require further investigations.*

Values [%]	ctrl		250 nM Centrinone		Drop between ctrl vs 250 nM Centrinone		500 nM Centrinone		Drop between ctrl vs 500 nM Centrinone	
	Mean	SD	Mean	SD	Absolute values	Relative values	Mean	SD	Absolute values	Relative values
w/o OVAp	2,93	1,80	5,70	6,25			7,54	6,40		
0.1 μ g/mL OVAp	74,38	17,79	62,04	16,98	12.34	16.59	61,99	18,95	12.39	16.65

1 $\mu\text{g/mL}$ OVAp	80,06	14,79	66,22	25,36	13.84	17.28	64,35	26,51	15.71	19.62
-------	-------	-------	-------	-------	-------	-------	-------	-------	-------

The meaning of the dotted box and the axis units in panel F are unclear and should be clarified to improve the reader's understanding of the data presentation.

*The dotted box in former Fig. 1F represents the gate that we analyzed. We excluded the histograms from the revised Figure 1f to make the figures less crowded (as suggested by reviewer 3). The gating strategy is now shown in **Supplementary Fig. 1d** to clearly demonstrate which gates were analyzed.*

Figure 2:

F: Quantification of T cell proliferation after co-culture with pretubulysin-treated BMDCs

Line 155: "T cell activation was markedly reduced in the presence of pretubulysin (Fig. 2f)"

In line 155, the authors seem to equate T cell activation with CFSE dilution in Figure 2f, which measures proliferation. It is unclear that CFSE dilution can be used as a proxy for T cell activation, as proliferation and activation are distinct processes. Ideally, they should present data on early activation markers to strengthen their argument that pretubulysin impacts activation, rather than just proliferation.

*We thank the reviewer for this comment. In the revised manuscript we distinguish more carefully between T cell proliferation and activation. Briefly, we measured IL-2 levels (Smith & Kendall, 1988; PMID: 3131876) in our DC-T cell co-cultures and OX40 upregulation on T cells (Gramaglia et al, 1998; PMID: 9862675) as two prominent T cell activation markers (**Fig. 2e**; please see also comment by Reviewer 2).*

The authors state that DC MTs had "not fully recovered" by 24h after tubulysin wash-out. Is there any quantitative data to support this statement? The right-hand plot in Figure 2C does provide evidence that the tubulysin wash-out worked, but it's unclear what time point this plot represents. The authors should mention how much time passed between wash-out and quantification of MTs (Fig. 2C).

*The quantification of MT architecture was carried out 24 h after drug wash-out. We included this information directly in the figure panel (**Fig. 2a**) and the figure legend (**for Fig. 2b and c; p. 37, lines 1234 and 1236**). Moreover, we included the parameters 'MT length' and 'MT straightness' in our revised version of the manuscript to quantify MT recovery (**Fig. 2c**). 'MT straightness' is defined as Euclidean distance of start to end point of one MT divided by whole MT length. This is explained in a scheme above the graphs. MT length and straightness are significantly reduced after 24 h of pretubulysin wash-out indicating that the drug acts to some extent irreversibly. This further allows us to study the role of MTs specifically in DCs. We included these results in the main text of the manuscript (**p. 5, lines 153-161**).*

Is there any supplementary data that could show MT recovery at different time points? This point may be a bit nit-picky, but since the long duration of MT destabilization is what justified the authors' use of a wash-out prior to co-culture experiments (which is preferable because it allowed them to co-culture treated APCs and T cells without exposing T cells to the treatment).

The quantification of MT architecture was carried out after 24 h of wash-out as this was also the time point that we used for assessing T cell activation (Fig. 2e). MT length and straightness are significantly reduced after 24 h of pretubulysin wash-out. Additionally, beforehand we carried out a time course experiment to determine MT recovery, which we did not include in the initial version of the manuscript (Fig. R1). After 6 h of pretubulysin wash-out, the number of MT filaments is even more drastically reduced than after 24 h of wash-out. This result indicates that pretubulysin treatment is - to some extent - reversible, but does not lead to a full recovery of MT architecture after 24 h, which we quantified by additional parameters such as MT length and straightness (Fig. 2c) in the revised version of the manuscript. We did not include the 6 h timepoint in the manuscript as we felt that this rather leads to over-crowding of our figures. Nevertheless, if the reviewer believes that we should include this information we are more than happy to show Fig. R1 in the main manuscript.

Figure R1: Quantification of MT filaments after 6 h and 24 h of pretubulysin wash-out.

Figure 2C: Is there a reason why no data is available for the no wash-out condition? Perhaps filaments too extensively disrupted to reliably count filaments?

As the reviewer assumes correctly, we were hardly able to detect any MT filaments in the presence of 1 μ M pretubulysin, which is why we did not quantify parameters such as MT number, length or straightness under this condition.

Figure 3: (multiple centrioles are not needed for enhanced conjugate formation but are more effective?)

B/E: Immunostaining and quantification of MT filaments

The authors should clarify the method used to quantify microtubule filaments and explain how individual filaments were distinguished in crowded cellular regions. Additionally, details on the area or region where MT filaments were counted should be provided for reproducibility. What is the “defined area around the centrosome”?

We quantified MT numbers in a round-shaped region around the centrosome ($25\mu\text{m}^2$). Whole z-stack planes were used to identify individual filaments in MT dense regions. In **Supplementary Fig. 2c**, we now show an exemplary picture to illustrate this better. Additionally, we included the respective information in the ‘Materials and Methods’ section (p. 22, lines 792-797). We thank the reviewer for this important comment and generally extended the image analysis section to improve reproducibility.

Figure 4:

D: Quantification of MTOC polarization towards the IS in T cells after different timepoints of co-culture

How is it determined which T cells have a “reoriented centrosome”? Is the whole confocal stack used?

As pointed out by the reviewer, whole confocal stacks were used to determine the position of the T cell's centrosome. We included this information in the image analysis section of the 'Materials and Methods' part (p. 23, lines 799-801).

F. This is a bit confusing as the T cells express Nurr-GFP while the DCs express CEN2-GFP. How were the two signals differentiated to unambiguously identify T cells and DCs.

We thank the reviewer for pointing this out and carefully considered this issue. Importantly, in the T cell, GFP is expressed in the nucleus (and the GFP signal was quantified in the nucleus – defined via DAPI staining), while in the DC there is background GFP present in the cytoplasm and additionally – even higher signal intensity – at the centrioles. Whole z-stack planes were used to identify DCs and T cells. Additionally, both cell types differ in size and shape. T cell areas for quantification were carefully chosen to avoid overlap with the DC. All overlapping cells, e.g. T cells lying on top of the DC, were excluded from the analysis.

Figure 5:

B/C: Quantifications of intracentrosomal and average distances in DCs with one and multiple centrosome(s) at different time points of co-culture

The authors mention using a manual tracking plug-in to calculate intracentrosomal and intercentrosomal distances. However, they do not provide detailed criteria for defining what constitutes the boundaries of individual centrosomes or how the centroid positions were selected, which could introduce user bias.

The manual tracking plugin provided by ImageJ was only used in live cell imaging videos to extract the coordinates of the centrioles (x,y). Based on these coordinates, the distances were calculated in excel using the formular: $distance = \sqrt{(x_2 - x_1)^2 + (y_2 - y_1)^2}$. The centroid of the DC was identified according to the boundaries of the CETN2-GFP signal (marking the DC membrane). We explain this in more detail in the 'Materials and Methods' section (p. 23, lines 798-807) and refer to the schemes in Fig. 5b,c.

Line 1171 (figure caption): “Each data point represents on cell derived from 5 independent experiments.”

Minor spelling error– “Each data point represents one cell”

We corrected the spelling mistake in the revised version of the manuscript.

D: Sketches and confocal CETN2-GFP images indicating distances between centrosome center point (CP) and T cell CP, and DC CP and T cell CP in cells with one (upper) and multiple (lower) centrosomes.

What is the quantitative criteria for defining center points of the centrosome/multiple centrosomes, T cell, and dendritic cell? Also unclear whether centrosomal distance was computed in 3D from confocal stacks or from the maximum intensity projection.

For DCs and T cells, center points were measured in ImageJ using the centroid measurement after marking the DC boundaries with the 'freehand selection' tool on maximum z-projection images. The centroid displays the geometric center point without considering any signal

intensities. The centriole center point coordinates were calculated with the formulas: $x_{CP} = \frac{\sum x_n}{n}$ and $y_{CP} = \frac{\sum y_n}{n}$ with n : centriole number. We included this in the image analysis section of the manuscript (p. 23, lines 798-807).

• **Lines 262-264:** This is a minor detail, but I wonder whether the phrasing of this assertion makes sense:

"... average distances in cells with multiple centrosomes [...] did not show prominent differences between OVAp loaded and unloaded cells suggesting that multiple centrosomes congregate together and cluster during antigen-specific DC-T cell contacts."

- Does it make sense to assert that centrosomes actively congregated together, given that, to my understanding: (1) they did not actually observe this movement -- only the end result -- and (2) no clear differences were noted in the presence or absence of OVAp? If not, perhaps an alternate description could be "centrosomes remained clustered together..."
- Then again, perhaps the final centrosome position provides sufficient evidence to assert that clustering must have occurred, if clustering is not the dominant morphology in non-conjugated DCs.

We agree with the reviewer that our formulation is misleading and changed it to '...centrioles remain clustered...' (p. 8, line 288) as suggested by the reviewer.

Figure 6:

The confinement of DCs and T cells under an agarose block may mechanically restrict their movement and potentially alter the physiological dynamics of centrosome reorientation and immune synapse formation. The authors should comment on how/if this could impact centrosome configurations.

*We absolutely agree with the reviewer that the microenvironment and the level of confinement might have a strong impact on cellular behaviour and the physiological dynamics of centrioles/centrosomes. To more precisely mimic the conditions cells are facing in tissues, we performed adoptive transfer experiments which are presented in **Figure 6**. Briefly, we injected OT-II T cells intravenously into WT mice and after 24 h CETN2-GFP expressing BMDCs into the hock. After 48 h, popliteal and inguinal lymph nodes were collected. Cryosections were stained with PCM markers such as CDK5RAP2 or pericentrin. 3D image analysis shows that also within lymph nodes, centrioles are clustered and localize near the cell centre in close proximity to the nucleus. We added these new results in the main text of our revised manuscript (p. 9, lines 311-325).*

Line 625 (methods): "...DC-T cell co-cultures were injected under a block of agarose as previously described"

The cited paper describes a different set-up that includes two glass surfaces spaced by PDMS pillars.

We thank the reviewer for this notification and apologize for the mistake. We removed the respective citation and corrected the paragraph to '...DC-T cell co-cultures were injected under a block of agarose to prevent cell floating during the imaging period' (p. 20, line 713).

Figure 7:

PARP enzymes play crucial roles in DNA repair, transcriptional regulation, and cellular metabolism. PARP inhibition could disrupt these processes in DCs, impairing antigen presentation or cytokine production, which would independently affect T cell activation. Is the observed reduction in T cell activation solely due to centrosome declustering, or might it also involve broader PARP-related cellular changes?

*The reviewer raises important points here. In the revised version of our manuscript, we carefully checked for potential side effects of PJ-34 treatment using different concentrations of PJ-34. We determined the capacity to present antigens by staining for cell surface MHCII levels (**Supplementary Fig. 7a,b**) and measured mRNA levels of DC-specific cytokines (Cxcl1, Il-6, Ccl5) in the presence or absence of PJ-34 (**Supplementary Fig. 7c,d**). All mRNA levels seem unaffected by PJ-34 treatment. Moreover, intracellular protein levels measured by intracellular cytokine staining showed no significant differences between the treatments. These results strongly suggest, that PJ-34 treatment for 3 h does not lead to large changes in cytokine levels or impaired antigen presentation in DCs.*

The authors should clarify why intracentrosomal distance was affected by PJ-34 treatment in non-cocultured cells (panel B), but not in co-cultured cells (panel D). There is one way I could see this being relevant: in order to confidently assert that activation is impaired upon declustering of multi-numerous centrosomes, it seems important to emphasize that the declustering agent only affects **inter**centrosomal distances, not intracentrosomal distances (and thus, the results arise due to declustering of centrosomes, rather than abnormal spacing between centrioles in each individual centrosome). Still, as long as the authors can show that intracentrosomal distance was not affected in the context of the coculture experiment, I think they should be able to defend their assertion.

We agree with the reviewer that the observed phenotype on T cell activation in the presence of PJ-34 can only be assigned to de-clustering of centrioles, when the linker between the two centrioles (reflected as intracentrosomal distances) is intact. As pointed out by the reviewer, intracentrosomal distances in DC-T cell co-culture experiments were largely unaffected by PJ-34 treatment, which is why we feel confident about our conclusion that predominantly de-clustering of multiple centrioles causes the observed phenotype. We cannot really explain the difference between co-cultured and single cells.

E: The authors should clarify whether these results drawn only from 2N cells with multiple centrosomes, or was data collected from all 2N cells (single and multiple centrosomes)?

Lines 303-306: *“Interestingly, when co-culturing with either PJ-34-treated or control OVAp-loaded DCs with OT-II T cells, we found that **in the presence of de-clustered centrosomes**, T cell activation was significantly diminished compared to control cells.”*

This is relevant because when centrosomes were counted in figure 3A, it seemed as though multicentrosome cells only comprised ~25% of all 2N cells (centriole number ≥ 3). Therefore, in a population of single- and multi-centrosome cells, I wonder whether any effects on multicentrosome cells would be prominent enough to yield the significant differences shown in Figure 7E. If not, perhaps there is some other mechanism at play by which PJ-34 could affect the ability of DCs to stimulate T cell proliferation?

Data from Figure 7e were collected on sorted 2N cells. We included this information in the respective figure legend. In addition to sorting on 2N cells, we also slightly enriched for cells with multiple centrioles leading to 30% of the cells containing >2 centrioles. This enrichment allows to monitor differences in the presence of de-clustering agents as more cells with multiple centrioles are present. Furthermore, we want to highlight that the PJ-34 treated cells show significantly increased MTOCs which is the functional effect of de-clustered centrioles and very likely to account for decreased T cell activation capacity by those DCs. In addition, we show in the revised version of the manuscript that we tested for potential side effects of PJ-34: antigen-presentation (MHCII surface expression) as well as cytokine production (Supplementary Figure 7a-d). We apologize for the confusion and now clearly state BMDC sorting in the figure legend (p. 42, line 1361).

Figure 8:

B: Plot of $\langle d_{short} \rangle$ vs h_{CS} in the presence of a nucleus located centrally to the cell

Line 340: "Our data indicate an optimal perinuclear positioning of the centrosome, marginally shifted from the nuclear surface by $< 1 \mu m$ "

A visual indicator of the minimum $\langle d_{short} \rangle$ on plot B would better emphasize the optimal positioning, as the x-axis scale makes it difficult to identify.

We thank the reviewer for this constructive suggestion. To improve clarity, we have added a red vertical dotted line on the x-axis in Fig. 8 b to indicate the optimal centrosome position - defined by its distance from the cell centre (h_{CS}) and from the nuclear surface (d_{CS-NS}) - that corresponds to the minimum value of $\langle d_{short} \rangle$ for a centrally located nucleus (p. 43, lines 1372-1373).

General comments: While the authors report that DCs with multiple centrosomes exhibit increased cytokine secretion, the referenced study lacks temporal data on when these cytokines are released relative to IS formation. To establish a direct link between MT contact and T cell activation, the authors should investigate the time-scale of cytokine release using real-time measurements. Alternatively, they should suggest this as a future direction to determine whether early MT contacts trigger cytokine exocytosis or if other mechanisms contribute to the observed increase in T cell activation.

We agree with the reviewer that real-time measurements of cytokine secretion and simultaneous T cell activation would be very informative and establish a more detailed link between elevated MT numbers and enhanced T cell activation. We believe that these experiments are technically very challenging and might go beyond the scope of this study. To directly link enhanced cytokine trafficking and/or secretion in the presence of multiple centrioles and increased MT nucleation with T cell activation, we would need a robust T cell activation marker, that can be detected instantaneously. Calcium signalling is fast, but whether it allows to detect differences on the time scale of seconds in cells with different numbers of centrioles is questionable. In parallel, we would need to monitor centrioles, MT filaments and fluorescent cytokines that are secreted. For all structures, we would need to record z-stacks over time, which means high photo stress and requires superfast imaging techniques. All experiments require single cell resolution as we are dealing with a mixed population of cells regarding centriole numbers. As suggested by the reviewer, we rephrased our claims on cytokine secretion and vesicle transport as missing mechanistic link between multiple

centrioles, increased numbers of MT filaments and enhanced T cell activation. Moreover, we included a sentence in the discussion where we propose future studies on this topic (p. 17, lines 602-605).

- Minor typos –

- Line 61-62: “Yet, B cells have a rather week T cell priming capacity” → “weak”
- Line 119: poly-L-lysin -> poly-L-lysine
- Line 150-151: “thus allowing to study the role of MTs...” → “thus allowing us to study...”
- Line 205: promotor -> promoter

We thank the reviewer for thoroughly reading the manuscript and corrected the mentioned typos in the revised version of the manuscript.

Response to Reviewer 2**Reviewer #2 (Comments to the Authors):**

In the manuscript entitled “Multiple clustered centrosomes in antigen-presenting cells foster T cell activation without MTOC polarization”, by Stötzel et al. explored how centrosome amplification in DCs affects immune synapse formation and T cell activation. They integrated experimental data with computational modeling to demonstrate that extra centrioles in DCs lead to the formation of overactive centrosomes. These centrosomes cluster during DC-T cell interactions and are positioned near the cell center. Disrupting either the number of centrioles or the centrosomes configuration results in impaired T cell activation.

The entire study is quite interesting, as it establishes the relationship between optimal centrosome positioning, centrosome numbers, and T cell activation.

However, the author seems to have a limited understanding of the concepts, distinctions, and general functions between centrosomes and centrioles, leading to some misinterpretations of experimental data. Furthermore, I believe that using computational modeling to explain centrosome positioning may be somewhat ambitious. A more comprehensive model should incorporate additional intracellular variables and factors. If most of the issues below are resolved, I recommend that this manuscript be accepted.

We thank the reviewer for her/his interest in our work and for the insightful comments. We understand the concerns raised, particularly on the distinction between ‘centrioles’ and ‘centrosomes’ and a specific part of our computational analysis. We recognize the importance of these points and addressed them by carefully rephrasing the respective passages in the revised version of the manuscript. Briefly, we now use the term ‘centrioles’ in the first part of our manuscript as we are counting CETN2-GFP⁺ foci that are part of the centriole (please see also our answer to your comment on Fig. 4G). Later, after showing the functionality of extra centrioles by pericentrin, γ -tubulin and CDK5RAP2 staining, we define the condition >2 centrioles as ‘over-active MTOCs’. When we talk about numbers, we use the term multiple or extra ‘centrioles’ instead of ‘centrosomes’ as we detect only one MTOC. Regarding our computational model please find our detailed response and extension of the model below.

Specific comments:

The author raises several detailed scientific questions in the Introduction and Results sections, but it appears that few of them are thoroughly addressed.

The Introduction section is somewhat redundant, and the presentation of background knowledge is a bit disorganized and lacks logical coherence. It is recommended to succinctly present the background information and clearly state the key scientific question.

*We thank the reviewer for this comment. We completely rephrased the introduction to make it more concise and logical, discarded redundant passages and elaborated in more detail on novel aspects of our work. Similarly, in the results section we anticipated to further strengthen our results on having a centrally localized centriole-cluster and included in vivo data on DC-T cell interactions within lymph nodes (**Fig. 6**). We hope that these new data address more thoroughly the importance of centriole configuration during DC-T cell interactions.*

Pericentrin is the most typical marker of the PCM; it would be best if the author included additional staining for it.

*We thank the reviewer for this important comment. We included pericentrin (PCNT) staining as an additional PCM marker in the revised version of our manuscript in **Figure 3**. We re-analyzed centriole numbers (CENT2-GFP⁺ foci co-localizing with PCNT staining) in diploid cells and show this in **Figure 3a**. Moreover, we quantified PCNT intensity around centrioles which showed increased PCNT levels at multiple centrioles in comparison to only two centrioles (**Fig. 3c**). In addition, we performed PCNT staining on cryosections of popliteal and inguinal lymph nodes (**Fig. 6**). This allowed us to counterstain centrioles in the injected CENT2-GFP expressing BMDCs while at the same time also the T cell's centrosome is marked.*

The assertion that "MT organization was not affected by the loss of centrioles" is a perspective held by only some researchers. If the author wishes to make this claim, they should cite 2-3 high-quality references.

*We agree with the reviewer that this statement is controversially discussed in the community and cannot be generalized. We now emphasize that we only refer to our own study, where we show that MT numbers per cell are not altered in Centrinone-treated vs. control cells (**Fig. 1e**). We rephrased the respective paragraph to '...demonstrating that under our assay conditions DCs lacking centrioles are able to nucleate and organize MT filaments to form a functional MTOC.' (p. 4-5, lines 146-148).*

I am not an expert in the field of immunology, but I still feel that using only one experimental method, flow cytometry (Fig. 2F), to support such an important conclusion "We found that T cell activation was markedly reduced in the presence of pretubulysin", is insufficient and not solid.

*As suggested by the reviewer, we included additional T cell activation measurements in our revised version of the manuscript such as IL-2 secretion (measured by ELISA; Smith & Kendall, 1988; PMID: 3131876) and OX40 upregulation on the T cell - a prominent activation marker (Gramaglia et al., 1998; PMID: 9862675) (**Figure 2e, p. 5, lines 164-171**). We hope that we can convince the reviewer, that these additional read-outs emphasize the importance of the MT cytoskeleton in DCs for T cell activation and strengthen our initial claims.*

For Figure 4G and its related data: Centrin-2 is primarily associated with centrioles and is a marker for centrioles. If the centrosome is concentrated in a single focus, it is defined as "one centrosome," even if it contains multiple centrioles. It is defined as "multiple centrosomes" only when there are multiple foci or if it appears dispersed. For identifying multiple centrosomes, PCNT (Pericentrin) should be used as the marker instead of Centrin-2.

We thank the reviewer for this important comment. We rephrased all sections and paid attention on the nomenclature of centrioles and centrosomes as outlined above. In addition, we included PCNT staining as a marker for the PCM and thereby assess precisely centriole vs. centrosome numbers. In most cases where more than two centrioles are present, only one clear pericentrin signal can be detected, which appears more intense compared to cells with

only two centrioles. Similar results were obtained for γ -tubulin and CDK5RAP2 staining. Moreover, MTs emanating from multiple clustered centrioles are significantly increased. Therefore, we refer to this condition as one over-active MTOC (Fig. 3a-g) (p. 6, lines 184-205).

I understand the novelty the author is aiming for in the final section of the Results, but explaining centrosome positioning using a geometric model seems somewhat far-fetched and unusual. If a computational model is to be used for this explanation, the evidence provided by the author is not sufficient. For instance, there are many interacting factors with the centrosome within the cell, including various organelles and variables such as fluidity. I believe this represents a highly complex computational process, unlike the current model in the paper.

We appreciate the reviewer's thoughtful critique and acknowledge that centrosome positioning is governed by a complex interplay of cellular components and mechanical forces - including interactions with organelles, cytoplasmic viscosity, cortical tension, and actomyosin contractility. However, our primary focus in this study was to understand how a specific centrosome location affects the efficiency of the microtubule-based search process for the synapse region important for T cell activation, rather than to investigate the mechanisms underlying centrosome positioning. Our simulations show that a centrally positioned centrosome (or centriole cluster) minimizes search time by dynamic MTs, thereby potentially enhancing activation of T cells attached to the dendritic cell membrane. Complementing this, our experiments highlight the critical role of centriole clustering and central positioning in antigen-presenting cells (APCs) for modulating T cell responses. To explore a possible physical basis for this centring, we introduced a simplified geometric model that considers MT-generated pushing forces from the cell cortex and nuclear surface. This minimal model yields a mechanically stable configuration near the nuclear periphery and close to the cell centre, offering a plausible explanation for centrosome positioning without invoking detailed cytoplasmic modelling.

We fully agree that centrosome positioning likely arises from multiple, overlapping mechanisms. Prior work has implicated MT pushing against the cell boundary (PMID: 9177199; PMID: 9346483; PMID: 30780383), cortical dynein pulling (PMID: 20980619; PMID: 22304918; <https://www.nature.com/articles/s41567-023-02223-z>), pushing forces due to MT-organelle interactions (PMID: 20980619), cytoplasmic dynein-mediated pulling via vesicles attachment (PMID: 21866267; PMID: 20980619), and actomyosin-driven centripetal flows (PMID: 20980619; PMID: 40243666). Our model suggests that MT-generated pushing forces play a leading role under specific conditions, promoting centrosome centring in dendritic cells, while an alternative cortical pulling-dominated regime could lead to peripheral positioning and even declustering of centrioles. While a detailed integration of these mechanisms is beyond the scope of this study, we now include a brief summary in the revised discussion section (p. 16-17, lines 583-590) and propose such extensions as a direction for future work.

Response to Reviewer 3**Reviewer #3 (Remarks to the Author):**

The authors have tried to understand how the MTOC (Microtubule organization center) organization in APC (antigen-presenting cell) has an effect on T-cell activation. It is well established that cellular polarization is essential for immune regulation, often involving centrosome reorientation. However, how centrosomes are oriented from the perspective of APCs is not entirely known. Here, the authors consider murine DCs (Dendritic cells) and observe, using experimental and mathematical models, that DCs' additional centrioles create over-active centrosomes that cluster near the cell center during DC-T cell interactions. Disrupting centriole count or centrosome positioning in DCs impairs T cell activation, emphasizing the critical role of centrosome amplification and positioning in APCs for effective T cell responses.

The manuscript details an important and often overlooked aspect of T cell activation. The complementary nature of the study (experiments supported with computational models) is commendable (please note that the reviewer is not an expert in computational modeling). While the enthusiasm for the topic is high, the reviewer is less excited about the manuscript in its present form. Mainly, the loss of enthusiasm stems from overcrowding of the figures (which makes it very challenging to follow and understand the novelty of work), low sample numbers, and a few irregularities noted below.

We are grateful that reviewer 3 describes our study as important and has notified the current lack of data on the DC side of the immune synapse. Moreover, we are happy that this reviewer explicitly acknowledges the interdisciplinary nature of our study, which we believe is a great strength of our work. We highly appreciate her/his thorough and insightful evaluation of our work.

We also understand the importance of addressing the issues raised regarding figure presentation and increasing N/n numbers, which we thoroughly addressed in the revised version of the manuscript. Please find below our detailed responses to the individual comments.

Major comments:

1. The figures are very dense, making them hard to follow. The figure captions are very lengthy, and some require a single page. The authors are encouraged to reconsider how they should present their data.

*We rearranged each individual figure and shifted parts of the main figures to the supplementary section (former Fig. 3f-h and former Fig. 6) to make the figures less crowded and thereby hopefully easier for the reader to capture the results. Large histogram panels of individual conditions were removed as exemplary gating strategies are shown in the respective supplementary figure panels (please see new **Fig. 1f and 2f** and **Supplementary Fig. 2a**). Moreover, we included a detailed description of image analysis in the 'Materials and Methods' section which further allowed us to shorten some figure legends without impacting the reader's understanding.*

2. Following up on the above comment, the authors should clearly highlight the novelty of their

work. Does the work provide a new interpretation of T cell activation, a new method, or a new computational model? How are the experiments and model tied together?

We thank the reviewer for this constructive comment. We modified the introduction section to highlight the novelty of our work and made the content more concise as also suggested by reviewer 2. Briefly, we aimed to center our research focus on the role of the centrosome and the MT cytoskeleton in antigen-presenting cells for efficient T cell activation. The process of T cell activation is key for the induction of an adaptive immune response and fundamental intracellular processes shape this cell-cell interaction. Our results highlight the crucial role of the MT cytoskeleton (including centrioles and MTs) for immune-related functions in interphase. We tackle the arising question on intracellular processes by using wet lab experiments and computational modelling, because we believe that this gives a more comprehensive understanding.

*We also aimed to tie our experiments and computational model closer together by extending the model for flattened cells to make it comparable to the 2D situation (confinement) during live cell imaging and vice versa, we analyzed centriole configuration in a 3D tissue microenvironment (**Fig. 6**).*

3. The reviewer was puzzled in interpreting the data in the following figures:

a. In Fig 1E (left graph), what is the percentage of cells with two centrioles and no centrioles in the presence of 500 nM Centrinone?

*We find around 60% of cells without any centriole and 12% of cells with two centrioles when treating cells with 500 nM Centrinone (quantified now in **Fig. 1b**). In total, >130 cells have been analyzed per condition. We included the number of cells analyzed in the figure legend. As described by Wong et al, 2015, Centrinone-treatment does not ‘destroy’ existing centrioles but prevents the formation of new daughter centrioles. We treated our cells directly after isolating the bone marrow, when we started to differentiate into the DC lineage. The cells undergo roughly 6-7 rounds of division until we stimulate them with LPS. This implies that the cells can proliferate even in the absence of centrioles (most likely by forming acentriolar MTOCs). This is a very interesting phenomenon, that we did not comment on further as we are not focusing on the role of centrioles during cell division.*

b. For Fig 2, the low sample size from only two experiments raises doubts about the claims of no changes in MTOC.

*We repeated the experiments for DC MTOC and MT analysis in the presence of pretubulysin to increase sample sizes (former Fig. 2C and 2E; now presented in **Fig. 2b, c**). In total, we analyzed N = 33 (ctrl) /34 (pretubulysin wash-out) cells (left: MT number); N = 39 (ctrl) /43 (pretubulysin wash-out) cells (right: MTOC number) derived from n = 3 independent experiments. Cells were differentiated from three different mice. We included all n/N numbers in the figure legend according to the Nature Communications policy on samples sizes (e.g.: Figure legend Figure 2b,c, **p. 37, lines 1234-1241**).*

In line with this, the analysis of T cell MT and MTOC numbers was extended as well. Now showing in Supplementary Figure 2e four independent experiments with total cells analyzed: N = 93 (ctrl) /183 (pretubulysin wash-out) /60 (pretubulysin w/o wash-out) (left: MT number); N = 103 (ctrl) /184 (pretubulysin wash-out) /61 (pretubulysin w/o wash-out) (right: MTOC number).

c. The authors must provide high-resolution magnified images (insets and similar to image quality in subsequent figures) and supporting extra images in the supplementary materials (for all figures). From Figure 2b, the MTs look intact in the representative image w/o pre-tubulysin washout and with washout, which is contrary to the claims made by the authors in Line 149.

*We thank the reviewer for this comment. In the revised version of the manuscript, we now provide high-resolution images for all figures either in the main part or in the supplementary information. For former Fig. 2B (now **Fig. 2a**) we show α -tubulin staining separately from the merged image, which will hopefully make the differences in MT architecture more visible. We also included additional parameters to assess MT structure such as length and straightness (**Fig. 2c**) to demonstrate that after pretubulysin wash-out not only MT numbers are affected but also MT structure.*

4. Following up on comment 2b, the number of MT filaments shows high variability in control experiments (Fig 1F and 2C). These numbers should be the same or similar. How are the authors defining an MT filament? Does it have a minimum length?

*We thank the reviewer for this comment. In former Fig 1F and 2C MT numbers have been analyzed by different people in the lab, which is our explanation why MT numbers vary between different experiments. We re-analyzed all pictures/experiments for our pretubulysin experiment by one single person and used a precise ROI with a defined area (25 μm^2) to count MT filaments (**Supplementary Fig. 2c**). In **Fig. 2c** also MT length is determined after pretubulysin wash-out.*

5. The authors mentioned that tubulin intensity is higher during IS (Immune synapse) formation in the presence of two or more centrosomes (Figure 3b). However, the provided images do not clearly show that IS formation corresponds to the centrosome number. Hence, the authors should provide magnified microscopic images of the same.

*We apologize for this confusion. We now show an exemplary image how we quantified γ -tubulin intensities (**Fig. 1c**). Our data demonstrate that γ -tubulin intensity (and also PCNT and CDK5RAP2 intensity, **Fig. 3c,d** and **Supplementary Fig. 3a**) is higher in cells with >2 centrioles. We also added magnified microscopic images of cells bearing 2, 4 or more than 4 centrioles. The increased γ -tubulin intensity in cells with >2 centrioles could be shown in DCs without T cells and during IS formation.*

*A separate finding from our study shows that the formation of contacts between DCs and T cells does not correspond to the centriole number as shown in Supplementary Fig 3d,e. However, T cell activation, which was measured by Nur77-GFP upregulation (**Fig. 4b, e**), correlates with centriole numbers. We hope we could explain our results more precisely and also re-ordered this figure for a better understanding. In addition, we added a more detailed description of our image analysis section (**p. 22-23, lines 785-807**) and representative images of image analysis in **Fig. 1c**.*

6. In Fig 4d (graph), the authors have plotted the MTOC reorientation during the co-culture at different time points. Authors must provide representative microscopic images of the centrosome alignment at other time points (4, 6, 20h) in their supporting information.

*We included representative images of MTOC reorientation in T cells at 4, 6, 20 h after co-culture (now **Fig. 4c**).*

7. Since Figures 3 and 6 are inconclusive findings, the authors are encouraged to move them to supplementary materials.

*As suggested by the reviewer, we shifted former Fig. 3F-H (now **Supplementary Fig. 3c-e**) and former Fig. 6 (now **Supplementary Fig. 6**) to the supplementary information.*

8. Similar to the irregularity mentioned in comment 3, in Figure 3F, the relative frequency of mono conjugation between T cell and DC cells does not reflect the centrosome numbers in Figure 3G in the same condition.

*We apologize for the confusion. In former figure 3F (now **Supplementary Fig. 3c**) we do not distinguish between cells with two or multiple centrioles. Just the overall frequency distribution of mono- and multi-conjugated cells is shown depending whether DCs were loaded with OVAp or not. The data from former figure 3F+3G results from the same analysis (now **Supplementary Fig. 3c,d**). Data is displayed as frequencies from the respective condition e.g., Suppl. Fig. 3d left: from all the contacts formed in the w/o OVAp condition, Suppl. Fig. 3d right: from all the contacts formed in the + OVAp condition. Please find all details on cell numbers and frequencies in the figure legend (**Supplementary information p.10, lines 158-167**) and in the source sheet.*

9. In Figure 4C, It is unclear how the GFP values are observed to be below the threshold in 20h coculture. What does this signify? The reviewer is also confused on how the authors are claiming that multiple centrosomes are more efficient in activating T cells than with only one centrosome (lines 246-247), when there is no data provided for the latter.

*In previous Fig. 4C (now **Fig. 4b**) we quantified T cell activation via Nur77 expression by flow cytometry after 6 h and 20 h of co-culture. We distinguished between co-culture with CETN2-GFP^{high} cells (enriched for >2 centrioles) and with CETN2-GFP^{low} cells (enriched for two centrioles). After co-culture, the frequency of Nur77⁺ T cells of all CD4⁺ T cells was assessed according to the gating strategy shown in Supplementary figure 4a. We normalized data to the percentage of activated cells in the CETN2-GFP^{low} cell-condition. Thus, values below one indicate less T cell activation by DCs enriched for cells with multiple centrioles. To make this clearer, we changed the respective figure legend (**p. 39, lines 1287-1288**).*

10. In Figures 5b and 5e, the authors have provided intercentrosomal and intracentrosomal distances for 1, 2, and 4h of co-culture. Why did the authors not included the data for 6h and 20h where the maximum T cell activation was observed (Line 215 and Figure 4b)?

*We thank the reviewer for this comment and repeated the respective experiments. We now included 6 h and 20 h data points in our analysis (**Fig. 5b-c and 5e,f and Supplementary Fig. 5a-d and 5f**). We initially chose earlier timepoints, as we expected IS formation and MTOC reorientation to occur before Nur77-GFP gene expression. In summary, we did not detect changes in intercentrosomal and intracentrosomal distances at all time points tested and therefore concluded, that the DC MTOC stays at the cell center. To further elaborate on this*

point, we now also included in vivo data to strengthen our claim in physiologically more relevant environments (Fig. 6).

11. In Fig 6A and 6C (graphs), the mean values can be misleading as the individual data points are highly inconsistent. The authors should reconsider increasing the 'N' values to determine the distances.

*We increased the number of cells (N) for the respective experiment and included the dataset in the revised version of the manuscript (now **Supplementary Fig. 6a and c**). The following number of cells were analyzed: 6a: N = 17/13/24/13 (left to right) and 6c (left): N = 17/14/57/34 (left to right). Overall, we do not observe changes in centrosome position within the first hour of DC-T cell contact.*

12. In Figure 7b, c: Why did the authors use two different concentrations of PJ34 to study α -tubulin and centrosomal distance?

*We initially used different concentrations of PJ-34 to test the drug effect as well as potential side-effects (as also pointed out by reviewer 1). Ideally, we aim to perturb only intercentrosomal distances and leave the centrosomal linker intact. Also, we needed to make sure that MT stability was not affected by PJ-34, which is why we counted MT filaments in the presence of 50 μ M PJ-34. Now, we included data using 100 μ M PJ-34 in the revised version of the manuscript and analyzed distances and MT numbers (**Fig. 7c**) as we agree with the reviewer that this is more consistent with the remaining experiments in Figure 7.*

13. The reviewer is not an expert in computational modeling. However, the conditions outlined in the model for DCs (spherical cells) do not match the shape and culture conditions in experiments (2D). How does this impact the findings in Figures 8 and 9 and, generally, the force balance model with MT sliding?

*We thank the reviewer for this observation. Indeed, confined dendritic cells under agarose are largely flattened rather than spherical. In the previous version of the manuscript, we showed that even in flattened cells, the simulations yield similar optimal positioning of the centrosome as in spherical cells, minimizing the time required for dendritic cell microtubules to locate the immune synapse (former Supplementary Fig. 5I-K). We found that optimal positioning occurs near the nuclear periphery when the nucleus is concentric with the cell ($d_{offset} = 0$), and near the cell center, close to the nuclear surface, when the nucleus is moderately off-centered ($d_{offset} = 6 \mu\text{m}$, with the cell center just outside the nuclear boundary). In the revised version, we have extended this analysis to include a more significantly off-centered nucleus ($d_{offset} = 12 \mu\text{m}$), along with a more strongly flattened cell geometry. To ensure unrestricted microtubule growth along nuclear periphery, we flattened the cell such that the nuclear envelope does not contact the cell boundary. We also examined the effects of multi-centriolar clustering under these flattened conditions and compared the outcomes with those from spherical geometries. Interestingly, we found that the optimization trends in centrosome positioning and centriole clustering remain consistent across both geometries (**Supplementary Fig. 9**).*

Additionally, regarding the force balance model, we clarify that it does not incorporate forces

*arising from microtubule (MT) sliding. Instead, we focus on MT-generated pushing forces: (i) from MTs growing against the cell periphery, and (ii) from a subset of MTs exerting force on the nuclear surface, while the remaining MTs are assumed to glide along the nucleus without contributing to the overall forces acting on the centrosome. This assumption of MTs gliding is motivated by our frequent experimental observation of MTs residing in the 'geometric shadow' behind the nucleus - regions inaccessible to straight MTs extending directly from the centrosome. Our simplified model thus identifies a regime where MT-driven pushing against both the cell boundary and the nucleus provides a plausible mechanism for centrosome centering in dendritic cells. Such pushing-mediated centrosome positioning near cell centre has been highlighted in earlier experiments (PMID: 33027648; PMID: 9177199; PMID: 33026931; PMID: 38426416; PMID: 22499806) and supported by several theoretical studies across different 2D and 3D cell geometries, with or without a nucleus (PMID: 28960439; PMID: 30780383; PMID: 27440925; Pavin, Nenad, et al. "Positioning of microtubule organizing centres by cortical pushing and pulling forces." *New Journal of Physics* 14.10 (2012): 105025; <https://link.springer.com/article/10.1007/s12648-022-02309-z>) - underscoring the applicability of our minimal framework in distinct conditions.*

14. A general question (and a perceived weakness of the manuscript) is how flat 2D environments recover in vivo behavior. How would the accumulation of T cells change in more physiologically relevant environments?

*We thank the reviewer for this important aspect. As pointed out also by reviewer 1, the microenvironment and the level of confinement might have a strong impact on cellular behaviour and the dynamics of centrioles/centrosomes. To more precisely mimic the physiological conditions cells are facing in tissues, we carried out adoptive transfer experiments within the OT-II context and visualized centrioles/centrosomes in DC-T cell contacts directly within lymph nodes. We found that also within lymph nodes, centrioles in DCs are clustered and localize at the cell centre in close proximity to the nucleus (**Fig. 6**).*

Minor comments:

The reviewer has some minor questions (mostly clarifications) on data organization.
1) Is the 500 nM Centrinone effect on depleting the 2N cells % statistically significant in supp fig 1E?

We tested for statistical significance between ctrl and 250 or 500 nM Centrinone in Suppl. Fig. 1E (now Supplementary Fig. 2b). Indeed, there are significantly less 2N cells in the 500 nM Centrinone condition compared to control cells. This is not surprising as Centrinone inhibits PLK4 – a master regulator of centriole duplication (Bettencourt-Dias et al, 2005, PMID: 16326102; Habedanck et al, 2005, PMID: 16244668) which is required for faithful chromosome segregation. However, as we sorted each treatment condition for diploid cells and performed all subsequent experiments with diploid cells, we believe that this does not impact our findings.

2) The authors should mention the different colors in the figure legends, especially for Figure 2a.

*We now included figure legends for Fig. 1B (now **Fig. 1a**), 1F (now **Fig. 1f**), Fig. 2A (now **Fig. 2d**), Fig. 7D (now **Fig. 7d**) to better define the colored structures.*

3) Line 62: Please correct the typo in "... week T cell priming...?"

We corrected 'week' to 'weak'.

4) clarify the Fig 4D legend: "...each datapoint is one individual experiment, whereas N> 100 cells analyzed per condition". What does this mean?

In Fig. 4d each datapoints represents one independent experiment in which cells from different mice were used. In each experiment >100 cells were analyzed to calculate the mean Nur77^{GFP} intensity.

5) In Figures 3c and d, the Y-axis should be the same for easy interpretation of data.

As suggested, we modified all Y-axes in Fig. 3 (now Fig. 3c,d,f) to improve data presentation and understanding.

6) The authors should clarify the sample size information in Figure 4G- the authors state that analyzed cells are greater than 100 (line 1156), yet the data point shows only ~4.

In former Fig. 4G (now Fig. 4e) each datapoint represents one independent experiment (n=3-4). For calculating the percentage of activated T cells, >100 cells were analyzed for each experiment.

7) The numbering of figures in the text is out of order.

We carefully checked the numbering of all figures in the revised version of the manuscript.

8) Authors should modify the Y-axis label of Figure 5e. indicating the centrosome numbers (as per the figure legend).

We modified the layout of Fig. 5d,e and f by indicating centriole numbers. In addition, to improve data presentation, we changed the Y-axis label as suggested by the reviewer.

Reviewer #4 (Remarks to the Author):

Reviewer #5 (Remarks to the Author):

We thank reviewer 4 and 5 (and also Nature Communications) for giving early career researcher the change to get the credit for their reviewing activities. Our manuscript greatly benefited from this thorough reviewing process.

Point-by-point response on **NCOMMS-24-49110A:**
'Multiple clustered centrioles in antigen-presenting cells foster T cell activation without centrosome polarization'

We thank the reviewers for the time and effort invested in evaluating our revised manuscript entitled '*Multiple clustered centrioles in antigen-presenting cells foster T cell activation without centrosome polarization*'. We greatly appreciate the additional comments and the opportunity to clarify and improve our work.

Following reviewer's 1 recommendation, we have adjusted the manuscript title to '*A centrally positioned cluster of multiple centrioles in antigen-presenting cells fosters T cell activation*' to avoid potential misunderstandings. Overall, we revised the text for clarity and precision, and we added a dedicated paragraph in the discussion that emphasizes both the novelty and the limitations of our study. Below, we provide a detailed, point-by-point response to the reviewers' comments and concerns. We highlighted all edits in the revised version in yellow (1st revision) and blue (2nd revision).

Response to Reviewer 1

Reviewer #1 (Remarks to the Authors):

The manuscript "Multiple clustered centrioles in antigen-presenting cells foster T cell activation without centrosome polarization" examines how centrosomal organization in dendritic cells contributes to immune synapse formation and activation of T cells. Overall, the manuscript addresses an important aspect of T cell activation and is timely and significant. In this revision and rebuttal, the authors have successfully addressed previous comments and improved the clarity of the manuscript. The incorporation of new adoptive transfer experiments strengthens the relevance of the authors' findings to more complex 3D environments. The authors now clearly distinguish between single-plane and projection images and have expanded the image analysis section for reproducibility, which now substantially improves transparency and replicability of their quantifications. The revised manuscript addresses most of the previous concerns and is substantially strengthened by the inclusion of additional experiments, improved figure clarity, and clearer methodological detail.

We thank reviewer 1 for highlighting the significance of our topic and the improvement of the manuscript.

One point of note: the current title gives the impression to the reader that T cell activation does not require centrosome polarization in T cells and is thus somewhat misleading. The authors should change the title to emphasize that centrosome reorientation to the immune synapse in antigen-presenting cells is not required.

*We appreciate this important comment and modified the manuscript title to '*A centrally positioned cluster of multiple centrioles in antigen-presenting cells fosters T cell activation*'. This version avoids potential misunderstandings between centrosome positioning in T cells and antigen-presenting cells while preserving the focus on the manuscripts' central findings.*

The gamma-tubulin intensity normalization methods add to the clarity. One follow-up question: were gamma-tubulin intensities normalized using the mean intensity for all two-centriole cells within the same treatment, or relative to all two-centriole cells across all treatments? Can the authors please clarify this point in the manuscript.

We thank the reviewer for this constructive comment. The values were normalized using the mean intensity for all two-centriole cells within the same treatment. We now clearly state this in the figure legend (p. 36; line 1214).

The authors responded constructively to concerns regarding potential off-target effects of PJ-34. By measuring MHCII expression, cytokine mRNA and protein levels, and testing multiple concentrations, they make a strong case that the observed effects are not due to off-target disruptions in antigen presentation or transcription. However, impacts on synapse organization or vesicle trafficking cannot be fully ruled out, and authors should acknowledge this in the discussion to strengthen the interpretation.

We thank the reviewer for this suggestion and included a section in the discussion regarding limitations of PJ-34 drug treatment (p. 17; line 602f).

PCM recruitment in co-cultured vs. non-co-cultured DCs: Were pericentrin and CDK5RAP2 levels enhanced in conjugated DCs with extra centrioles (as they were in non-conjugated DCs), or was that only the case for gamma-tubulin? In non-conjugated cells, three different PCM proteins (pericentrin, gamma-tubulin, and CDK5RAP2) were more highly expressed in DCs with extra centrioles. In line 197, it is stated that “enhanced PCM recruitment” was also observed in conjugated DCs with extra centrioles. However, only gamma-tubulin levels are presented for conjugated DCs. This should be explained better.

In the revised version of the manuscript, we now formulate precisely ‘enhanced γ -tubulin recruitment’ (p. 6; line 194f). In conjugated DCs we focused solely on γ -tubulin quantification. While we did quantify pericentrin and CDK5RAP2 in non-conjugated cells, these markers are better suited for identifying centrosomes qualitatively (e.g. see Fig. 6c).

Additionally, the enhancement in gamma-tubulin appears to be slightly more pronounced in non-conjugated DCs (Fig. 3d) compared to conjugated DCs (Fig. 3f), which raises the question of whether similar trends might be observed for the other two PCM proteins of interest. The authors’ assertion that enhanced PCM recruitment supports MTOC function hinges mainly upon the role of gamma-tubulin in promoting MT nucleation, so it is understandable why pericentrin and CDK5RAP2 results may have been omitted for concision, or perhaps not collected in favor of focusing on more critical experiments for this revision. (If the latter is true, I do not think it is critical to perform additional experiments). However, if the other two PCM proteins were indeed quantified in conjugated DCs and did not show the same enhancement, it would imply that enhanced PCM recruitment occurs to the same extent in the context of activation, and OVAp loading and/or co-culture conditions somehow affected PCM recruitment. A brief explanation would be useful.

In conjugated DCs we performed solely γ -tubulin quantification and not pericentrin or CDK5RAP2 quantification to save resources and focus on experiments which give new insights. γ -tubulin is the critical MT nucleating protein, which we further focused on. It would

certainly be very interesting to follow up on additional PCM proteins, and test whether there are differences regarding conjugation with T cells or without. However, this is not the primary focus of our study.

Minor notes:

- Line 160: Typo – change "extend" to "extent"
- Line 420: Typo – change "nucleaeted" to "nucleated"
- Line 599: Typo – change "immunogeniecy" to "immunogenicity"

We thank the reviewer for reading thoroughly our revised manuscript and corrected all spelling mistakes.

Response to Reviewer 2

Reviewer #2 (Remarks to the Authors):

The authors have addressed all the concerns I raised. I recommend acceptance of the revised manuscript.

We thank the reviewer for the positive evaluation of our work. We greatly appreciate the time and effort invested in reviewing our revised manuscript and we are pleased that our revisions addressed all concerns sufficiently.

Response to Reviewer 3

Reviewer #3 (Remarks to the Authors):

In the revised manuscript, NCOMMS-24-49110, Stotzel et al. have made efforts to revise the manuscript, including the addition of in vivo experiments. Some of the concerns raised in the previous version remain insufficiently addressed. The current version of the manuscript remains quite densely written and difficult to follow. The reviewer found the mathematical modeling to be a great strength of the paper.

*We thank the reviewer for their thorough evaluation of our revised manuscript. As recommended, we included further changes in the manuscript to improve the readers understanding. We also refer to major restructuring of our figures (shifting parts into the supplementary section and **now split Supplementary Fig. 4 and 5**) and our discussion section compared to the original version.*

1. In Fig. 1 b, what is the time period at which the quantification was performed? Additionally, it is unclear why the zero centrin population exists in centrinone-treated cells.

Centriole numbers were quantified in diploid mature dendritic cells on day 8 or 9 of culture. We added this information in the figure and figure legend (p. 36; line 1206). Mechanistically, Centrinone treatment prevents new procentriole formation via PLK4 inhibition (p.4; line 120f). Thus, during each division cycle in the course of DC differentiation, centriole duplication is blocked which results in a subset of cells that lack centrioles. Cell divisions might be

accomplished by acentriolar MTOC formation of cells lacking centrioles (Fig. 1e) as shown before (Wong et al. 2015).

2. The authors are encouraged to provide high-resolution images (Fig. 2a) for each condition.

Figure 2a displays MT depolymerization after pretubulysin treatment. We agree with the reviewer's suggestion and now included also high-resolution images for each condition (Figure 2a and Supplementary Fig. 2d; right panels).

3. In Fig. 2c, the reviewer is unclear about how the MT can have low straightness with decreasing MT length. With a known high persistence length, one would expect microtubules of short lengths to be straighter.

We thank the reviewer for this comment. There are two putative explanations, why short MTs show lower straightness:

- 1. It is possible that primarily those MTs remain after pretubulysin treatment that are both stable and particularly flexible. Acetylation has been associated with reduced microtubule stiffness (Portran et al., Nat Cell Biol, 2017; Xu et al., Science, 2017), and it has long been known that acetylated microtubules are more resistant to depolymerization.*
- 2. The second possibility is that pretubulysin induces lattice damage in MTs that do not depolymerize from their ends. Pretubulysin binds to β -tubulin, and its binding site is normally inaccessible within an intact lattice. However, pre-existing lattice defects - where an upper tubulin neighbor is missing - could allow pretubulysin to bind along the lattice, promoting local tubulin loss and, consequently, a reduction in MT stiffness.*

4. The reviewer is confused about why the pre-tubulysin data in the revised Fig. 2b is reduced compared to the original submission.

The number of MT filaments was re-analyzed by one single person by using a ROI with a defined area (25 μm^2) to count MT filaments (Supplementary Fig. 2c). The re-assessed data show a similar significant reduction in MT numbers in pretubulysin-treated cells compared to control cells.

5. In Fig. 2e, can the authors expand upon why the OX40+ data are missing in the pre-tubulysin data?

We apologize that we did not include this condition (DCs cultured in the absence of OVAp and treated with pretubulysin (wash-out and w/o wash-out)) as a control in this experiment in the first place. We repeated the respective experiment (including all conditions) and pooled the experiments. We observe no effect of pretubulysin treatment in the absence of OVAp (Fig. 2e).

6. One general concern is the quality of immunostained images. For example, it is difficult to distinguish between the number of centrioles. The authors are encouraged to provide high-resolution images that clearly show the distinction between the number of centrioles. The reviewer appreciates the challenges in imaging these structures.

*As suggested by the reviewer, we are providing high-resolution images in all figures discussing multiple centrioles (Fig. 3c, d, e, Fig. 4e, Fig. 6c, Fig. 7a, d, Suppl. Fig. 3a, b, **newly added** insets in Fig. 1b, c; Fig. 7d and Suppl. Fig. 7a, b). We would like to highlight that we are assessing the number of centrioles by inspecting all planes of a multi-z-stack, while displaying the centrioles in maximum projections (2D) in the manuscript (see Methods: image analysis p. 22; line 792f).*

7. In Fig. 3E, the γ -tubulin staining in the left ROI images for 'two centrioles' doesn't appear to correspond accurately with the merged image.

We carefully re-checked the ROI selection. We confirm that the same ROI was used consistently for both the individual channel images and the merged image. The appearance of grey foci in the merged image results from the overlay of green and magenta signals representing centrioles and microtubules, respectively.

8. The authors have not included CETN2-GFP data in Fig. 4C of the revised manuscript, despite a prior request. Furthermore, the authors should improve the quality of Fig. 6h.

We apologize for the misunderstanding and carefully checked the previous comment 6 of reviewer 3. Now, we addressed both requests by showing representative images of all timepoints analyzed (2h, 4h, 6h, 20h) and also included the CETN2-GFP channel separately. In this figure panel (Fig. 4c) we are demonstrating that T cell MTOC reorientation towards the DC occurs at all time points in an antigen-dependent manner which indicates IS formation in fixed samples of DC-T cell co-cultures. By displaying the CETN2-GFP channel, we visualize how the DC outline was marked. To prevent overcrowding of the figures we decided to shift the 2h and 6h timepoints to Supplementary Fig. 5a.

We adjusted brightness/contrast and conducted smoothing of the images. All details on image editing have been included in the methods section (image analysis p. 22; line 791f).

9. The authors should provide images in Fig. 4c, where the number of T cells interacting with the DC with and without OVAp is similar, and the associated analysis of the reorientation of MTOC in T cells.

We appreciate the comment and provide images, where the number of interacting T cells is comparable in the condition w/o OVAp and with OVAp (Fig. 4c and Suppl. Fig. 5a). Regarding the analysis, in Fig. 4c the percentage of T cells with reoriented MTOC is calculated across all interacting T cells within the respective condition (and not in relation to a single DC). To improve clarity, we included a more detailed description in the figure legend (p. 39; line 1302).

10. The reviewer is puzzled by the size difference between T cells and DCs, as shown in Fig. 6b. Aren't the DC cells supposed to be larger than the T cells, or is this an isolated case?

In Fig. 6b, a low-resolution overview image of a popliteal lymph node is presented. In this image, the T cell signal appears stronger than the CENT2-GFP signal, which might lead to different size estimation. Additionally, T cells typically have a rounded morphology while DCs exhibit protrusions within the tissue that are challenging to discern at this resolution. To clearly inspect the size differences between T cells and DCs, we refer to the high-resolution images in Fig. 6c.

11. It is encouraged to expand on the novelty of the work, while also acknowledging its shortcomings.

We thank the reviewer for this comment. We included a section in the discussion highlighting both the novelty and the limitations of our study (p. 17; line 600f).

We would also like to refer to one previous comment of reviewer 3:

In Figure 3F, the relative frequency of mono conjugation between T cell and DC cells does not reflect the centrosome numbers in Figure 3G in the same condition.

We carefully checked all numbers in Supplementary Fig. 3c. We realized that the relative frequency with- and without OVA has been miscalculated and corrected this mistake.

Reviewer #4 (Remarks to the Author):

Reviewer #5 (Remarks to the Author):

Point-by-point response on **NCOMMS-24-49110B:**
'A centrally positioned cluster of multiple centrioles in antigen-presenting cells fosters T cell activation'

We thank all reviewers for the time and effort invested in evaluating our revised manuscript entitled '*A centrally positioned cluster of multiple centrioles in antigen-presenting cells fosters T cell activation*'. We are pleased that our manuscript is accepted (in principle) for publication by *Nature Communications*.

Below, we provide our response to the remaining comment from Reviewer 3.

Response to Reviewer 3

Reviewer #3 (Remarks to the Authors):

The authors have, for the most part, addressed the issues. They might want to look into the following minor comment:

The authors propose (comment 1) that cell division may be facilitated by acentriolar microtubule-organizing center (MTOC) formation in cells lacking centrioles (Fig. 1e), as previously demonstrated (Wong et al. 2015; <https://doi.org/10.1126/science.aaa5111>). In Figure 1E, the authors do not present evidence of cell division in acentriolar cells. Furthermore, if the authors are citing Wang et al. (2015) in *Science*, the cited authors did not provide evidence of acentriolar division in immune cells. Moreover, normal cells enter a quiescent phase rather than dividing.

We agree with the reviewer that we did not provide evidence for acentriolar cell divisions in dendritic cells. Therefore, we did not include this statement in our final manuscript but we still cite Wong et al. 2015 who were the first ones using Centrinone for centriole depletion in cells.

The manuscript “Multiple clustered centrosomes in antigen-presenting cells foster T cell activation without MTOC polarization” examines the how centrosomal organization in dendritic cells contributes to immune synapse formation and activation of T cells. The authors combine imaging, cell biological manipulations and mathematical modeling to study how centrosome number, integrity and spatial positioning influence efficient T cell activation. Overall, the manuscript is well presented and the conclusions appear reasonable.

Some specific comments for the presented results

Figure 1:

Authors should specify whether the images are single-plane or maximum intensity projections. If single-plane images were used, the authors should justify how the imaging plane was selected.

B: (Right) *Quantification of centriole numbers after Centrinone treatment*

It is unclear how the authors define centriole detection through intensity thresholds. A description of how centrosomes were identified based on fluorescence intensity should be included. Also, what was the rationale for testing two centrinone concentrations (250 nM, 500 nM)? Was there a reason why certain panels only report results for one of the two concentrations? For instance, **Panel C**: Plotted results for 3 conditions (control, 250 nM, 500 nM centrinone), but only showed montages of 2 conditions (control & 250 nM centrinone). Perhaps this was done because there was no significant difference between the 250 nM and 500 nM conditions? This is not apparent from the presented results, since the effect of the two concentrations are not compared to each other in the beeswarm plot.

E: *Quantification of MTOCs (left) and MT filaments emanating from defined regions around centrosomes (right) in mature CETN2-GFP expressing BMDCs after Centrinone treatment*

Quantification of MTOCs and microtubule filaments is not clearly described. The authors should provide details on the image analysis methods used, including how regions were selected and how filament numbers were counted. Perhaps the authors could elaborate on how the gamma-tubulin intensity was normalized? It could just be my lack of understanding, but I found this a bit vague. For instance, lines 1054-1055: “Graph shows normalized values relative to cells with two centrioles” but does not elaborate on the normalization procedure. Perhaps the authors wish to account for the fact that all three treatments yield a mixed population of cells, all with different numbers of centrioles (as shown in panel B) but this should be specified.

F: (Left) *Quantification of proliferating T cells after Centrinone treatment according to CFSE labeling.*

There seems to be a clear interaction effect between the centrinone concentration and OVA loading (high OVA loading reduces the difference between the untreated and centrinone treated conditions for T cell proliferation. The interaction terms in the ANOVA should be examined more carefully to delineate how enhanced loading can overcome lack of centrioles.

The meaning of the dotted box and the axis units in panel F are unclear and should be clarified to improve the reader's understanding of the data presentation.

Figure 2:

F: *Quantification of T cell proliferation after co-culture with pretubulysin-treated BMDCs*

Line 155: “T cell activation was markedly reduced in the presence of pretubulysin (Fig. 2f)”

In line 155, the authors seem to equate T cell activation with CFSE dilution in Figure 2f, which measures proliferation. It is unclear that CFSE dilution can be used as a proxy for T cell activation, as proliferation and activation are distinct processes. Ideally, they should present data on early activation markers to strengthen their argument that pretubulysin impacts activation, rather than just proliferation.

The authors state that DC MTs had “not fully recovered” by 24h after tubulysin wash-out. Is there any quantitative data to support this statement? The right-hand plot in Figure 2C does provide evidence that

the tubulysin wash-out worked, but it's unclear what time point this plot represents. The authors should mention how much time passed between wash-out and quantification of MTs (Fig. 2C).

Is there any supplementary data that could show MT recovery at different time points? This point may be a bit nit-picky, but since the long duration of MT destabilization is what justified the authors' use of a wash-out prior to co-culture experiments (which is preferable because it allowed them to co-culture treated APCs and T cells without exposing T cells to the treatment).

Figure 2C: Is there a reason why no data is available for the no wash-out condition? Perhaps filaments too extensively disrupted to reliably count filaments?

Figure 3: (multiple centrioles are not needed for enhanced conjugate formation but are more effective?)

B/E: *Immunostaining and quantification of MT filaments*

The authors should clarify the method used to quantify microtubule filaments and explain how individual filaments were distinguished in crowded cellular regions. Additionally, details on the area or region where MT filaments were counted should be provided for reproducibility. What is the "defined area around the centrosome"?

Figure 4:

D: *Quantification of MTOC polarization towards the IS in T cells after different timepoints of co-culture*

How is it determined which T cells have a "reoriented centrosome"? Is the whole confocal stack used?

F. This is a bit confusing as the T cells express Nurr-GFP while the DCs express CEN2-GFP. How were the two signals differentiated to unambiguously identify T cells and DCs.

Figure 5:

B/C: *Quantifications of intracentrosomal and average distances in DCs with one and multiple centrosome(s) at different time points of co-culture*

The authors mention using a manual tracking plug-in to calculate intracentrosomal and intercentrosomal distances. However, they do not provide detailed criteria for defining what constitutes the boundaries of individual centrosomes or how the centroid positions were selected, which could introduce user bias.

Line 1171 (figure caption): "Each data point represents one cell derived from 5 independent experiments."

Minor spelling error-- "Each data point represents one cell"

D: *Sketches and confocal CETN2-GFP images indicating distances between centrosome center point (CP) and T cell CP, and DC CP and T cell CP in cells with one (upper) and multiple (lower) centrosomes.*

What is the quantitative criteria for defining center points of the centrosome/multiple centrosomes, T cell, and dendritic cell?

Also unclear whether centrosomal distance was computed in 3D from confocal stacks or from the maximum intensity projection.

- **Lines 262-264:** This is a minor detail, but I wonder whether the phrasing of this assertion makes sense:

"... average distances in cells with multiple centrosomes [...] did not show prominent differences between OVAp loaded and unloaded cells suggesting that multiple centrosomes **congregate together and cluster** during antigen-specific DC-T cell contacts."

- Does it make sense to assert that centrosomes actively congregated together, given that, to my understanding: (1) they did not actually observe this movement -- only the end result -- and (2) no clear differences were noted in the presence or absence of OVAp? If not, perhaps an alternate description could be "centrosomes *remained* clustered together...".
- Then again, perhaps the final centrosome position provides sufficient evidence to assert that clustering must have occurred, if clustering is not the dominant morphology in non-conjugated DCs.

Figure 6:

The confinement of DCs and T cells under an agarose block may mechanically restrict their movement and potentially alter the physiological dynamics of centrosome reorientation and immune synapse formation. The authors should comment on how/if this could impact centrosome configurations.

Line 625 (methods): "...DC-T cell co-cultures were injected under a block of agarose as previously described"

The cited paper describes a different set-up that includes two glass surfaces spaced by PDMS pillars.

Figure 7:

PARP enzymes play crucial roles in DNA repair, transcriptional regulation, and cellular metabolism. PARP inhibition could disrupt these processes in DCs, impairing antigen presentation or cytokine production, which would independently affect T cell activation. Is the observed reduction in T cell activation solely due to centrosome declustering, or might it also involve broader PARP-related cellular changes?

The authors should clarify why intracentrosomal distance was affected by PJ-34 treatment in non-co-cultured cells (panel B), but not in co-cultured cells (panel D). There is one way I could see this being relevant: in order to confidently assert that activation is impaired upon de-clustering of multi-numerous centrosomes, it seems important to emphasize that the de-clustering agent only affects **intercentrosomal** distances, not intracentrosomal distances (and thus, the results arise due to declustering of centrosomes, rather than abnormal spacing between centrioles in each individual centrosome). Still, as long as the authors can show that intracentrosomal distance was not affected in the context of the coculture experiment, I think they should be able to defend their assertion.

E: The authors should clarify whether these results drawn only from 2N cells with multiple centrosomes, or was data collected from all 2N cells (single and multiple centrosomes)?

Lines 303-306: "*Interestingly, when co-culturing with either PJ-34-treated or control OVAp-loaded DCs with OT-II T cells, we found that **in the presence of de-clustered centrosomes**, T cell activation was significantly diminished compared to control cells.*"

This is relevant because when centrosomes were counted in figure 3A, it seemed as though multi-centrosome cells only comprised ~25% of all 2N cells (centriole number ≥ 3). Therefore, in a population of single- and multi-centrosome cells, I wonder whether any effects on multi-centrosome cells would be prominent enough to yield the significant differences shown in Figure 7E. If not, perhaps there is some other mechanism at play by which PJ-34 could affect the ability of DCs to stimulate T cell proliferation?

Figure 8:

B: Plot of $\langle d_{short} \rangle$ vs h_{CS} in the presence of a nucleus located centrally to the cell

Line 340: "Our data indicate an optimal perinuclear positioning of the centrosome, marginally shifted from the nuclear surface by $< 1 \mu m$ "

A visual indicator of the minimum $\langle d_{short} \rangle$ on plot B would better emphasize the optimal positioning, as the x-axis scale makes it difficult to identify.

General comments: While the authors report that DCs with multiple centrosomes exhibit increased cytokine secretion, the referenced study lacks temporal data on when these cytokines are released relative to IS formation. To establish a direct link between MT contact and T cell activation, the authors should investigate the time-scale of cytokine release using real-time measurements. Alternatively, they should suggest this as a future direction to determine whether early MT contacts trigger cytokine exocytosis or if other mechanisms contribute to the observed increase in T cell activation.

- Minor typos –

- Line 61-62: “Yet, B cells have a rather week T cell priming capacity” → “weak”
 - Line 119: poly-L-lysin -> poly-L-lysine
 - Line 150-151: “thus allowing to study the role of MTs...” → “thus allowing us to study...”
 - Line 205: promotor -> promoter
-